# FAIRNESS THROUGH MATCHING
## FOR BETTER GROUP FAIRNESS

## ABSTRACT

Group unfairness, which refers to socially unacceptable bias favoring certain groups (e.g., white, male), is a frequently observed ethical concern in AI. Various algorithms have been developed to mitigate such group unfairness in trained models. However, a significant limitation of existing algorithms for group fairness is that trained group-fair models can discriminate against specific subsets or not be fair for individuals in the same sensitive group. The primary goal of this research is to develop a method to find a good group-fair model in the sense that it discriminates less against subsets and treats individuals in the same sensitive group more fairly. For this purpose, we introduce a new measure of group fairness called Matched Demographic Parity (MDP). An interesting feature of MDP is that it corresponds a matching function (a function matching two individuals from two different sensitive groups) to each group-fair model. Then, we propose a learning algorithm to seek a group-fair model whose corresponding matching function matches similar individuals well. Theoretical justifications are fully provided, and experiments are conducted to illustrate the superiority of the proposed algorithm.

## 1 INTRODUCTION

AI (Artificial Intelligence) technologies have become increasingly prevalent as crucial decision-making tools for human beings across diverse areas, including credit scoring, criminal risk assessment, and college admissions. As we train models based on observed data from the real world, any biases presenting in the data can significantly influence the trained models. This becomes a particular concern when the observed data contain unfair historical biases, because it can lead to unfair decisions (Calders et al., 2009; Feldman et al., 2015; Angwin et al., 2016; Barocas & Selbst, 2016; Chouldechova, 2016; Kleinberg et al., 2018; Mehrabi et al., 2019; Zhou et al., 2021). Unfair preferences for specific groups such as white individuals or men have been reported (Angwin et al., 2016; Ingold & Soper, 2016; Dua & Graff, 2017).

Under such circumstances, ensuring fairness in AI decision-making becomes a socially crucial mission. In response, plenty of algorithms have been developed to mitigate such bias by searching models that treat sensitive groups similarly in some sense. For example, the ratio of positive predictions for each sensitive group is required to be similar (Calders et al., 2009; Feldman et al., 2015; Barocas & Selbst, 2016; Zafar et al., 2017; Donini et al., 2018; Agarwal et al., 2018; Quadrianto et al., 2019).

However, a well-known problem of group-fair models is their potential risk of discrimination against certain subsets or individuals, even when they are fair at the group level as a whole (Dwork et al., 2012; Kearns et al., 2018; Hebert-Johnson et al., 2018; Okati et al., 2023). For instance, Dwork et al. (2012) argued such problems, including subset targeting and self-fulfilling prophecy, and subsequently introduced the concept of individual fairness.

The aim of this paper is to find a group-fair model that discriminates less between sensitive groups within a subset or among individuals in the same sensitive group. For this purpose, we propose a new group fairness measure, *Matched Demographic Parity (MDP)*, using a matching function designed to match two individuals from different groups. Remarkably, we show that MDP is closely related to the well-known group fairness measures, such as the strong demographic parity with the total variation or Wasserstein distance.

An interesting observation is that undesirable discrimination between sensitive groups within a subset or among individuals in the same sensitive group occurs when the matching function matches dissimilar individuals. Based on this result, we develop a learning algorithm to seek a group-fair model whose corresponding matching function matches similar individuals well.

The proposed algorithm consists of two steps: (1) learning a relaxed optimal transport (OT) map (Monge, 1781; Kantorovich, 2006) that is to be used as a desirable matching function, and (2) searching for the most accurate model among those treating similarly two matched individuals by the relaxed OT map. We call this two-step procedure as *Fairness Through Matching (FTM)*.

It is interesting that FTM is aligned with Fair Representation Learning (FRL). Conceptually, FTM is understood as an FRL method where the OT map is used as the representation encoder. Empirical results show that FTM offers improvement in accuracy and flexibility compared to FRL.

**Main contributions**

**1.** We propose a new group fairness measure, MDP, and prove that it is closely related to the strong demographic parity measure with the total variation or Wasserstein distance.

**2.** We propose a new group fairness algorithm, FTM, which uses the MDP constraint to find a group-fair model discriminating less between subsets or individuals in the same sensitive group.

**3.** Theoretical justifications are provided: FTM (1) achieves group fairness asymptotically, and (2) improves subset/within-group fairness, which are two representative fairness concepts for discriminating less between subsets or individuals in the same sensitive group of a given group-fair model.

**4.** Experiments on benchmark datasets illustrate that FTM (1) improves subset/within-group fairness much without significant degradation in prediction accuracy and (2) is superior to FRL algorithms in terms of accuracy and flexibility in model selection.

## 1.1 Related works and backgrounds

**Fair AI algorithms** Various notions of group fairness such as Demographic Parity (DP) and Strong Demographic Parity (SDP) (Calders et al., 2009; Feldman et al., 2015; Agarwal et al., 2019; Chzhen et al., 2020; Jiang et al., 2020), as well as Equal OPportunity (EOP) and Equalized Odds (EO) (Hardt et al., 2016) have been introduced, and corresponding learning algorithms have been developed (Zafar et al., 2019; Donini et al., 2018; Madras et al., 2018; Chuang & Mroueh, 2021).

To resolve the problem of a group-fair model discriminating between subsets or individuals, other fairness notions are introduced, including individual fairness (Dwork et al., 2012; Yona & Rothblum, 2018; Yurochkin & Sun, 2021; Petersen et al., 2021) and counterfactual fairness (Kusner et al., 2017; Chiappa & Gillam, 2018; von Kügelgen et al., 2022; Nilforoshan et al., 2022). However, individual fairness does not guarantee group fairness when the gap between the two sensitive groups is large, whose detailed discussion is in Section C.7 of Appendix. The use of counterfactual fairness is limited as it requires causal models which are not easy to be obtained solely from observed data.

**Optimal Transport (OT) map** Optimal transport theory is initially formulated by Monge (1781). For given source and target distributions $\mathcal{P}, \mathcal{Q}$ in $\mathbb{R}^d$ and a cost function $c$ (e.g., $L_2$ distance), the OT map from $\mathcal{P}$ to $\mathcal{Q}$ is the solution of $\min_{\mathbf{T}:\mathbf{T}_\#\mathcal{P}=\mathcal{Q}} \mathbb{E}_{\mathbf{X}\sim\mathcal{P}}\left(c\left(\mathbf{X}, \mathbf{T}(\mathbf{X})\right)\right)$, where $\mathbf{T}_\#\mathcal{P}$ is the push-forward measure of $\mathcal{P}$ induced by a given map $\mathbf{T}: \text{Supp}(\mathcal{P}) \to \mathbb{R}^d$. Kantorovich (2006) modified this problem by finding the optimal coupling between two empirical measures. Subsequently, efficient optimization algorithms have been developed with regularizations (Cuturi, 2013; Genevay et al., 2016). Learning the optimal transport map with a functional form instead has also received much attention (Seguy et al., 2018; Yang & Uhler, 2019; Hütter & Rigollet, 2021). Several studies such as Gordaliza et al. (2019); Chzhen et al. (2020); Jiang et al. (2020) have applied the optimal transport theory to fair AI. More details of the OT map are presented in Section C.1 of Appendix.

## 1.2 Preliminaries

**Notations** We denote $\mathbf{X} \in \mathcal{X} \subset \mathbb{R}^d$ as the random vector, $Y \in \mathcal{Y} = \{-1, 1\}$ as the binary label, and $S \in \mathcal{S} = \{0, 1\}$ as the binary sensitive variable. Let $\mathcal{D} = \{(\mathbf{x}_i, y_i, s_i)\}_{i=1}^n$ be a set of training data of size $n$ which are independent copies of the random tuple $(\mathbf{X}, Y, S)$ on $\mathcal{X} \times \mathcal{Y} \times \mathcal{S}$. Whenever

necessary, we split the domain of $\mathbf{X}$ with respect to $S$ and write $\mathcal{X}_s$ as the domain of $\mathbf{X}|S = s$. For a given $s \in \{0, 1\}$, we denote $s' = 1 - s$.

Denote $\mathcal{P}$ the joint distribution of $(\mathbf{X}, Y, S)$, $\mathcal{P}_{\mathbf{X}}$ the marginal distribution of $\mathbf{X}$, and $\mathcal{P}_s = \mathcal{P}_{\mathbf{X}|s}, s \in \{0, 1\}$ the conditional distributions of $\mathbf{X}|S = s$. Similarly, we write the conditional expectation as $\mathbb{E}_s(\cdot) = \mathbb{E}(\cdot|S = s)$, and $\mathbb{E}_{n,s}$ as its empirical counterpart.

We consider group-fair binary classification algorithms which yield a real-valued function $f \in \mathcal{F} \subset \{f : \mathcal{X} \times \mathcal{S} \to \mathbb{R}\}$. Let $\mathcal{F}_s = \{f(\cdot, s) : f \in \mathcal{F}\}, s \in \{0, 1\}$ and $f^\star$ be the optimal model defined by $f^\star := \arg\min_{f \in \mathcal{F}} \mathbb{E}l(Y, f(\mathbf{X}, S))$ for a given loss function $l : \{-1, 1\} \times \mathbb{R} \to \mathbb{R}_+$.

**Integral Probability Metric (IPM)**   Integral Probability Metric (IPM) is a metric between two probability measures. For a real-valued discriminator class $\mathcal{H}$, the IPM between two given probability measures $\mathcal{P}$ and $\mathcal{Q}$ on $\mathcal{X}$ is defined as $d_{\mathcal{H}}(\mathcal{P}, \mathcal{Q}) := \sup_{h \in \mathcal{H}} \left| \int h(\mathbf{X}) \, d\mathcal{P}(\mathbf{X}) - \int h(\mathbf{Y}) \, d\mathcal{Q}(\mathbf{Y}) \right|$. Three popularly known IPMs are Wasserstein distance, MMD (Maximum Mean Discrepancy), and TV (Total Variation), where $\mathcal{H}$ is the set of all 1-Lipschitz functions, the unit ball of an RKHS, and all measurable functions in $\{h : \mathcal{X} \to [0, 1]\}$, respectively (Sriperumbudur et al., 2009).

**Fairness measures**   In this paper, we consider the demographic parity as the notion of group fairness (see Section C.6 of Appendix for equal opportunity). Let $\Delta\mathrm{DP}(f; \tau) = |\mathcal{P}(f(\mathbf{X}, S) > \tau|S = 0) - \mathcal{P}(f(\mathbf{X}, S) > \tau|S = 1)|$. The original demographic parity $\Delta\mathrm{DP}(f)$ (Feldman et al., 2015; Zafar et al., 2017) is equal to $\Delta\mathrm{DP}(f; 0)$, which measures the difference in the ratio of positive predictions between the two groups. A smoother version of $\Delta\mathrm{DP}(f)$ is the mean score parity defined as $\Delta\overline{\mathrm{DP}}(f) = |\mathbb{E}(f(\mathbf{X}, S)|S = 0) - \mathbb{E}(f(\mathbf{X}, S)|S = 1)|$ (Madras et al., 2018; Agarwal et al., 2018; Chuang & Mroueh, 2021; Buyl & Bie, 2022). In addition, we consider the following three measures for strong demographic parity (Agarwal et al., 2019; Chzhen et al., 2020; Jiang et al., 2020): $\Delta\mathrm{SDP}(f) = \mathbb{E}_{\tau \sim U(-1,1)}\Delta\mathrm{DP}(f; \tau)$, $\Delta\mathrm{TVDP}(f) := d_{\mathrm{TV}}\left(\mathcal{P}_{f(\mathbf{X},0)|S=0}, \mathcal{P}_{f(\mathbf{X},1)|S=1}\right)$ and $\Delta\mathrm{WDP}(f) = d_{\mathcal{L}_1}\left(\mathcal{P}_{f(\mathbf{X},0)|S=0}, \mathcal{P}_{f(\mathbf{X},1)|S=1}\right)$, where $\mathcal{P}_{f(\mathbf{X},s)|S=s}$ is the distribution of $f(\mathbf{X}, s)|S = s$ and $d_{\mathrm{TV}}$ and $d_{\mathcal{L}_1}$ are the total variation and 1-Wasserstein distances, respectively.

## 2   Matched Demographic Parity

We propose a new group fairness measure called **Matched Demographic Parity (MDP)**, a measure for strong demographic parity. Remarkably, it provides a new perspective on group fairness in the sense that strong group-fair models inherently adhere to specific mechanisms and constraints. Moreover, this finding serves as the main inspiration for our proposed algorithm.

Let $\mathcal{T}_s := \{\mathbf{T}_s : \mathcal{X}_s \to \mathcal{X}_{s'}\}$ be a given set of maps from $\mathcal{X}_s$ to $\mathcal{X}_{s'}$. We say $\mathbf{T}_s \in \mathcal{T}_s$ is a **matching function**[1] if $\mathbf{T}_{s\#}\mathcal{P}_s = \mathcal{P}_{s'}$. For a given model $f \in \mathcal{F}$, the MDP measure is defined by

$$\Delta\mathrm{MDP}(f) := \inf_{s \in \{0,1\}} \inf_{\mathbf{T}_s \in \mathcal{T}_{s,0}} \mathbb{E}_s |f(\mathbf{X}, s) - f(\mathbf{T}_s(\mathbf{X}), s')|$$

where $\mathcal{T}_{s,0} := \{\mathbf{T}_s \in \mathcal{T}_s : \mathbf{T}_{s\#}\mathcal{P}_s = \mathcal{P}_{s'}\}$ is the class of matching functions.

Proposition 2.1 below shows that ***MDP can be considered as a measure for strong demographic parity*** under regularity conditions, whose proof is in Section B.1 of Appendix. Let $\mathcal{F}^\Delta(\delta) := \{f \in \mathcal{F} : \Delta(f) \leq \delta\}$ be the set of group-fair models for the fairness level $\delta \geq 0$ with respect to a group fairness measure $\Delta : \mathcal{F} \to \mathbb{R}^+ \cup \{0\}$. We assume that (C1) $\mathcal{F}$ is the collection of bounded functions; (C2) $\mathcal{P}_s, s = 0, 1$, is absolutely continuous.

**Proposition 2.1.** *Under* (C1) *and* (C2)*, for any given fairness level $\delta \geq 0$, we have $\mathcal{F}^{\Delta\mathrm{TVDP}}(C\delta) \subset \mathcal{F}^{\Delta\mathrm{MDP}}(\delta) \subset \mathcal{F}^{\Delta\mathrm{WDP}}(\delta)$ for some constant $C > 0$.*

## 3   Group fairness through the optimal matching

We present an example of problematic group-fair models (Section 3.1) and then suggest an idea of using a restricted class of matching functions for searching desirable group-fair models (Section

---

[1]The matching function is equivalent to the 'transport map' in Villani (2008). We use the term 'matching function' since we use it to match similar individuals.

3.2). Then, we propose the corresponding algorithm, FTM (Section 3.3), and discuss several closely related methods to FTM (Section 3.4).

## 3.1 AN EXAMPLE OF PROBLEMATIC GROUP-FAIR MODELS

We present an example of a group-fair model that discriminates subsets seriously. Consider the following model, which is visualized in Figure 1.

- $\mathbf{X}|S = s \sim \text{Unif}(0,1), s \in \{0,1\}$.

- $\widehat{f}(\mathbf{x}, s) = \text{sign}(2\mathbf{x}-1)(1-2s)$.

Note that $\widehat{f}$ is perfectly fair (i.e. $\widehat{f}(\mathbf{X},0) \overset{d}{=} \widehat{f}(\mathbf{X},1)$). However, there is a notable issue with the treatment of individuals in the subset $\{\mathbf{x} \geq 1/2\}$ (as well as $\{\mathbf{x} < 1/2\}$); $\widehat{f}$ predicts label 1 for $\mathbf{x} \geq 1/2, s = 0$ and $-1$ for $\mathbf{x} \geq 1/2, s = 1$. That is, $\widehat{f}$ discriminates individuals belonging to the subset $\{\mathbf{x} \geq 1/2\}$ (and $\{\mathbf{x} < 1/2\}$).

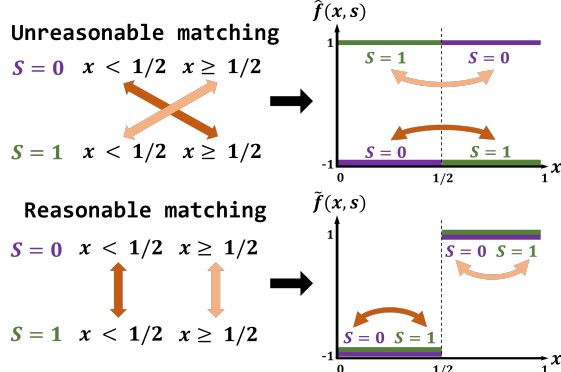

Figure 1: (Top) Example of problematic group-fair model with a unreasonable matching function. (Bottom) Mitigation of discrimination over subsets using a reasonable matching function.

This issue can be attributed to the unreasonable matching function of $\widehat{f}$. For given $f$, we define the corresponding matching function as $\mathbf{T}_s^f := \arg\min_{\mathbf{T}_s \in \mathcal{T}_{s,0}} \mathbb{E}_s |f(\mathbf{X}, s) - f(\mathbf{T}_s(\mathbf{X}), s')|$ if exists. It turns out that the matching function correspond to $\widehat{f}$ is $\mathbf{T}_s^{\widehat{f}}(\mathbf{x}) = \mathbf{x} - \text{sign}((2\mathbf{x}-1)(1-2s))/2$. This function matches an individual in $\{\mathbf{x} < 1/2, S = s\}$ with one in $\{\mathbf{x} \geq 1/2, S = s'\}$, who are far apart from each other.

In contrast, $\tilde{f}(\mathbf{x}, s) = \text{sign}(2\mathbf{x} - 1)$ is perfectly fair but does not discriminate any subset. Note that the corresponding matching function for $\tilde{f}$ turns out to be $\mathbf{T}_s^{\tilde{f}}(\mathbf{x}) = \mathbf{x}$. If we compare the transport cost $\mathbb{E}_s \|\mathbf{X} - \mathbf{T}_s^{\widehat{f}}(\mathbf{X})\|^2$ of $\widehat{f}$ with that of $\tilde{f}$, we can see that the former is much larger than the latter. Refer to Figure 1 for overall illustration of this example.

## 3.2 MDP WITH RESTRICTED CLASSES OF MATCHING FUNCTIONS

The example in the previous subsection implies that desirable group-fair models can be discerned by examining the corresponding matching functions. For desirable matching functions, we consider one that matches **similar** individuals, i.e., $\mathbf{x} \approx \mathbf{T}_s(\mathbf{x})$. The OT map from $\mathcal{P}_s$ to $\mathcal{P}_{s'}$, denoted as $\mathbf{T}_s^\star$, is the best one since it minimizes the transport cost $\mathbb{E}_s \|\mathbf{X} - \mathbf{T}_s(\mathbf{X})\|^2$ among all matching functions.

For better group fairness, we only consider group-fair models whose corresponding matching functions are close to the OT map. For given $\delta > 0$ and $\gamma > 0$, we define

$$\mathcal{F}^{\Delta\text{MDP}}(\delta; \mathcal{T}_{0,0}(\gamma), \mathcal{T}_{1,0}(\gamma)) := \{f \in \mathcal{F} : \inf_{s \in \{0,1\}} \inf_{\mathbf{T}_s \in \mathcal{T}_{s,0}(\gamma)} \mathbb{E}_s |f(\mathbf{X}, s) - f(\mathbf{T}_s(\mathbf{X}), s')| \leq \delta\},$$

where $\mathcal{T}_{s,0}(\gamma) := \{\mathbf{T}_s \in \mathcal{T}_{s,0} : \mathbb{E}_s \|\mathbf{T}_s(\mathbf{X}) - \mathbf{T}_s^\star(\mathbf{X})\|^2 \leq \gamma\} \subseteq \mathcal{T}_{s,0}$ for $s \in \{0,1\}$ are restricted classes of matching functions. Then, we search an accurate prediction model among those in $\mathcal{F}^{\Delta\text{MDP}}(\delta; \mathcal{T}_{0,0}(\gamma), \mathcal{T}_{1,0}(\gamma))$. This restriction would help to learn group-fair models that discriminate less between subsets or individuals in the same sensitive group, which is supported by Theorems 3.1 and 3.2.

**Improvement in terms of subset and within-group fairness** We show that the MDP with restricted classes of matching functions can lead to improvement in subset and within-group fairness, both of which are frequently violated when focusing solely on group fairness (Dwork et al., 2012; Kearns et al., 2018; Hebert-Johnson et al., 2018; Binns, 2020; Kim et al., 2023).

**(1) Subset fairness** (Dwork et al., 2012; Kearns et al., 2018): The measure for subset fairness with respect to a collection $\mathcal{C}$ of bounded subsets of $\mathcal{X}$ is defined as $\Delta\overline{\mathrm{DP}}_\mathcal{C}(f) := \sup_{A \in \mathcal{C}} |\mathbb{E}(f(\mathbf{X}, 0)|S = 0, \mathbf{X} \in A) - \mathbb{E}(f(\mathbf{X}, 1)|S = 1, \mathbf{X} \in A)|$. We note that this is a general formulation of the 'subset targeting' in Dwork et al. (2012). We expect that a group-fair model whose corresponding matching function has a lower transport cost will have a higher subset fairness. It is because the chance of two matched individuals (from different sensitive groups) belonging to a given subset increases as the transport cost decreases. Theorem 3.1 theoretically supports this conjecture.

**Theorem 3.1** (Improvement in subset fairness). *Suppose $\mathcal{F}$ is the collection of L-Lipschitz functions. Then, for all $f \in \mathcal{F}^{\Delta\mathrm{MDP}}(\delta)$, we have $\Delta\overline{\mathrm{DP}}_\mathcal{C}(f) \leq L \left( \min_s \left( \mathbb{E}_s \|\mathbf{X} - \mathbf{T}_s^f(\mathbf{X})\|^2 \right)^{1/2} + U_1 \right) + U_2\delta$, where $U_1, U_2 > 0$ are constants only depending on $\mathcal{C}$ and $\mathcal{P}_s, s = 0, 1$.*

**(2) Within-group fairness** (Kim et al., 2023; Okati et al., 2023): Suppose that there are two individuals, $\mathbf{x}_s^{(1)}$ and $\mathbf{x}_s^{(2)}$, belonging to the same sensitive group $s$ and that the unfair but optimal prediction model prefers $\mathbf{x}_s^{(1)}$ to $\mathbf{x}_s^{(2)}$. In this case, within-group fairness requires that a group-fair model should also prefer $\mathbf{x}_s^{(1)}$. That is, a group-fair model should not treat individuals in the same sensitive group unfairly. It is a general formulation of the 'self-fulfilling prophecy' in Dwork et al. (2012).

The concept of within-group fairness can be formulated as follows. Suppose that $\mathcal{X}_s = \mathcal{X}_{s'} = \mathcal{X}$. For any $\mathbf{x}^{(1)}$ and $\mathbf{x}^{(2)}$ in $\mathcal{X}$ such that $f^\star(\mathbf{x}^{(1)}, s) > f^\star(\mathbf{x}^{(2)}, s)$ and $f^\star(\mathbf{x}^{(1)}, s') > f^\star(\mathbf{x}^{(2)}, s')$, if there exists $W_s(\mathbf{x}^{(1)}, \mathbf{x}^{(2)}, f)$ such that $f(\mathbf{x}^{(1)}, s) - f(\mathbf{x}^{(2)}, s) \geq W_s(\mathbf{x}^{(1)}, \mathbf{x}^{(2)}, f)$ for a given model $f$, then we say $f$ is within-group fair supported by $W_s(\mathbf{x}^{(1)}, \mathbf{x}^{(2)}, f)$. A larger value of $W_s(\mathbf{x}^{(1)}, \mathbf{x}^{(2)}, f)$ means higher within-group fairness. Theorem 3.2 shows that reducing the transport cost is helpful to improve within-group fairness. Assume that $\mathcal{P}(S = 0) = \mathcal{P}(S = 1)$, $l$ be the squared loss and $\mathcal{F}$ consists of all measurable functions. For given two matching functions $\mathbf{T} = (\mathbf{T}_s \in \mathcal{T}_{s,0}, s \in \{0, 1\}$, let $f_\mathbf{T}$ be the minimizer of $\mathbb{E}(Y - f(\mathbf{X}, S))^2$ on $\{f \in \mathcal{F} : \min_s \mathbb{E}_s |f(\mathbf{X}, s) - f(\mathbf{T}_s(\mathbf{X}), s')| = 0\}$. Assume that $\mathbf{T}_s, s \in \{0, 1\}$ are invertible. Let $\mathcal{X}^2(f^\star) := \{(\mathbf{x}^{(1)}, \mathbf{x}^{(2)}) \in \mathcal{X} \times \mathcal{X} : f^\star(\mathbf{x}^{(1)}, s) > f^\star(\mathbf{x}^{(2)}, s), f^\star(\mathbf{x}^{(1)}, s') > f^\star(\mathbf{x}^{(2)}, s')\}$.

**Theorem 3.2** (Improvement in within-group fairness). *Let $\mathbf{x}^{(1)}$ and $\mathbf{x}^{(2)}$ be two individuals such that $(\mathbf{x}^{(1)}, \mathbf{x}^{(2)}) \in \mathcal{X}^2(f^\star)$. Suppose that $f^\star$ is L-Lipshitz. Then, for $s = 0, 1$, we have*

$$f_\mathbf{T}(\mathbf{x}^{(1)}, s) - f_\mathbf{T}(\mathbf{x}^{(2)}, s) > \frac{M_s}{2} - \frac{L}{2} \max \left( \sum_{i=1,2} \|\mathbf{x}^{(i)} - \mathbf{T}_s(\mathbf{x}^{(i)})\|, \sum_{i=1,2} \|\mathbf{x}^{(i)} - \mathbf{T}_{s'}^{-1}(\mathbf{x}^{(i)})\| \right),$$

*for $M_s = M_s(\mathbf{x}^{(1)}, \mathbf{x}^{(2)}) := f^\star(\mathbf{x}^{(1)}, s) - f^\star(\mathbf{x}^{(2)}, s) > 0$.*

Theorem 3.2 implies that within-group fairness of $f_\mathbf{T}$ is expected to be improved when the transport costs of $\mathbf{T}_s, s \in \{0, 1\}$ become smaller. Let $f_\gamma$ be the minimizer of $\mathbb{E}(Y - f(\mathbf{X}, S))^2$ among $f \in \mathcal{F}^{\Delta\mathrm{MDP}}(0; \mathcal{T}_{0,0}(\gamma), \mathcal{T}_{1,0}(\gamma))$, where $\mathcal{T}_{s,0}(\gamma), s \in \{0, 1\}$ consist of invertible matching functions. Then we have $f_\gamma = f_{\mathbf{T}^\gamma}$, where $\mathbf{T}^\gamma = \mathrm{argmin}_{\mathbf{T}_s \in \mathcal{T}_{s,0}(\gamma), s \in \{0,1\}} \mathbb{E}_s |f_\gamma(\mathbf{X}, s) - f_\gamma(\mathbf{T}_s(\mathbf{X}), s')|$. Thus, within-group fairness of $f_\gamma$ is expected to increase as $\gamma$ decreases.

The proofs of Theorems 3.1 and 3.2 are presented in Sections C.2 and C.3 of Appendix, respectively, and numerical evidence is given in Section 4.2.

## 3.3 FTM ALGORITHM

Unfortunately, obtaining $\mathcal{T}_{s,0}(\gamma)$ is nearly infeasible due to the limited access to the entire set of matching functions, i.e., $\mathcal{T}_{s,0}$. Instead, for practical implementation, we propose to use the relaxed OT map, which is a modified version of the OT map. For a given $\epsilon \geq 0$, the ($\epsilon$-)**relaxed OT map** $\mathbf{T}_{s,\epsilon}^\star$ is defined as the minimizer of $\mathbb{E}_s \|\mathbf{X} - \mathbf{T}_s(\mathbf{X})\|^2$ among all $\mathbf{T}_s$ satisfying $d_\mathcal{H}(\mathbf{T}_{s\#}\mathcal{P}_s, \mathcal{P}_{s'}) \leq \epsilon$. Instead of $\mathcal{F}^{\Delta\mathrm{MDP}}(\delta; \mathcal{T}_{0,0}(\gamma), \mathcal{T}_{1,0}(\gamma))$, we consider $\mathcal{F}^{\Delta\mathrm{MDP}}(\delta; \{\mathbf{T}_{0,\epsilon}^\star\}, \{\mathbf{T}_{1,\epsilon}^\star\})$. Even though $\{\mathbf{T}_{s,\epsilon}^\star\}$ consists of a single element, $\mathcal{F}^{\Delta\mathrm{MDP}}(\delta; \{\mathbf{T}_{0,\epsilon}^\star\}, \{\mathbf{T}_{1,\epsilon}^\star\})$ is not overly restrictive. This claim is supported by Proposition 3.3 below, which shows that $\mathcal{F}^{\Delta\mathrm{MDP}}(\delta; \mathcal{T}_{0,0}(\gamma), \mathcal{T}_{1,0}(\gamma))$ is close to $\mathcal{F}^{\Delta\mathrm{MDP}}(\delta; \{\mathbf{T}_{0,\epsilon}^\star\}, \{\mathbf{T}_{1,\epsilon}^\star\})$ when $\gamma, \epsilon$ are small. The proof is in Section B.2 of Appendix.

**Proposition 3.3.** *For any given fairness level $\delta \geq 0$, there exist $\delta_1, \delta_2$ depending on $\delta, \gamma, \epsilon$ such that $\mathcal{F}^{\Delta\text{MDP}}(\delta_1; \{\mathbf{T}_{0,\epsilon}^\star\}, \{\mathbf{T}_{1,\epsilon}^\star\}) \subset \mathcal{F}^{\Delta\text{MDP}}(\delta; \mathcal{T}_{0,0}(\gamma), \mathcal{T}_{1,0}(\gamma)) \subset \mathcal{F}^{\Delta\text{MDP}}(\delta_2; \{\mathbf{T}_{0,\epsilon}^\star\}, \{\mathbf{T}_{1,\epsilon}^\star\})$. Furthermore, $\delta_1, \delta_2 \to \delta$ as $\gamma, \epsilon \to 0$.* [2]

Another justification of using $\mathcal{F}^{\Delta\text{MDP}}(\delta; \{\mathbf{T}_{0,\epsilon}^\star\}, \{\mathbf{T}_{1,\epsilon}^\star\})$ is its relation with FRL. Conceptually, FTM can be interpreted as an algorithm of FRL where the relaxed OT map is used as the representation encoder. See Section 3.4 for details.

Regularized OT maps (Cuturi, 2013; Alaya et al., 2019), which are similar to the relaxed OT map, have been developed and widely applied. However, these maps are only defined on the training dataset and require recalculation for test data, leading us to consider the relaxed OT map instead. A detailed comparison between the relaxed and regularized OT maps is in Section C.1 of Appendix.

We call the learning algorithm with MDP constraint whose matching function is the relaxed OT map as **F**airness **T**hrough **M**atching (FTM). FTM consists of the following two steps. Let $\epsilon$ (IPM level) and $\delta$ (fairness level) be two given positive constants.

**(STEP 1)** We train the matching networks $\widehat{\mathbf{T}}_s, s \in \{0, 1\}$ using $\mathcal{D}$ as

$$\widehat{\mathbf{T}}_s = \widehat{\mathbf{T}}_{s,\epsilon} := \underset{\mathbf{T}_s \in \mathcal{T}_s}{\arg\min} \, \mathbb{E}_{n,s} \|\mathbf{X} - \mathbf{T}_s(\mathbf{X})\|^2 \text{ s.t. } d_{\mathcal{H}}(\mathbf{T}_{s\#}\mathcal{P}_{n,s}, \mathcal{P}_{n,s'}) \leq \epsilon \tag{1}$$

where $\mathcal{T}_s$ and $\mathcal{H}$ are given classes of transport maps and discriminator functions, respectively. To make the optimization implementable, we choose DNN (Deep Neural Network) function classes $\mathcal{T}_n^{\text{DNN}}$ and $\mathcal{H}_n^{\text{DNN}}$ as $\mathcal{T}_s$ and $\mathcal{H}$, respectively, whose details are in Theorem B.5 and its proof.

**(STEP 2)** We train FTM classifiers $\widehat{f}_s^{\text{FTM}}, s \in \{0, 1\}$ using $\mathcal{D}$ as

$$\widehat{f}_s^{\text{FTM}} := \underset{f \in \mathcal{F}}{\arg\min} \, \mathbb{E}_n l(Y, f(\mathbf{X}, S)) \text{ s.t. } \text{REG}_s(f) \leq \delta \tag{2}$$

where $\mathcal{F}$ is a given class of prediction models, $l$ is a given loss function, such as the cross-entropy, and $\text{REG}_s(f) := \mathbb{E}_{n,s} |f(\mathbf{X}, s) - f(\widehat{\mathbf{T}}_s(\mathbf{X}), s')|$. Then, we select one from $\{\widehat{f}_0^{\text{FTM}}, \widehat{f}_1^{\text{FTM}}\}$ as the final model (e.g., using validation data).

**Choice of $\epsilon$ and $\delta$** For $\epsilon$, we set it as small as possible so that the level of fairness predominantly depends on $\delta$. However, using too small $\epsilon$ would make optimization unstable. Therefore, we set $\epsilon$ to the smallest value with which $\widehat{\mathbf{T}}_s, s \in \{0, 1\}$ are learnable (see Section D.3 of Appendix). Then, we control $\delta$ to make the trained prediction model achieve a given level of group fairness.

**Fairness consistency of FTM** We also prove that $\widehat{f}_s^{\text{FTM}}$ asymptotically achieves a given level of group fairness by controlling $\delta$ accordingly. We say that an estimator $\widehat{f}$ is *fairness-consistent* with respect to a fairness measure $\Delta$ of level $\delta > 0$ if there exists a positive sequence $a_n$ converging to 0 such that $\Delta(\widehat{f}) \leq \delta + a_n$ with probability converging to 1 as $n \to \infty$. The fairness consistency of $\widehat{f}_s^{\text{FTM}}$ under regularity conditions is given in Theorem B.5 in Section B.5 of Appendix.

### 3.4 CLOSELY RELATED APPROACHES

**Fair Representation Learning (FRL)** Fair Representation Learning (FRL) algorithm aims at searching a fair representation space (Zemel et al., 2013) in the sense that the distributions of the encoded representation vectors of each sensitive group are similar. Initiated by Edwards & Storkey (2016), various FRL algorithms have been developed (Madras et al., 2018; Zhang et al., 2018).

Remarkably, the matching function $\mathbf{T}_s$ in FTM can be interpreted as a fair representation encoder, where the representation space is $\mathcal{X}_{s'}$. That is, FTM is a variant of FRL that uses barycentric mapping as the representation encoder. On the other hand, there is a difference in how they achieve a given level of fairness: FTM sets (or tunes) $\delta$ to control fairness under $\epsilon \approx 0$, whereas FRL sets $\epsilon$ to control fairness under $\delta = 0$. In fact, Gordaliza et al. (2019) proposed to use the barycentric mapping for FRL. FTM has several advantages compared to Gordaliza et al. (2019). Section 4.3 and Section C.5 of Appendix provide empirical and conceptual comparisons, respectively, between FTM and FRL.

---

[2] When $\gamma$ is large, we can still use $\mathcal{F}^{\Delta\text{MDP}}(\delta_2; \{\mathbf{T}_{0,\epsilon}^\star\}, \{\mathbf{T}_{1,\epsilon}^\star\})$ to exclude undesirable group-fair models. See Section C.4 of Appendix.

**Counterfactual fairness**   FTM is also related to counterfactual fairness (Kusner et al., 2017), where a given input and its counterfactual are treated similarly. Instead of using graphical models to define counterfactuals, the MDP constraint uses a matching function. In fact, under a simple Structural Equation Model (SEM), the OT map results in the counterfactual. Particularly, write $\mathbf{X}_s = \mathbf{X}|S = s$ for $s \in \{0, 1\}$ and consider an SEM $\mathbf{X}_s = \mu_s + A\mathbf{X}_s + \epsilon_s$ for given $A \in \mathbb{R}^{d \times d}$, $\mu_s \in \mathbb{R}^d$ with random Gaussian noise $\epsilon_s \sim \mathcal{N}(0, \sigma_s^2 D)$ of a diagonal matrix $D \in \mathbb{R}^{d \times d}$ and variance scaler $\sigma_s^2$. An example DAG (Directed Acyclic Graph) is Figure 2.

Let $\mathbf{x}_s$ be a realization of $\mathbf{X}_s$ and assume $(I - A)$ has its inverse $B := (I - A)^{-1}$ where $I \in \mathbb{R}^{d \times d}$ is the identity matrix. Then, its counterfactual becomes $\tilde{\mathbf{x}}_s^{\mathrm{CF}} = B\mu_{s'} + \sigma_{s'}\sigma_s^{-1}(\mathbf{x}_s - B\mu_s)$ (proved with Proposition 3.4). On the other hand, by Lemma A.7 in Section A of Appendix, the image of $\mathbf{x}_s$ by the OT map is given as $\tilde{\mathbf{x}}_s^{\mathrm{OT}} = B\mu_{s'} + \mathbf{W}_s(\mathbf{x}_s - B\mu_s)$ for some $\mathbf{W}_s \in \mathbb{R}^{d \times d}$. Proposition 3.4 shows $\tilde{\mathbf{x}}_s^{\mathrm{CF}} = \tilde{\mathbf{x}}_s^{\mathrm{OT}}$, whose proof is referred to Section B.4 of Appendix.

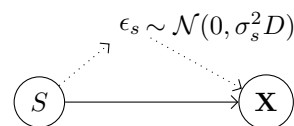

Figure 2: An example DAG of the SEM $\mathbf{X}_s = \mu_s + A\mathbf{X}_s + \epsilon_s$.

**Proposition 3.4** (Counterfactual fairness and the OT map). *For all $A$ having $(I - A)^{-1}$, $\tilde{\mathbf{x}}_s^{\mathrm{CF}} = \tilde{\mathbf{x}}_s^{\mathrm{OT}}$.*

**Fairness assessment**   In FlipTest (Black et al., 2020), the OT map is used to measure the group fairness of a given prediction model. However, in Black et al. (2020), there is no discussion about how to learn a group-fair model. Moreover, the measure for FlipTest is fundamentally different from ours. See Section C.8 of Appendix for details.

## 4   EXPERIMENTS

This section presents the experimental results supporting the superiority of FTM to existing group fairness algorithms. In particular, we empirically show that FTM achieves better subset/within-group fairness without degrading accuracy significantly. Additionally, we illustrate the advantages of FTM over FRL in terms of fairness-accuracy trade-off and flexibility in model selection. Finally, we provide an ablation study to compare the relaxed OT map with other possible matching functions.

### 4.1   SETTINGS

**Datasets**   We use five real benchmark tabular datasets, *Adult* (Dua & Graff, 2017), *German* (Dua & Graff, 2017), *COMPAS*[3], *Dutch* (Van der Laan, 2001), and *Law* (Ramsey et al., 1998), whose basic information is provided in Table 4 in Section D of Appendix. We disclaim that *COMPAS* dataset is used solely for experimental purposes, despite its several known limitations regarding policing practices (Fabris et al., 2022). We partition the datasets randomly into 8:2 for training and test datasets except for *Adult*, which has a separate test dataset. We repeat this procedure 5 times over 5 random initial parameters and take the average of these test results.

**Implementation details and baseline algorithms**   For FTM, we use 2-layer MLP networks as $\mathcal{T}_s$, a single-layer neural network as $\mathcal{F}$, and MMD as $d_{\mathcal{H}}$ for stable optimization. We train $\hat{\mathbf{T}}_s$ with $\epsilon$ as small as possible to make $\hat{\mathbf{T}}_s$ be close to the true OT map. Then, we train $\hat{f}_s^{\mathrm{FTM}}, s \in \{0, 1\}$ for many given $\delta$s. For the FRL algorithms, we consider fair AutoEncoder (AE) regularized by MMD (AE-MMD, Deka & Sutherland (2023)), and two adversarial learning-based approaches from Madras et al. (2018) (LAFTR) and Kim et al. (2022) (sIPM-LFR). To ensure fair comparisons, we use 2-layer MLP networks for the encoder and single-layer networks for the prediction head. For the in-processing algorithms, we consider Reg (minimizing cross-entropy + $\lambda\Delta\overline{\mathrm{DP}}$ (Chuang & Mroueh, 2021; Donini et al., 2018), Adv (Zhang et al., 2018), Fair-Mixup (Chuang & Mroueh, 2021), and Reduction (Agarwal et al., 2018). The unfair baseline (abbr. Unfair) is the classifier trained without fairness regularization. For the in-processing and unfair algorithms, we use single-layer neural networks as the model. For fairness measures, we consider $\Delta\overline{\mathrm{DP}}$, $\Delta\mathrm{DP}$, and $\Delta\mathrm{SDP}$. More implementation details are explained in Section D of Appendix.

---

[3]https://www.propublica.org/article/machine-bias-risk-assessments-in-criminal-sentencing

## 4.2 MAIN RESULTS: COMPARISON WITH GROUP-FAIR ALGORITHMS

The key to the success of FTM is lies in the use of the relaxed OT map as the matching function. It can improve subset and within-group fairness without degrading accuracy significantly, which we empirically investigate in this section. We compare FTM with the seven state-of-the-art baseline algorithms described in Section 4.1.

**Subset fairness** To evaluate subset fairness, we generate a random subset $\mathcal{D}_{\text{sub}}$ of the test data by $\mathcal{D}_{\text{sub}} = \{i : \mathbf{v}^\top \mathbf{x}_i \geq 0\}$ for a random vector $\mathbf{v}$ generated from the uniform distribution on $[-1, 1]^d$. Then, we calculate $\Delta\overline{\text{DP}}$ on $\mathcal{D}_{\text{sub}}$. Figure 3 draws the boxplots of the $\Delta\overline{\text{DP}}$ values on *COMPAS* dataset for 1,000 randomly generated $\mathcal{D}_{\text{sub}}$ for FTM and baselines. Outliers in the boxplots (points in red box) are the cases of subset unfairness. Note that FTM has the smallest number of outliers consistently, indicating that FTM achieves higher levels of subset fairness.

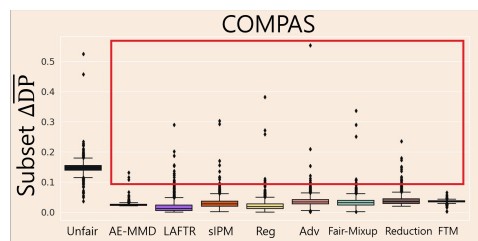

Figure 3: Subset fairness: boxplots of the levels of $\Delta\overline{\text{DP}}$ on 1,000 randomly generated subsets $\mathcal{D}_{\text{sub}}$ for *COMPAS* test dataset. Boxplots for the other datasets are presented in Figure 5 in Section E of Appendix.

**Within-group fairness** A reasonable measure for within-group fairness would be Spearman's rank correlation between unfair and fair prediction scores of each sensitive group: a higher correlation implies better within-group fairness. Table 1 presents the two rank correlations of the two sensitive groups, showing that FTM achieves better within-group fairness with large margins in most cases. Table 7 in Section E of Appendix presents an additional experimental result, indicating that FTM classifier has relatively fewer cases of flipping its prediction as $\widehat{Y} = 0$ for samples that the unfair classifier predicts as $\widehat{Y} = 1$, when compared to other group-fair classifiers.

Table 1: Within-group fairness: Spearman's correlation coefficient between the scores of the unfair model and group-fair models under a fixed level of $\Delta\overline{\text{DP}}$. The bold faces are the best, and underlined ones are the second placers. Table 6 in Section E of Appendix shows standard errors.

| Dataset | | Adult | | German | | COMPAS | | Dutch | | Law | |
|---|---|---|---|---|---|---|---|---|---|---|---|
| $\Delta\overline{\text{DP}}$ : Unfair → Fair | | $0.19 \to 0.10$ | | $0.09 \to 0.04$ | | $0.19 \to 0.10$ | | $0.34 \to 0.14$ | | $0.17 \to 0.07$ | |
| Sensitive attribute $S$ | | 0 | 1 | 0 | 1 | 0 | 1 | 0 | 1 | 0 | 1 |
| AE-MMD | | 0.771 | 0.872 | 0.651 | 0.779 | 0.436 | 0.463 | 0.825 | 0.929 | 0.790 | 0.585 |
| LAFTR | | 0.710 | 0.876 | 0.677 | 0.772 | 0.457 | 0.468 | 0.835 | 0.912 | 0.820 | 0.703 |
| sIPM-LFR | | 0.745 | 0.880 | 0.698 | 0.809 | 0.402 | 0.587 | 0.794 | 0.920 | 0.674 | 0.710 |
| Reg | | 0.907 | 0.885 | **0.852** | 0.863 | 0.852 | 0.792 | 0.950 | 0.916 | 0.775 | 0.553 |
| Adv | | 0.885 | 0.845 | 0.830 | 0.804 | 0.795 | 0.742 | 0.944 | 0.927 | 0.807 | 0.599 |
| Fair-Mixup | | 0.894 | 0.905 | 0.829 | 0.758 | 0.904 | 0.812 | **0.953** | 0.907 | 0.791 | 0.653 |
| Reduction | | 0.905 | 0.890 | 0.840 | 0.851 | 0.848 | 0.800 | 0.950 | 0.916 | 0.867 | 0.583 |
| FTM | | **0.921** | **0.945** | 0.836 | **0.906** | **0.907** | **0.864** | 0.931 | **0.975** | **0.915** | **0.738** |

**Accuracy** We also observe that FTM does not excessively degrade prediction accuracy (see Table 2: the averaged relative drop of accuracy compared to the averaged accuracy of the seven baselines is $0.8\%$). This represents a trade-off; this slight degradation in accuracy is made to improve subset/within-group fairness. For more detailed results, see Table 8 in Section E of Appendix.

Table 2: The averaged relative change of Acc under a fixed $\Delta\overline{\text{DP}}$.

| Dataset ($\Delta\overline{\text{DP}}$) | | *Adult* (0.06) | *German* (0.05) | *COMPAS* (0.12) | *Dutch* (0.03) | *Law* (0.04) | | Average |
|---|---|---|---|---|---|---|---|---|
| Relative change of accuracy | | -1.3% | -0.4% | -1.1% | -1.2% | +0.2% | | -0.8% |

### 4.3 ANOTHER ADVANTAGE OF FTM: COMPARISON WITH FRL

As discussed earlier in Section 3.4, FTM and unsupervised FRL are closely related in the sense that the distribution matching approach is applied to achieve group fairness. Therefore, we specifically compare with FRL in two aspects: (1) fairness-accuracy trade-off and (2) flexibility in model selection.

**Fairness-accuracy trade-off**  Figure 4 depicts Pareto-front lines as is done by Madras et al. (2018); Kim et al. (2022); Chuang & Mroueh (2021) to present the fairness-accuracy trade-off with respect to $\Delta\overline{\mathrm{DP}}$. The larger-sized plots with standard error intervals for the three fairness measures $\Delta\overline{\mathrm{DP}}$, $\Delta\mathrm{DP}$ and $\Delta\mathrm{SDP}$ are provided in Figure 6 in Section E of Appendix. It is obvious that FTM algorithm achieves superior or at least competitive performances compared to the FRL algorithms.

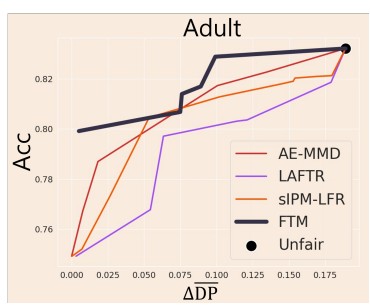

Figure 4: FTM vs FRL: Pareto-front lines for fairness-accuracy trade-off for *Adult* dataset. Full results on the five datasets with respect to the three fairness measures $\Delta\overline{\mathrm{DP}}$, $\Delta\mathrm{DP}$ and $\Delta\mathrm{SDP}$ are provided in Figure 6 in Section E of Appendix.

**Flexibility in model selection**  Another important advantage of FTM over FRL is that the selection of the classifier model $\mathcal{F}$ and the transport network $\mathcal{T}_s$ can be made separately. In contrast, the full model network of FRL is the composition of the encoder on the input space and the prediction head on the representation space. For example of interpretability, when it is required to use simple models such as linear or generalized additive models, FRL's full network becomes overly simple. We compare FTM and FRL under this scenario: $\mathcal{F}$ is the class of linear models. The overall fairness-accuracy trade-offs are provided in Figure 7 in Section E of Appendix, which shows the outperformance of FTM.

### 4.4 ABLATION STUDY: COMPARISON WITH OTHER MATCHING FUNCTIONS

For ablation, we assess and compare three variations of FTM: FTM-coupling, FRL-bary, and FRL-match. This study aims to compare many matching functions in terms of the fairness-accuracy trade-off. (1) FTM-coupling is a variant of FTM where the matching function is defined by the optimal coupling (Kantorovich, 2006), (2) FRL-bary is an FRL where the encoder is the optimal transport map to the barycenter, which is similar to Gordaliza et al. (2019). (3) FRL-match is a variant of FTM whose matching is done using the encoder and decoder learned by Kim et al. (2022). Figure 8 of Section E of Appendix shows that the relaxed OT map performs consistently better than the three other alternatives. Refer to Section E of Appendix for more details.

## 5 CONCLUDING REMARKS

We have demonstrated that MDP can be considered as a measure of strong demographic parity. The proposed algorithm built on this finding, FTM, offers several advantages over its competitors, in terms of subset and within-group fairness, without significantly sacrificing accuracy. Using the relaxed OT map as the matching function may be inefficient when a large portion of the input comprises noise. In such cases, it is necessary to remove noise before learning the relaxed OT map.

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

# A   THEORETICAL TOOLS

## A.1   MATHEMATICAL DEFINITIONS

**Definition A.1** (Function norms). Let $\mathcal{X}$ be a compact subset of $\mathbb{R}^d$. Denote $|| \cdot ||_p$ be the $l_p$ norm defined as $||g||_p = ||g||_{p,\mathcal{X}} := \left( \int_{\mathcal{X}} |g(\mathbf{x})|^p \, d\mu(\mathbf{x}) \right)^{1/p}$ for a given function $g : \mathcal{X} \to \mathbb{R}$, where $\mu$ is the standard Lebesgue measure on $\mathbb{R}^d$. We write $||g||_\infty = ||g||_{\infty,\mathcal{X}} := \sup_{\mathbf{x} \in \mathcal{X}} |g(\mathbf{x})|$. For vector-valued function $\mathbf{T}(\cdot) = [T_1(\cdot), \cdots, T_d(\cdot)]^\top : \mathcal{X} \to \mathbb{R}^d$, we write $\|\mathbf{T}\|_\infty := \sup_{\mathbf{x} \in \mathcal{X}} \frac{1}{d} \sum_{j=1}^d |T_j(\mathbf{x})|$.

**Definition A.2** (Function derivatives). For a given $t > 0$, let $[t]$ be the largest integer less than or equal to $t$ and $\lceil t \rceil$ as the smallest integer greater or equal to $t$. For given $\mathbf{s} = [s_1, \cdots, s_d]^\top \in \mathbb{N}_0^d$, where $\mathbb{N}_0$ is the set of non-negative integers, we define the derivative of a function $g$ of order $\mathbf{q}$ as

$$\partial^{\mathbf{q}} g = \frac{\partial^{|\mathbf{q}|} g}{\partial x_1^{q_1} \cdots \partial x_d^{q_d}},$$

where $|\mathbf{q}| = q_1 + \cdots + q_d$. Further, for $r \in (0, 1]$, we denote

$$[g]_{r,\mathcal{X}} = \sup_{\mathbf{x},\mathbf{x}' \in \mathcal{X}, \mathbf{x} \neq \mathbf{x}'} \frac{|g(\mathbf{x}) - g(\mathbf{x}')|}{|\mathbf{x} - \mathbf{x}'|^r}.$$

**Definition A.3** (Smooth functions). For an integer $m$, we denote $C^m(\mathcal{X})$ the space of functions on $\mathcal{X}$ whose partial derivatives of order $\mathbf{q}$ with $|\mathbf{q}| \leq m$ exist are continuous. That is,

$$C^m(\mathcal{X}) = \{g : \mathcal{X} \to \mathbb{R}, \partial^{\mathbf{q}} g \text{ are continuous for } \forall \mathbf{q} \text{ such that } |\mathbf{q}| \leq m\}.$$

**Definition A.4** (Hölder function class). The real-valued Hölder function class with smoothness $\gamma > 0$ (i.e., $\gamma$-Hölder smooth function class) is a function space defined as

$$\mathcal{G}_\gamma = \mathcal{G}_\gamma(\mathcal{X}) := \{g \in C^{[\gamma]}(\mathcal{X}) : ||g||_{\mathcal{G}_\gamma(\mathcal{X})} < \infty\}$$

where

$$||g||_{\mathcal{G}_\gamma(\mathcal{X})} = \max_{|\mathbf{m}| \leq [\gamma]} ||\partial^{\mathbf{m}} g||_{\infty,\mathcal{X}} + \max_{|\mathbf{m}| = [\gamma]} [\partial^{\mathbf{m}} g]_{\gamma,\mathcal{X}}.$$

The vector-valued ($d$-dimensional) Hölder function class with smoothness $\beta > 0$ is a function space defined as

$$\mathcal{T}_\beta = \mathcal{T}_\beta(\mathcal{X}) = \{\mathbf{T} : \mathbf{x} \mapsto [T_1(\mathbf{x}), \cdots, T_d(\mathbf{x})]^\top, \mathbf{x} \in \mathcal{X}, T_j \in \mathcal{G}_\beta, j \in [d]\}.$$

**Definition A.5** (Rademacher complexity). Let $\sigma$ be a binary random variable whose distribution is $Uniform(\{-1, 1\})$. For $n$ many independent realizations $\sigma_1, \ldots, \sigma_n$ of $\sigma$, we define the empirical Rademacher complexity of $\mathcal{G}$ as

$$\mathcal{R}_n(\mathcal{G}) := \frac{1}{n} \mathbb{E}_\sigma \left( \sup_{g \in \mathcal{G}} \sum_{i=1}^n \sigma_i g(\mathbf{X}_i) \right).$$

The (population) Rademacher complexity $\mathcal{R}(\mathcal{G})$ is the expectation of the empirical Rademacher complexity over $\mathbf{X} = [\mathbf{X}_1, \cdots, \mathbf{X}_n]$, i.e.,

$$\mathcal{R}(\mathcal{G}) := \mathbb{E}_{\mathbf{X}} \left( \mathcal{R}_n(\mathcal{G}) \right).$$

## A.2   TECHNICAL LEMMAS

**Lemma A.6** (Generalization bound: Theorem 4.10 of Wainwright (2019) or Theorem 26.5 of Shalev-Shwartz & Ben-David (2014)). *Let $P$ be the distribution of a random vector $\mathbf{X}$. For $n$ many i.i.d. samples $\mathbf{X}_1, \cdots, \mathbf{X}_n \sim P$, we write $P_n$ as the empirical distribution of $\mathbf{X}_1, \cdots, \mathbf{X}_n$. Let $\mathcal{G}$ be a set of real-valued functions such that $\sup_{g \in \mathcal{G}} \|g\|_\infty \leq B_{\mathcal{G}}$ for a $B_{\mathcal{G}} > 0$. Then,*

$$\sup_{g \in \mathcal{G}} \left| \int g(\mathbf{x}) d(P - P_n)(\mathbf{x}) \right| \leq 2\mathcal{R}_n(\mathcal{G}) + B_{\mathcal{G}} \sqrt{\frac{2\log(1/\delta)}{n}}$$

*with probability at least $1 - \delta > 0$.*

**Lemma A.7** (Optimal transport map between two Gaussians). *For mean vectors $\mu_{\mathbf{X}}, \mu_{\mathbf{Y}} \in \mathbb{R}^d$ and covariance matrices $\Sigma_{\mathbf{X}}, \Sigma_{\mathbf{Y}} \in \mathbb{R}^{d \times d}$, the OT map from $\mathcal{N}(\mu_{\mathbf{X}}, \Sigma_{\mathbf{X}})$ to $\mathcal{N}(\mu_{\mathbf{Y}}, \Sigma_{\mathbf{Y}})$ is given as*

$$\mathbf{T}^{\text{OT}}(\mathbf{x}) = \mathbf{W}^{\text{OT}}\mathbf{x} + \mathbf{b}^{\text{OT}} \text{ where } \mathbf{W}^{\text{OT}} = \Sigma_{\mathbf{X}}^{-\frac{1}{2}} \left( \Sigma_{\mathbf{X}}^{\frac{1}{2}} \Sigma_{\mathbf{Y}} \Sigma_{\mathbf{X}}^{\frac{1}{2}} \right)^{\frac{1}{2}} \Sigma_{\mathbf{X}}^{-\frac{1}{2}} \text{ and } \mathbf{b}^{\text{OT}} = \mu_{\mathbf{Y}} - \mathbf{W}^{\text{OT}}\mu_{\mathbf{X}}.$$

*Proof.* Consider the centered Gaussians, i.e., $\mu_{\mathbf{X}} = \mu_{\mathbf{Y}}$ at first. Based on Theorem 4 of Olkin & Pukelsheim (1982), we have that $\mathcal{W}_2 \left( \mathcal{N}(0, \Sigma_{\mathbf{X}}), \mathcal{N}(0, \Sigma_{\mathbf{Y}}) \right) = Tr(\Sigma_{\mathbf{X}} + \Sigma_{\mathbf{Y}} - 2 \left( \Sigma_{\mathbf{X}}^{1/2} \Sigma_{\mathbf{Y}} \Sigma_{\mathbf{X}}^{1/2} \right)^{1/2}) = \|\Sigma_{\mathbf{X}}^{1/2} - \Sigma_{\mathbf{Y}}^{1/2}\|_F^2$ where $\| \cdot \|$ is the Frobenius norm. Correspondingly, Knott & Smith (1984) derived the optimal transport map as $\mathbf{x} \mapsto \Sigma_{\mathbf{X}}^{-1/2} \left( \Sigma_{\mathbf{X}}^{-1/2} \Sigma_{\mathbf{Y}} \Sigma_{\mathbf{X}}^{1/2} \right)^{1/2} \Sigma_{\mathbf{X}}^{-1/2}\mathbf{x}$.

Combining these results, we can extend the OT map formula of Gaussians with nonzero means as follows. Since $\mathbb{E}\|\mathbf{X} - \mathbf{Y}\|^2 = \mathbb{E}\|(\mathbf{X} - \mu_{\mathbf{X}}) - (\mathbf{Y} - \mu_{\mathbf{Y}}) + (\mu_{\mathbf{X}} - \mu_{\mathbf{Y}})\|^2 = \mathbb{E}\|(\mathbf{X} - \mu_{\mathbf{X}}) - (\mathbf{Y} - \mu_{\mathbf{Y}})\|^2 + \|\mu_{\mathbf{X}} - \mu_{\mathbf{Y}}\|^2$, the Wasserstein distance is given as $\mathbf{W}_2 \left( \mathcal{N}(\mu_{\mathbf{X}}, \Sigma_{\mathbf{X}}), \mathcal{N}(\mu_{\mathbf{Y}}, \Sigma_{\mathbf{Y}}) \right) = \|\mu_{\mathbf{X}} - \mu_{\mathbf{Y}}\|^2 + \|\Sigma_{\mathbf{X}}^{1/2} - \Sigma_{\mathbf{Y}}^{1/2}\|_F^2$, and so the corresponding optimal transport map is also given as $\mathbf{x} \mapsto \Sigma_{\mathbf{X}}^{-1/2} \left( \Sigma_{\mathbf{X}}^{-1/2} \Sigma_{\mathbf{Y}} \Sigma_{\mathbf{X}}^{1/2} \right)^{1/2} \Sigma_{\mathbf{X}}^{-1/2}\mathbf{x} + \mu_{\mathbf{Y}} - \Sigma_{\mathbf{X}}^{-1/2} \left( \Sigma_{\mathbf{X}}^{-1/2} \Sigma_{\mathbf{Y}} \Sigma_{\mathbf{X}}^{1/2} \right)^{1/2} \Sigma_{\mathbf{X}}^{-1/2}\mu_{\mathbf{X}}$, which completes the proof. $\square$

**Lemma A.8** (Approximation of the class of Hölder smooth functions by DNNs). *For a given $\beta$, there exists a DNN function class $\mathcal{T}_n^{\text{DNN}}$ such that $\exists \mathbf{T} \in \mathcal{T}_n^{\text{DNN}}$ satisfying $\|\mathbf{T} - \mathbf{T}_\beta\|_\infty \leq \mathcal{O}(n^{-\beta/(2\beta+d)})$ for any $\mathbf{T}_\beta \in \mathcal{T}_\beta$ and $\mathcal{R}_n(\mathcal{T}_n^{\text{DNN}}) \leq \mathcal{O}(n^{-\beta/(2\beta+d)})$ up to a logarithm factor. For a given $\gamma$, there exists a DNN function class $\mathcal{H}_n^{\text{DNN}}$ such that $\exists h \in \mathcal{H}_n^{\text{DNN}}$ satisfying $\|h - h_\gamma\|_\infty \leq \mathcal{O}(n^{-\gamma/(2\gamma+d)})$ for any $h_\gamma \in \mathcal{H}_\gamma$ and $\mathcal{R}_n(\mathcal{H}_n^{\text{DNN}}) \leq \mathcal{O}(n^{-\gamma/(2\gamma+d)})$ up to a logarithm factor.*

The proof of Lemma A.8 can be found in Theorem 5 and Lemma 5 of Schmidt-Hieber (2020).

# B PROOFS OF MAIN THEOREMS

## B.1 PROOF OF PROPOSITION 2.1

Let $B > 0$ be the bound of sup-norm of $f$, i.e., $\sup_{\mathbf{x}} |f(\mathbf{x}, s)| \in [-B, B]$.

**Definition B.1.** For $s = 0, 1$ and any measurable set $A \subseteq [-B, B]$, we denote

$$f_s^{-1}(A) := \{\mathbf{x} \in \mathcal{X}_s : f(\mathbf{x}, s) \in A\}.$$

**Lemma B.2.** *For any given $f \in \mathcal{F}^{\Delta\text{TVDP}}(\delta)$, there exists $\mathbf{T}_s$ satisfying $d_{\text{TV}}(\mathbf{T}_{s\#}\mathcal{P}_s, \mathcal{P}_{s'}) = 0$ and $\mathbb{E}_s |f(\mathbf{X}, s) - f(\mathbf{T}_s(\mathbf{X}), s')| \leq C\delta$ for some constant $C > 0$.*

*Proof.* Without loss of generality, let $s = 0$ and $s' = 1$.

Denote $F_0 : [-B, B] \rightarrow [0, 1]$ and $F_1 : [-B, B] \rightarrow [0, 1]$ the CDFs (Cumulative Distribution Function) of $f(\mathbf{X}, 0)|S = 0$ and $f(\mathbf{X}, 1)|S = 1$, respectively. Note that $F_0$ and $F_1$ have at most countably many discontinuous points. We define the set of all discontinuous points of $F_s$ as $D_s$, which is countable.

1. We define the sub-CDF $F_s^{cont}(v) := F_s(v) - \sum_{t \in D_s, t \leq v} \Delta F_s(t)$ for $v \in [-B, B]$.

   We prove the lemma for the case of $F_1^{cont}(B) \leq F_0^{cont}(B)$. The case of $F_1^{cont}(B) > F_0^{cont}(B)$ can be treated similarly.

   There exists $z \leq B$ such that $F_1^{cont}(B) = F_0^{cont}(z)$. Define $v_k = -B + \lfloor\delta\rfloor k$ for $k \in \{0, \ldots, m-1\}$ where $m \in \mathbb{N}$ satisfies $-B + (m-1)\lfloor\delta\rfloor \leq z \leq -B + m\lfloor\delta\rfloor$. We also let $v_m = z$.

   Fix $k \in \{1, \ldots, m\}$. Suppose that $F_1^{cont}((v_{k-1}, v_k]) \leq F_0^{cont}((v_{k-1}, v_k])$. Then, there exists $z_k \leq v_k$ such that $F_1^{cont}((v_{k-1}, v_k]) = F_0^{cont}((v_{k-1}, z_k])$. Define $\mathcal{X}_{0,k} := f_0^{-1}((v_{k-1}, z_k] \setminus D_0)$ and $\mathcal{X}_{1,k} := f_1^{-1}((v_{k-1}, v_k] \setminus D_1)$. We can define $\mathcal{X}_{0,k}$ and $\mathcal{X}_{1,k}$ similarly when $F_1^{cont}((v_{k-1}, v_k]) \geq F_0^{cont}((v_{k-1}, v_k])$.

   For each $k \in \{1, \ldots, m\}$, we define probability measures $\mathcal{P}_{s,k}, s \in \{0, 1\}$ such that $\mathcal{P}_{s,k}(A) := \mathcal{P}_s(A \cap \mathcal{X}_{s,k})/\mathcal{P}_s(\mathcal{X}_{s,k})$ for measurable subsets $A \subseteq \mathcal{X}$.

   Then, there exists a matching function $\mathbf{T}_{0,k}^{(1)}$ from $\mathcal{P}_{0,k}(\cdot)$ to $\mathcal{P}_{1,k}(\cdot)$, by Breiner's Theorem (Villani, 2008; Hütter & Rigollet, 2021) under (C2). Since $v_k - v_{k-1} \leq \delta, \forall k$, we have that $|f(\mathbf{x}, 0) - f(\mathbf{T}_{0,k}^{(1)}(\mathbf{x}), 1)| \leq \delta$ for $\mathbf{x} \in \mathcal{X}_{0,k}$.

2. Second, we consider the intersection of $D_0$ and $D_1$. Let $D_{0,1} := D_0 \cap D_1$.

   Fix $d \in D_{0,1}$. Suppose that $\mathcal{P}_1(f_1^{-1}(\{d\})) \leq \mathcal{P}_0(f_0^{-1}(\{d\}))$. Then, there exists $f_0^{-1}(\{d\})' \subset f_0^{-1}(\{d\})$ such that $\mathcal{P}_0(f_0^{-1}(\{d\}))' = \mathcal{P}_1(f_1^{-1}(\{d\}))$. Define $\tilde{\mathcal{X}}_{0,d} := f_0^{-1}(\{d\})'$ and $\tilde{\mathcal{X}}_{1,d} := f_1^{-1}(\{d\})$. We can define $\tilde{\mathcal{X}}_{0,d}$ and $\tilde{\mathcal{X}}_{1,d}$ similarly when $\mathcal{P}_1(f_1^{-1}(\{d\})) > \mathcal{P}_0(f_0^{-1}(\{d\}))$.

   For each $d \in D_{0,1}$, we define probability measures $\tilde{\mathcal{P}}_{s,d}, s \in \{0, 1\}$ such that $\tilde{\mathcal{P}}_{s,d}(A) := \mathcal{P}_s(A \cap \tilde{\mathcal{X}}_{s,d})/\mathcal{P}_s(\tilde{\mathcal{X}}_{s,d})$ for measurable subsets $A \subseteq \mathcal{X}$.

   Then, there exists a matching function $\mathbf{T}_{0,d}^{(2)}$ from $\tilde{\mathcal{P}}_{0,d}(\cdot)$ to $\tilde{\mathcal{P}}_{1,d}(\cdot)$. By definition of $D_{0,1}$, we note that $f(\mathbf{x}, 0) = f(\mathbf{T}_{0,d}^{(2)}(\mathbf{x}), 1)$ for $\mathbf{x} \in \tilde{\mathcal{X}}_{0,d}$.

3. Third, we collect the complement sets as

$$\mathcal{X}_0' := \mathcal{X}_0 \setminus \left( \bigcup_{k \in \{1, \ldots, m\}} \mathcal{X}_{0,k} \cup \bigcup_{d \in D_{0,1}} \tilde{\mathcal{X}}_{0,d} \right)$$

   and

$$\mathcal{X}_1' := \mathcal{X}_1 \setminus \left( \bigcup_{k \in \{1, \ldots, m\}} \mathcal{X}_{1,k} \cup \bigcup_{d \in D_{0,1}} \tilde{\mathcal{X}}_{1,d} \right).$$

Because $\mathcal{P}_0(\bigcup_{k \in \{1,\dots,m\}} \mathcal{X}_{0,k}) = \mathcal{P}_1(\bigcup_{k \in \{1,\dots,m\}} \mathcal{X}_{1,k})$ and $\mathcal{P}_0(\bigcup_{d \in D_{0,1}} \tilde{\mathcal{X}}_{0,d}) = \mathcal{P}_1(\bigcup_{d \in D_{0,1}} \tilde{\mathcal{X}}_{1,d})$, we have $\mathcal{P}_0(\mathcal{X}'_0) = 1 - \mathcal{P}_0(\bigcup_{k \in \{1,\dots,m\}} \mathcal{X}_{0,k}) - \mathcal{P}_0(\bigcup_{d \in D_{0,1}} \tilde{\mathcal{X}}_{0,d}) = \mathcal{P}_1(\mathcal{X}'_1)$.

We define probability measures $\mathcal{P}'_s, s \in \{0, 1\}$ such that $\mathcal{P}'_s(A) := \mathcal{P}_s(A \cap \mathcal{X}'_s)/\mathcal{P}_s(\mathcal{X}'_s)$ for measurable subsets $A \subseteq \mathcal{X}$.

Then, there exists a matching function $\mathbf{T}_0^{(3)}$ from $\mathcal{P}'_0(\cdot)$ to $\mathcal{P}'_1(\cdot)$.

Furthermore, because $d_{\text{TV}}\left(\mathcal{P}_{f(\mathbf{X},0)|S=0}, \mathcal{P}_{f(\mathbf{X},1)|S=1}\right) \leq \delta$, we have $\mathcal{P}_0(\mathcal{X}'_0) \leq \delta$, and by $f(\cdot) \in [-B, B]$, it holds that

$$
\begin{aligned}
\mathbb{E}_0 &\left( |f(\mathbf{X}, 0) - f(\mathbf{T}_0^{(3)}(\mathbf{X}), 1)| \cdot \mathbb{I}(\mathbf{X} \in \mathcal{X}'_0) \right) \\
&= \int |f(\mathbf{X}, 0) - f(\mathbf{T}_0^{(3)}(\mathbf{X}), 1)| \cdot \mathbb{I}(\mathbf{X} \in \mathcal{X}'_0) d\mathcal{P}_0(\mathbf{X}) \\
&\leq 2B \int \mathbb{I}(\mathbf{X} \in \mathcal{X}'_0) d\mathcal{P}_0(\mathbf{X}) = 2B\mathcal{P}_0(\mathcal{X}'_0) \leq 2B\delta.
\end{aligned}
\tag{3}
$$

4. Finally, combining 1 to 3, we define

$$
\mathbf{T}_0(\cdot) := \sum_{k=1}^{m} \mathbf{T}_{0,k}^{(1)}(\cdot) \mathbb{I}(\cdot \in \mathcal{X}_{0,k}) + \sum_{d \in D_{0,1}} \mathbf{T}_{0,d}^{(2)}(\cdot) \mathbb{I}(\cdot \in \tilde{\mathcal{X}}_{0,d}) + \mathbf{T}_0^{(3)}(\cdot) \mathbb{I}(\cdot \in \mathcal{X}'_0).
\tag{4}
$$

We note that $\left\{ \{\mathcal{X}_{0,k}\}_{k=1}^m, \{\tilde{\mathcal{X}}_{0,d}\}_{d \in D_{0,1}}, \mathcal{X}'_0 \right\}$ and $\left\{ \{\mathcal{X}_{1,k}\}_{k=1}^m, \{\tilde{\mathcal{X}}_{1,d}\}_{d \in D_{0,1}}, \mathcal{X}'_1 \right\}$ are partitions of $\mathcal{X}_0$ and $\mathcal{X}_1$, respectively. Moreover, $\mathcal{P}_0(\mathcal{X}_{0,k}) = \mathcal{P}_1(\mathcal{X}_{1,k}), \forall k$, $\mathcal{P}_0(\tilde{\mathcal{X}}_{0,d}) = \mathcal{P}_1(\tilde{\mathcal{X}}_{1,d}), \forall d$, and $\mathcal{P}_0(\mathcal{X}'_0) = \mathcal{P}_1(\mathcal{X}'_1)$. Hence, $\mathbf{T}_0$ is a matching function from $\mathcal{P}_0$ to $\mathcal{P}_1$.

Furthermore, we have that

$$
\begin{aligned}
\mathbb{E}_0 |f(\mathbf{X}, 0) - f(\mathbf{T}_0(\mathbf{X}), 1)| &= \int |f(\mathbf{X}, 0) - f(\mathbf{T}_0(\mathbf{X}), 1)| d\mathcal{P}_0(\mathbf{X}) \\
&= \sum_{k \in \{1,\dots,m\}} \int |f(\mathbf{X}, 0) - f(\mathbf{T}_{0,k}^{(1)}(\mathbf{X}), 1)| \cdot \mathbb{I}(\mathbf{X} \in \mathcal{X}_{0,k}) d\mathcal{P}_0(\mathbf{X}) \\
&\quad + \sum_{d \in D_{0,1}} \int |f(\mathbf{X}, 0) - f(\mathbf{T}_{0,d}^{(2)}(\mathbf{X}), 1)| \cdot \mathbb{I}(\mathbf{X} \in \tilde{\mathcal{X}}_{0,d}) d\mathcal{P}_0(\mathbf{X}) \\
&\quad + \int |f(\mathbf{X}, 0) - f(\mathbf{T}_0^{(3)}(\mathbf{X}), 1)| \cdot \mathbb{I}(\cdot \in \mathcal{X}'_0) d\mathcal{P}_0(\mathbf{X}) \\
&\leq \delta \sum_{k \in \{1,\dots,m\}} \mathcal{P}_0(\mathcal{X}_{0,k}) + \int |f(\mathbf{X}, 0) - f(\mathbf{T}_0^{(3)}(\mathbf{X}), 1)| \cdot \mathbb{I}(\cdot \in \mathcal{X}'_0) d\mathcal{P}_0(\mathbf{X}) \\
&\leq \delta + 2B\delta.
\end{aligned}
\tag{5}
$$

Letting $C = 1 + 2B$ completes the proof.

$\square$

**Lemma B.3.** *If a given map* $\mathbf{T}_s \in \mathcal{T}_s$ *satisfies* $d_{\mathrm{TV}}(\mathbf{T}_{s\#}\mathcal{P}_s, \mathcal{P}_{s'}) = 0$, *then we have* $\Delta\mathrm{WDP}(f), \Delta\overline{\mathrm{DP}}(f) \leq \delta$ *for any* $f \in \{f \in \mathcal{F} : \mathbb{E}_s|f(\mathbf{X}, s) - f(\mathbf{T}_s(\mathbf{X}), s')| \leq \delta\}$.

*Proof of Lemma B.3.* Fix $f \in \{f \in \mathcal{F} : \mathbb{E}_s|f(\mathbf{X}, s) - f(\mathbf{T}_s(\mathbf{X}), s')| \leq \delta\}$.

$$
\begin{aligned}
\Delta\mathrm{WDP}(f) &= d_{\mathcal{L}_1}\left(\mathcal{P}_{f(\mathbf{X},0)|S=0}, \mathcal{P}_{f(\mathbf{X},1)|S=1}\right) \\
&= \sup_{u \in \mathcal{L}_1} |\mathbb{E}_s(u \circ f(\mathbf{X}, s)) - \mathbb{E}_{s'}(u \circ f(\mathbf{X}, s'))| \\
&\leq \sup_{u \in \mathcal{L}_1} |\mathbb{E}_s(u \circ f(\mathbf{X}, s)) - \mathbb{E}_s(u \circ f(\mathbf{T}_s(\mathbf{X}), s'))| \\
&\quad + \sup_{u \in \mathcal{L}_1} |\mathbb{E}_s(u \circ f(\mathbf{T}_s(\mathbf{X}), s')) - \mathbb{E}_{s'}(u \circ f(\mathbf{X}, s'))| \\
&\leq \sup_{u \in \mathcal{L}_1} \mathbb{E}_s |u \circ f(\mathbf{X}, s) - u \circ f(\mathbf{T}_s(\mathbf{X}), s')| \\
&\quad + \sup_{u \in \mathcal{L}_1} |\mathbb{E}_s(u \circ f(\mathbf{T}_s(\mathbf{X}), s')) - \mathbb{E}_{s'}(u \circ f(\mathbf{X}, s'))| \\
&\overset{u \in \mathcal{L}_1}{\leq} \mathbb{E}_s |f(\mathbf{X}, s) - f(\mathbf{T}_s(\mathbf{X}), s')| \\
&\quad + \sup_{u \in \mathcal{L}_1} |\mathbb{E}_s(u \circ f(\mathbf{T}_s(\mathbf{X}), s')) - \mathbb{E}_{s'}(u \circ f(\mathbf{X}, s'))| \\
&\leq \delta + \sup_{u \in \mathcal{L}_1} |\mathbb{E}_s(u \circ f(\mathbf{T}_s(\mathbf{X}), s')) - \mathbb{E}_{s'}(u \circ f(\mathbf{X}, s'))| \\
&\leq \delta + \sup_{f \in \mathcal{F}} \sup_{u \in \mathcal{L}_1} |\mathbb{E}_s(u \circ f(\mathbf{T}_s(\mathbf{X}), s')) - \mathbb{E}_{s'}(u \circ f(\mathbf{X}, s'))| \\
&\leq \delta + d_{\mathrm{TV}}(\mathbf{T}_{s\#}\mathcal{P}_s, \mathcal{P}_{s'}) \\
&\leq \delta.
\end{aligned}
\tag{6}
$$

For $\Delta\overline{\mathrm{DP}}(f)$, because the identity map is 1-Lipschitz, we have that $\Delta\overline{\mathrm{DP}}(f) \leq \Delta\mathrm{WDP}(f)$, which completes the proof. $\square$

**Proposition 2.1** Under (C1) and (C2), for any given fairness level $\delta \geq 0$, we have $\mathcal{F}^{\Delta\mathrm{TVDP}}(C\delta) \subset \mathcal{F}^{\Delta\mathrm{MDP}}(\delta) \subset \mathcal{F}^{\Delta\mathrm{WDP}}(\delta)$ for some constant $C > 0$.

*Proof of Proposition 2.1.* Lemma B.3 implies $\mathcal{F}^{\Delta\mathrm{MDP}}(\delta) \subset \mathcal{F}^{\Delta\mathrm{WDP}}(\delta)$, while Lemma B.2 implies $\mathcal{F}^{\Delta\mathrm{TVDP}}(C\delta) \subset \mathcal{F}^{\Delta\mathrm{MDP}}(\delta)$ for some constant $C > 0$. $\square$

### B.2 PROOF OF PROPOSITION 3.3

The following lemma is used to prove Proposition 3.3.

**Lemma B.4** (Distance between $\mathbf{T}^\star_{s,\epsilon}$ and $\mathbf{T}^\star_s$).

$$\mathbb{E}_s \|\mathbf{T}^\star_{s,\epsilon}(\mathbf{X}) - \mathbf{T}^\star_s(\mathbf{X})\|^2 \leq \left( \frac{2}{\lambda_{\phi^\star}} + 1 \right) \epsilon$$

*where $\lambda_{\phi^\star}$ is the supremum of $\lambda$ such that $\phi^\star(x) - \lambda|x|^2/2$ is convex on $\mathcal{X}$.*

*Proof of Lemma B.4.* By definition of Wasserstein distance,

$$
\begin{aligned}
\mathcal{W}_2(\mathbf{T}^\star_{s,\epsilon\#}\mathcal{P}_s, \mathcal{P}_{s'}) &= \min_{\mathbf{T}:\mathbf{T}_\# \mathbf{T}^\star_{s,\epsilon\#}P=Q} \mathbb{E}_s \|\mathbf{T}^\star_{s,\epsilon}(\mathbf{X}) - \mathbf{T} \circ \mathbf{T}^\star_{s,\epsilon}(\mathbf{X})\|^2 \\
&= \mathbb{E}_s \|\mathbf{T}^\star_{s,\epsilon}(\mathbf{X}) - \tilde{\mathbf{T}}_{s,\epsilon} \circ \mathbf{T}^\star_{s,\epsilon}(\mathbf{X})\|^2 \leq \epsilon.
\end{aligned}
\tag{7}
$$

Using equation (7), we have

$$
\begin{aligned}
\mathbb{E}_s \|\mathbf{T}^\star_{s,\epsilon}(\mathbf{X}) - \mathbf{T}^\star(\mathbf{X})\|^2 &\leq \mathbb{E}_s \|\mathbf{T}^\star_{s,\epsilon}(\mathbf{X}) - \tilde{\mathbf{T}}_{s,\epsilon} \circ \mathbf{T}^\star_{s,\epsilon}(\mathbf{X})\|^2 + \mathbb{E}_s \|\tilde{\mathbf{T}}_{s,\epsilon} \circ \mathbf{T}^\star_{s,\epsilon}(\mathbf{X}) - \mathbf{T}^\star_s(\mathbf{X})\|^2 \\
&\leq \epsilon + \mathbb{E}_s \|\mathbf{T}'_{s,\epsilon} \circ \mathbf{T}^\star_{s,\epsilon}(\mathbf{X}) - \mathbf{T}^\star_s(\mathbf{X})\|^2
\end{aligned}
\tag{8}
$$

where $\mathbf{T}'_{s,\epsilon}$ is the OT map from $\mathbf{T}^\star_{s,\epsilon\#}\mathcal{P}_s$ to $\mathcal{P}_{s'}$.

Now, we derive the upper bound of the second term $\mathbb{E}_s \|\tilde{\mathbf{T}}_{s,\epsilon} \circ \mathbf{T}^\star_{s,\epsilon}(\mathbf{X}) - \mathbf{T}^\star_s(\mathbf{X})\|^2$ in the right-hand side of equation (8). Because $\tilde{\mathbf{T}}_{s,\epsilon} \circ \mathbf{T}^\star_{s,\epsilon}$ is a transport map, i.e., $\mathbf{T}'_{s,\epsilon\#}\mathbf{T}^\star_{s,\epsilon\#}\mathcal{P}_s = \mathcal{P}_{s'}$, we can apply Proposition 3.3 of Gigli (2011) directly resulting in

$$
\begin{aligned}
\mathbb{E}_s \|\tilde{\mathbf{T}}_{s,\epsilon} \circ \mathbf{T}^\star_{s,\epsilon}(\mathbf{X}) - \mathbf{T}^\star_s(\mathbf{X})\|^2 &\leq \frac{2}{\lambda_{\phi^\star}} \left( \mathbb{E}_s \|\tilde{\mathbf{T}}_{s,\epsilon} \circ \mathbf{T}^\star_{s,\epsilon}(\mathbf{X}) - \mathbf{X}\|^2 - \mathbb{E}_s \|\mathbf{T}^\star_s(\mathbf{X}) - \mathbf{X}\|^2 \right) \\
&\leq \frac{2}{\lambda_{\phi^\star}} \left( \mathbb{E}_s \|\tilde{\mathbf{T}}_{s,\epsilon} \circ \mathbf{T}^\star_{s,\epsilon}(\mathbf{X}) - \mathbf{T}^\star_{s,\epsilon}(\mathbf{X})\|^2 + \mathbb{E}_s \|\mathbf{T}^\star_{s,\epsilon}(\mathbf{X}) - \mathbf{X}\|^2 - \mathbb{E}_s \|\mathbf{T}^\star_s(\mathbf{X}) - \mathbf{X}\|^2 \right) \\
&\leq \frac{2}{\lambda_{\phi^\star}} \epsilon
\end{aligned}
\tag{9}
$$

where $\lambda_{\phi^\star}$ is the supremum of $\lambda$ such that $\phi^\star(x) - \lambda|x|^2/2$ is convex on $\mathcal{X}$. The third inequality of (9) is because the first term of the right-hand-side of the second line is smaller than $\epsilon$ and the second term of the right-hand side is smaller than 0 since the transport cost of $\mathbf{T}^\star_{s,\epsilon}$ is smaller than that of $\mathbf{T}^\star_s$ because $\{\mathbf{T}_s : \mathbf{T}_{s\#}\mathcal{P}_0 = \mathcal{P}_1\} \subset \{\mathbf{T}_s : \mathcal{W}_2(\mathbf{T}_{s\#}\mathcal{P}_0, \mathcal{P}_1) \leq \epsilon\}$. Finally, by plugging equation (9) to (8), we conclude

$$\mathbb{E}_s \|\mathbf{T}^\star_{s,\epsilon}(\mathbf{X}) - \mathbf{T}^\star_s(\mathbf{X})\|^2 \leq \left( \frac{2}{\lambda_{\phi^\star}} + 1 \right) \epsilon.
\tag{10}$$

$\square$

We now prove Proposition 3.3 using Lemma B.4.

**Proposition 3.3** For any given fairness level $\delta \geq 0$, there exist $\delta_1, \delta_2$ depending on $\delta, \gamma, \epsilon$ such that $\mathcal{F}^{\Delta\text{MDP}}(\delta_1; \{\mathbf{T}^\star_{0,\epsilon}\}, \{\mathbf{T}^\star_{1,\epsilon}\}) \subset \mathcal{F}^{\Delta\text{MDP}}(\delta; \mathcal{T}_{0,0}(\gamma), \mathcal{T}_{1,0}(\gamma)) \subset \mathcal{F}^{\Delta\text{MDP}}(\delta_2; \{\mathbf{T}^\star_{0,\epsilon}\}, \{\mathbf{T}^\star_{1,\epsilon}\})$. Furthermore, $\delta_1, \delta_2 \to \delta$ as $\gamma, \epsilon \to 0$.

*Proof of Proposition 3.3.* Let $C_1 := \left(\frac{2}{\lambda_{\phi^\star}} + 1\right)$ in Lemma B.4.

1. Find $\delta_1$ such that $\mathcal{F}^{\Delta\text{MDP}}(\delta_1; \{\mathbf{T}^\star_{0,\epsilon}\}, \{\mathbf{T}^\star_{1,\epsilon}\}) \subset \mathcal{F}^{\Delta\text{MDP}}(\delta; \mathcal{T}_{0,0}(\gamma), \mathcal{T}_{1,0}(\gamma))$.
For any $\mathbf{T}_{s,\gamma} \in \mathcal{T}_{s,0}(\gamma)$, we have that

$$
\begin{aligned}
&\mathbb{E}_s |f(\mathbf{X}, s) - f(\mathbf{T}_{s,\gamma}(\mathbf{X}), s')| \\
&\leq \mathbb{E}_s |f(\mathbf{X}, s) - f(\mathbf{T}^\star_{s,\epsilon}(\mathbf{X}), s')| + \mathbb{E}_s |f(\mathbf{T}^\star_{s,\epsilon}(\mathbf{X}), s') - f(\mathbf{T}_{s,\gamma}(\mathbf{X}), s')| \\
&\leq \delta_1 + \mathbb{E}_s |f(\mathbf{T}^\star_{s,\epsilon}(\mathbf{X}), s') - f(\mathbf{T}^\star_s(\mathbf{X}), s')| + \mathbb{E}_s |f(\mathbf{T}^\star_s(\mathbf{X}), s') - f(\mathbf{T}_{s,\gamma}(\mathbf{X}), s')| \quad (11) \\
&\leq \delta_1 + L \mathbb{E}_s \|\mathbf{T}^\star_s(\mathbf{X}) - \mathbf{T}^\star_{s,\epsilon}(\mathbf{X})\|^2 + L \mathbb{E}_s \|\mathbf{T}^\star_s(\mathbf{X}) - \mathbf{T}_{s,\gamma}(\mathbf{X})\|^2 \\
&\leq \delta_1 + L C_1 \epsilon + L\gamma.
\end{aligned}
$$

By letting $\delta_1 := \delta - L C_1 \epsilon - L\gamma$, we have $\min_{\mathbf{T}_s \in \mathcal{T}_{s,0}(\gamma)} \mathbb{E}_s |f(\mathbf{X}, s) - f(\mathbf{T}_s(\mathbf{X}), s')| \leq \delta$, which concludes the desired result. Moreover, we have $\delta_1 = \delta - L C_1 \epsilon - L\gamma \to \delta$ as $\gamma, \epsilon \to 0$.

2. Find $\delta_2$ such that $\mathcal{F}^{\Delta\text{MDP}}(\delta; \mathcal{T}_{0,0}(\gamma), \mathcal{T}_{1,0}(\gamma)) \subset \mathcal{F}^{\Delta\text{MDP}}(\delta_2; \{\mathbf{T}^\star_{0,\epsilon}\}, \{\mathbf{T}^\star_{1,\epsilon}\})$.
For all $f \in \mathcal{F}^{\Delta\text{MDP}}(\delta; \mathcal{T}_{0,0}(\gamma), \mathcal{T}_{1,0}(\gamma))$, we have $\min_{\mathbf{T}_s \in \mathcal{T}_{s,0}(\gamma)} \mathbb{E}_s |f(\mathbf{X}, s) - f(\mathbf{T}_s(\mathbf{X}), s')| \leq \delta$.
For given $f \in \mathcal{F}^{\Delta\text{MDP}}(\delta; \mathcal{T}_{0,0}(\gamma), \mathcal{T}_{1,0}(\gamma))$, let $\mathbf{T}^f_s := \arg\min_{\mathbf{T}_s \in \mathcal{T}_{s,0}(\gamma)} \mathbb{E}_s |f(\mathbf{X}, s) - f(\mathbf{T}_s(\mathbf{X}), s')|$.
Then, we have that

$$
\begin{aligned}
&\mathbb{E}_s |f(\mathbf{X}, s) - f(\mathbf{T}^\star_{s,\epsilon}(\mathbf{X}), s')| \\
&\leq \mathbb{E}_s |f(\mathbf{X}, s) - f(\mathbf{T}^f_s(\mathbf{X}), s')| + \mathbb{E}_s |f(\mathbf{T}^f_s(\mathbf{X}), s') - f(\mathbf{T}^\star_{s,\epsilon}(\mathbf{X}), s')| \\
&\leq \delta + \mathbb{E}_s |f(\mathbf{T}^f_s(\mathbf{X}), s') - f(\mathbf{T}^\star_s(\mathbf{X}), s')| + \mathbb{E}_s |f(\mathbf{T}^\star_s(\mathbf{X}), s') - f(\mathbf{T}^\star_{s,\epsilon}(\mathbf{X}), s')| \quad (12) \\
&\leq \delta + L \mathbb{E}_s \|\mathbf{T}^\star_s(\mathbf{X}) - \mathbf{T}^f_s(\mathbf{X})\|^2 + L \mathbb{E}_s \|\mathbf{T}^\star_s(\mathbf{X}) - \mathbf{T}^\star_{s,\epsilon}(\mathbf{X})\|^2 \\
&\leq \delta + L\gamma + L C_1 \epsilon.
\end{aligned}
$$

By letting $\delta_2 := \delta + L\gamma + L C_1 \epsilon$, we have $\mathbb{E}_s |f(\mathbf{X}, s) - f(\mathbf{T}^\star_{s,\epsilon}(\mathbf{X}), s')| \leq \delta_2$, which concludes the desired result. Moreover, we have $\delta_2 = \delta + L\gamma + L C_1 \epsilon \to \delta$ as $\gamma, \epsilon \to 0$. $\square$

### B.3 PROOF OF THEOREM 3.1 AND 3.2

Proofs for Theorem 3.1 and 3.2 are provided in Section C.2 and C.3, respectively, with details.

### B.4 PROOF OF PROPOSITION 3.4

*Proof of Proposition 3.4.* Once we observe $\mathbf{x}_0$, the randomness $\epsilon_0$ is observed as $\epsilon_0 = B^{-1}\mathbf{x}_0 - \mu_0$. By replacing the sensitive attribute on the randomness $\epsilon_0$, we obtain $\sigma_1^{-1}(B^{-1}\tilde{\mathbf{x}}_0^{\mathrm{CF}} - \mu_1) = \sigma_0^{-1}(B^{-1}\mathbf{x}_0 - \mu_0)$. Then, its counterfactual becomes $\tilde{\mathbf{x}}_0^{\mathrm{CF}} = B\mu_1 + \sigma_1\sigma_0^{-1}(\mathbf{x}_0 - B\mu_0)$. Then, we prove Proposition 3.4 by showing the if and only if condition as follows.

$$
\begin{aligned}
\mathbf{W}_0 &= (\sigma_0^2 BDB^\top)^{-1/2}\left((\sigma_0^2 BDB^\top)^{1/2}\sigma_1^2 BDB^\top(\sigma_0^2 BDB^\top)^{1/2}\right)^{1/2}(\sigma_0^2 BDB^\top)^{-1/2} \\
&= \sigma_1\sigma_0^{-1}(BDB^\top)^{-1/2}\left((BDB^\top)^{1/2}BDB^\top(BDB^\top)^{1/2}\right)^{1/2}(BDB^\top)^{-1/2} \\
&= \sigma_1\sigma_0^{-1}(BDB^\top)^{-1/2}\left((BDB^\top)^2\right)^{1/2}(BDB^\top)^{-1/2} \\
&= \sigma_1\sigma_0^{-1}.
\end{aligned}
\tag{13}
$$

The same result can be done for $\mathbf{x}_1$. Hence, we conclude $\mathbf{W}_s = \sigma_{s'}\sigma_s^{-1}$. $\square$

## B.5 Proof of Theorem B.5

We provide sufficient conditions, i.e., (C3) and (C4) described below, under which the fairness consistencies of $\widehat{f}_s^{\mathrm{FTM}}$ with respect to $\Delta\mathrm{WDP}$ and $\Delta\overline{\mathrm{DP}}$ hold. (C3) Both $\mathcal{P}_0$ and $\mathcal{P}_1$ have densities and belong to $(\beta - 1)$-Hölder smooth function class with $\beta > 1$. (C4) $\mathcal{F}_s, s \in \{0, 1\}$ are subsets of $\zeta$-Hölder smooth function class with $\zeta \geq 1$ s.t. $\mathcal{R}_n(\mathcal{F}_s) \leq \mathcal{O}(b_n)$ for some positive sequence $b_n \to 0$, where $\mathcal{R}_n(\mathcal{F}_s)$ is the Rademacher complexity of $\mathcal{F}_s$.

The definitions of the Hölder smooth function class and Rademacher complexity are in Section A of Appendix. We assume the Hölder smoothness since they can be approximated well by DNNs (Schmidt-Hieber, 2020).

**Theorem B.5** (Fairness consistency of $\widehat{f}_s^{\mathrm{FTM}}$). *Under (C3) and (C4), there exist $\epsilon_n \downarrow 0$ and DNN function classes $\mathcal{T}_n^{\mathrm{DNN}}$ and $\mathcal{H}_n^{\mathrm{DNN}}$ such that $\widehat{f}_s^{\mathrm{FTM}}$ with $\epsilon = \epsilon_n, \mathcal{T}_s = \mathcal{T}_n^{\mathrm{DNN}}$ and $\mathcal{H} = \mathcal{H}_n^{\mathrm{DNN}}$ satisfies $\Delta\mathrm{WDP}(\widehat{f}_s^{\mathrm{FTM}}), \Delta\overline{\mathrm{DP}}(\widehat{f}_s^{\mathrm{FTM}}) \leq \delta + \mathcal{O}\left(\mathcal{R}_n(\mathcal{F}_s) + n^{-\frac{\gamma}{2\gamma+d}}(\log n)^\tau\right)$ for some constant $\tau > 0$ with probability at least $1 - 1/n$, where $\gamma = \beta \wedge \zeta$.*

Theorem B.5 implies that $\widehat{f}_s^{\mathrm{FTM}}$ is fairness-consistent as long as $\mathcal{R}_n(\mathcal{F}_s)$ converges to 0, which is satisfied by many popularly used $\mathcal{F}_s$ such as linear models and sparse DNNs (Schmidt-Hieber, 2020). The complexities of $\mathcal{H}_n^{\mathrm{DNN}}$ and $\mathcal{T}_n^{\mathrm{DNN}}$ depend on the smoothnesses of $\mathcal{F}_s$ and $\mathcal{P}_s, s \in \{0, 1\}$, respectively, whose relations are given in the proof of Theorem B.5 (Section B.5 of Appendix).

Let $\mathcal{T}_\beta$ be the (vector-valued) $\beta$-Hölder smooth function class and $\mathcal{H}_\gamma$ be the $\gamma$-Hölder smooth function class ($\gamma = \beta \wedge \zeta$). Also let $\mathcal{T}_n^{\mathrm{DNN}}$ and $\mathcal{H}_n^{\mathrm{DNN}}$ be the DNN function classes defined in Lemma A.8.

**Lemma B.6.** *There exists $\mathbf{T} \in \mathcal{T}_n^{\mathrm{DNN}}$ satisfying $d_{\mathcal{H}_n^{\mathrm{DNN}}}(\mathbf{T}_{\#}\mathcal{P}_0, \mathcal{P}_1) \leq \mathcal{O}\left(n^{-\frac{\beta}{2\beta+d}}\right)$ up to a logarithm factor.*

*Proof of Lemma B.6.* We know that $\exists \mathbf{T}_{\beta,0}^\star \in \mathcal{T}_\beta$ such that $d_{\mathcal{H}_n^{\mathrm{DNN}}}(\mathbf{T}_{\beta,0\#}^\star \mathcal{P}_0, \mathcal{P}_1) = 0$ under (C3) because there exists the population OT map in $\mathcal{T}_\beta$ (Theorem 12.50-(iii) of Villani (2008), Caffarelli (1996a)) and the IPM induced by the population OT map is exactly zero regardless of $\mathcal{H}_n^{\mathrm{DNN}}$.

By Lemma A.8, we have there exists $\mathbf{T} \in \mathcal{T}_n^{\mathrm{DNN}}$ such that

$$
\begin{aligned}
d_{\mathcal{H}_n^{\mathrm{DNN}}}(\mathbf{T}_{\#}\mathcal{P}_0, \mathcal{P}_1) &\leq d_{\mathcal{H}_n^{\mathrm{DNN}}}(\mathbf{T}_{\#}\mathcal{P}_0, \mathbf{T}_{\beta,0\#}^\star \mathcal{P}_0) + d_{\mathcal{H}_n^{\mathrm{DNN}}}(\mathbf{T}_{\beta,0\#}^\star \mathcal{P}_0, \mathcal{P}_1) \\
&= d_{\mathcal{H}_n^{\mathrm{DNN}}}(\mathbf{T}_{\#}\mathcal{P}_0, \mathbf{T}_{\beta,0\#}^\star \mathcal{P}_0) \\
&= \sup_{h \in \mathcal{H}_n^{\mathrm{DNN}}} \left| \int h \circ \mathbf{T}(\mathbf{X}) d\mathcal{P}_0(\mathbf{X}) - \int h \circ \mathbf{T}_{\beta,0}^\star(\mathbf{X}) d\mathcal{P}_0(\mathbf{X}) \right| \\
&\leq C \left\| \mathbf{T} - \mathbf{T}_{\beta,0}^\star \right\|_\infty \\
&\leq \mathcal{O}\left(n^{-\frac{\beta}{2\beta+d}}\right)
\end{aligned}
\tag{14}
$$

up to a logarithm factor for some constant $C$ only depending on the first derivative of $h$. □

**Lemma B.7.** *For any $\mathbf{T} \in \mathcal{T}_n^{\mathrm{DNN}}$ satisfying $d_{\mathcal{H}_n^{\mathrm{DNN}}}(\mathbf{T}_{\#}\mathcal{P}_0, \mathcal{P}_1) \leq \mathcal{O}\left(n^{-\frac{\beta}{2\beta+d}}\right)$ up to a logarithm factor, $d_{\mathcal{H}_n^{\mathrm{DNN}}}(\mathbf{T}_{\#}\mathcal{P}_{n,0}, \mathcal{P}_{n,1}) \leq \mathcal{O}\left(n^{-\frac{\gamma}{2\gamma+d}}\right)$ up to a logarithm factor with probability at least $1 - 1/n$.*

*Proof of Lemma B.7.* We derive the bound of $d_{\mathcal{H}_n^{\text{DNN}}}(\mathbf{T}_{\#}\mathcal{P}_{n,0}, \mathcal{P}_{n,1})$ as

$$
\begin{aligned}
d_{\mathcal{H}_n^{\text{DNN}}}(\mathbf{T}_{\#}\mathcal{P}_{n,0}, \mathcal{P}_{n,1}) &\leq d_{\mathcal{H}_n^{\text{DNN}}}(\mathbf{T}_{\#}\mathcal{P}_0, \mathcal{P}_1) + \sup_{h \in \mathcal{H}_n^{\text{DNN}}} |\int h \circ \mathbf{T}(\mathbf{X}) d(\mathcal{P}_0 - \mathcal{P}_{n,0})(\mathbf{X})| \\
&\quad + \sup_{h \in \mathcal{H}_n^{\text{DNN}}} |\int h(\mathbf{X}) d(\mathcal{P}_1 - \mathcal{P}_{n,1})(\mathbf{X})| \\
&\leq \mathcal{O}\left(n^{-\frac{\beta}{2\beta+d}}\right) + \sup_{h \in \mathcal{H}_n^{\text{DNN}}} |\int h \circ \mathbf{T}(\mathbf{X}) d(\mathcal{P}_0 - \mathcal{P}_{n,0})(\mathbf{X})| \quad (15) \\
&\quad + \sup_{h \in \mathcal{H}_n^{\text{DNN}}} |\int h(\mathbf{X}) d(\mathcal{P}_1 - \mathcal{P}_{n,1})(\mathbf{X})| \\
&\overset{\text{by Lemma } A.6, A.8}{\leq} \mathcal{O}\left(n^{-\frac{\beta}{2\beta+d}} + n^{-\frac{\gamma}{2\gamma+d}}\right)
\end{aligned}
$$

up to a logarithm factor with probability $1 - 1/n$ at least. Because we set $\gamma = \beta \wedge \zeta$, the rate becomes $\mathcal{O}\left(n^{-\frac{\beta}{2\beta+d}} + n^{-\frac{\gamma}{2\gamma+d}}\right) = \mathcal{O}\left(n^{-\frac{\gamma}{2\gamma+d}}\right)$.

$\square$

**Lemma B.8.** *For any* $\mathbf{T} \in \mathcal{T}_n^{\text{DNN}}$ *satisfying* $d_{\mathcal{H}_n^{\text{DNN}}}(\mathbf{T}_{\#}\mathcal{P}_{n,0}, \mathcal{P}_{n,1}) \leq \mathcal{O}(n^{-\frac{\gamma}{2\gamma+d}})$ *up to a logarithm factor with probability at least* $1 - 1/n$, $d_{\mathcal{H}_\gamma}(\mathbf{T}_{\#}\mathcal{P}_0, \mathcal{P}_1) \leq \mathcal{O}(n^{-\frac{\gamma}{2\gamma+d}})$ *up to a logarithm factor with probability at least* $1 - 1/n$.

*Proof of Lemma B.8.* We decompose the difference between $d_{\mathcal{H}_\gamma}(\mathbf{T}_{\#}\mathcal{P}_0, \mathcal{P}_1)$ and $d_{\mathcal{H}_n^{\text{DNN}}}(\mathbf{T}_{\#}\mathcal{P}_{n,0}, \mathcal{P}_{n,1})$ as:

$$
\begin{aligned}
d_{\mathcal{H}_\gamma}(\mathbf{T}_{\#}\mathcal{P}_0, \mathcal{P}_1) &= d_{\mathcal{H}_n^{\text{DNN}}}(\mathbf{T}_{\#}\mathcal{P}_{n,0}, \mathcal{P}_{n,1}) + d_{\mathcal{H}_\gamma}(\mathbf{T}_{\#}\mathcal{P}_0, \mathcal{P}_1) - d_{\mathcal{H}_n^{\text{DNN}}}(\mathbf{T}_{\#}\mathcal{P}_{n,0}, \mathcal{P}_{n,1}) \\
&= d_{\mathcal{H}_n^{\text{DNN}}}(\mathbf{T}_{\#}\mathcal{P}_{n,0}, \mathcal{P}_{n,1}) + d_{\mathcal{H}_\gamma}(\mathbf{T}_{\#}\mathcal{P}_0, \mathcal{P}_1) - d_{\mathcal{H}_n^{\text{DNN}}}(\mathbf{T}_{\#}\mathcal{P}_0, \mathcal{P}_1) \\
&\quad + d_{\mathcal{H}_n^{\text{DNN}}}(\mathbf{T}_{\#}\mathcal{P}_0, \mathcal{P}_1) - d_{\mathcal{H}_n^{\text{DNN}}}(\mathbf{T}_{\#}\mathcal{P}_{n,0}, \mathcal{P}_{n,1}) \\
&\leq d_{\mathcal{H}_n^{\text{DNN}}}(\mathbf{T}_{\#}\mathcal{P}_{n,0}, \mathcal{P}_{n,1}) + \sup_{h \in \mathcal{H}_\gamma} \left|\int h \circ \mathbf{T}(\mathbf{X}) d\mathcal{P}_0(\mathbf{X}) - \int h(\mathbf{X}) d\mathcal{P}_1(\mathbf{X})\right| \\
&\quad - \sup_{h \in \mathcal{H}_n^{\text{DNN}}} \left|\int h \circ \mathbf{T}(\mathbf{X}) d\mathcal{P}_0(\mathbf{X}) - \int h(\mathbf{X}) d\mathcal{P}_1(\mathbf{X})\right| + \left|d_{\mathcal{H}_n^{\text{DNN}}}(\mathbf{T}_{\#}\mathcal{P}_0, \mathcal{P}_1) - d_{\mathcal{H}_n^{\text{DNN}}}(\mathbf{T}_{\#}\mathcal{P}_{n,0}, \mathcal{P}_{n,1})\right|.
\end{aligned}
$$
$$(16)$$

The first term of the right-hand side of the equation (16), i.e., $d_{\mathcal{H}_n^{\text{DNN}}}(\mathbf{T}_{\#}\mathcal{P}_{n,0}, \mathcal{P}_{n,1})$, is bounded by $\mathcal{O}(n^{-\gamma/(2\gamma+d)})$ up to a logarithm factor by the assumption.

We denote $h_\gamma^\star \in \mathcal{H}_\gamma$ a $\gamma$-Hölder smooth function satisfying $\left|\int h_\gamma^\star \circ \mathbf{T}(\mathbf{X}) d\mathcal{P}_0(\mathbf{X}) - \int h_\gamma^\star(\mathbf{X}) d\mathcal{P}_1(\mathbf{X})\right| = \sup_{h \in \mathcal{H}_\gamma} \left|\int h \circ \mathbf{T}(\mathbf{X}) d\mathcal{P}_0(\mathbf{X}) - \int h(\mathbf{X}) d\mathcal{P}_1(\mathbf{X})\right|$. Similarly, let $\tilde{h}_n \in \mathcal{H}_n^{\text{DNN}}$ be the DNN function satisfying $\left|\int \tilde{h}_n \circ \mathbf{T}(\mathbf{X}) d\mathcal{P}_0(\mathbf{X}) - \int \tilde{h}_n(\mathbf{X}) d\mathcal{P}_1(\mathbf{X})\right| = \sup_{h \in \mathcal{H}_n^{\text{DNN}}} \left|\int h \circ \mathbf{T}(\mathbf{X}) d\mathcal{P}_0(\mathbf{X}) - \int h(\mathbf{X}) d\mathcal{P}_1(\mathbf{X})\right|$. On the other hand, by Lemma A.8, there exists $h_n^\star \in \mathcal{H}_n^{\text{DNN}}$ satisfying $\|h_n^\star - h_\gamma^\star\|_\infty \leq \mathcal{O}\left(n^{-\frac{\gamma}{2\gamma+d}}\right)$ up to a logarithm factor.

Then, we have

$$
\begin{aligned}
& \sup_{h \in \mathcal{H}_\gamma} \left| \int h \circ \mathbf{T}(\mathbf{X}) d\mathcal{P}_0(\mathbf{X}) - \int h(\mathbf{X}) d\mathcal{P}_1(\mathbf{X}) \right| - \sup_{h \in \mathcal{H}_n^{\mathrm{DNN}}} \left| \int h \circ \mathbf{T}(\mathbf{X}) d\mathcal{P}_0(\mathbf{X}) - \int h(\mathbf{X}) d\mathcal{P}_1(\mathbf{X}) \right| \\
&= \left| \int h_\gamma^\star \circ \mathbf{T}(\mathbf{X}) d\mathcal{P}_0(\mathbf{X}) - \int h_\gamma^\star(\mathbf{X}) d\mathcal{P}_1(\mathbf{X}) \right| - \left| \int \tilde{h}_n \circ \mathbf{T}(\mathbf{X}) d\mathcal{P}_0(\mathbf{X}) - \int \tilde{h}_n(\mathbf{X}) d\mathcal{P}_1(\mathbf{X}) \right| \\
&\leq \left| \int h_\gamma^\star \circ \mathbf{T}(\mathbf{X}) d\mathcal{P}_0(\mathbf{X}) - \int h_\gamma^\star(\mathbf{X}) d\mathcal{P}_1(\mathbf{X}) \right| - \left| \int h_n^\star \circ \mathbf{T}(\mathbf{X}) d\mathcal{P}_0(\mathbf{X}) - \int h_n^\star(\mathbf{X}) d\mathcal{P}_1(\mathbf{X}) \right| \\
&\leq \left| \int h_\gamma^\star \circ \mathbf{T}(\mathbf{X}) d\mathcal{P}_0(\mathbf{X}) - \int h_\gamma^\star(\mathbf{X}) d\mathcal{P}_1(\mathbf{X}) - \int h_n^\star \circ \mathbf{T}(\mathbf{X}) d\mathcal{P}_0(\mathbf{X}) + \int h_n^\star(\mathbf{X}) d\mathcal{P}_1(\mathbf{X}) \right| \\
&\leq \left| \int (h_\gamma^\star \circ \mathbf{T}(\mathbf{X}) - h_n^\star \circ \mathbf{T}(\mathbf{X})) d\mathcal{P}_0(\mathbf{X}) \right| + \left| \int (h_\gamma^\star(\mathbf{X}) - h_n^\star(\mathbf{X})) d\mathcal{P}_1(\mathbf{X}) \right| \\
&\leq 2 \left\| h_\gamma^\star - h_n^\star \right\|_\infty \\
&\leq \mathcal{O}\left( n^{-\frac{\gamma}{2\gamma+d}} \right)
\end{aligned}
\tag{17}
$$

up to a logarithm factor.

The last term of the right-hand side of (16) is bounded as:

$$
\begin{aligned}
& \left| d_{\mathcal{H}_n^{\mathrm{DNN}}} \left( \mathbf{T}_\# \mathcal{P}_0, \mathcal{P}_1 \right) - d_{\mathcal{H}_n^{\mathrm{DNN}}} \left( \mathbf{T}_\# \mathcal{P}_{n,0}, \mathcal{P}_{n,1} \right) \right| \\
&\leq \sup_{h \in \mathcal{H}_n^{\mathrm{DNN}}} \left| \int h \circ \mathbf{T}(\mathbf{X}) d(\mathcal{P}_0 - \mathcal{P}_{n,0})(\mathbf{X}) \right| + \sup_{h \in \mathcal{H}_n^{\mathrm{DNN}}} \left| \int h(\mathbf{X}) d(\mathcal{P}_1 - \mathcal{P}_{n,1})(\mathbf{X}) \right| \\
&\overset{\text{by Lemma } A.6, A.8}{\leq} \mathcal{O}\left( n^{-\frac{\gamma}{2\gamma+d}} \right)
\end{aligned}
\tag{18}
$$

up to logarithm with probability at least $1 - 1/n$.

Hence, we conclude that $d_{\mathcal{H}_\gamma}(\mathbf{T}_\# \mathcal{P}_0, \mathcal{P}_1) \leq \mathcal{O}(n^{-\frac{\gamma}{2\gamma+d}})$ up to a logarithm factor with probability at least $1 - 1/n$. $\square$

*Proof of Theorem B.5.* For given pair of transport maps $\mathbf{T}_s \in \mathcal{T}_n^{\mathrm{DNN}}, s \in \{0, 1\}$ satisfying $d_{\mathcal{H}_n^{\mathrm{DNN}}}(\mathbf{T}_{s\#} \mathcal{P}_{n,s}, \mathcal{P}_{n,s'}) \leq \mathcal{O}(n^{-\frac{\gamma}{2\gamma+d}})$ up to a logarithm factor with probability at least $1 - 1/n$, we define $\widehat{f}_{\mathbf{T}_s}$ as the minimizer of $\mathbb{E}_n(l(Y, f(\mathbf{X}, S)))$ on $\{f \in \mathcal{F}_s : \mathbb{E}_{n,s}(|f(\mathbf{X}, s) - f(\mathbf{T}_s(\mathbf{X}), s')|) \leq \delta\}$.

**($\Delta$WDP)**

Without loss of generality, consider $\widehat{f}_{\mathbf{T}_0}$ only, because it is proved clearly with the same arguments for the case of $\widehat{f}_{\mathbf{T}_1}$. We decompose the level of fairness $\Delta \mathrm{WDP}(\widehat{f}_{\mathbf{T}_0})$ as:

$$
\begin{aligned}
\Delta \mathrm{WDP}(\widehat{f}_{\mathbf{T}_0}) &= \sup_{u \in \mathcal{L}_1} \left| \mathbb{E}_0(u \circ \widehat{f}_{\mathbf{T}_0}(\mathbf{X}, 0)) - \mathbb{E}_1(u \circ \widehat{f}_{\mathbf{T}_0}(\mathbf{X}, 1)) \right| \\
&\leq \sup_{u \in \mathcal{L}_1} \left| \mathbb{E}_0(u \circ \widehat{f}_{\mathbf{T}_0}(\mathbf{X}, 0)) - \mathbb{E}_0(u \circ \widehat{f}_{\mathbf{T}_0}(\mathbf{T}_0(\mathbf{X}), 0) \right| \\
&\quad + \sup_{u \in \mathcal{L}_1} \left| \mathbb{E}_0(u \circ \widehat{f}_{\mathbf{T}_0}(\mathbf{T}_0(\mathbf{X}), 0) - \mathbb{E}_1(u \circ \widehat{f}_{\mathbf{T}_0}(\mathbf{X}, 1)) \right|.
\end{aligned}
\tag{19}
$$

The first term of the equation (19) is bounded as follows.

By Lemma A.6 and $|a - b| - |c - d| \le |a - b - c + d| \le |a - c| + |b - d|$, we have the generalization bound as

$$\sup_{u \in \mathcal{L}_1} \left| \mathbb{E}_0 \left( u \circ \widehat{f}_{\mathbf{T}_0}(\mathbf{X}, 0) \right) - \mathbb{E}_0 \left( u \circ \widehat{f}_{\mathbf{T}_0}(\mathbf{T}_0(\mathbf{X}), 0) \right) \right|$$

$$- \sup_{u \in \mathcal{L}_1} \left| \mathbb{E}_{n,0} \left( u \circ \widehat{f}_{\mathbf{T}_0}(\mathbf{X}, 0) \right) - \mathbb{E}_{n,0} \left( u \circ \widehat{f}_{\mathbf{T}_0}(\mathbf{T}_0(\mathbf{X}), 0) \right) \right|$$

$$\le \sup_{u \in \mathcal{L}_1} \left| \mathbb{E}_0 \left( u \circ \widehat{f}_{\mathbf{T}_0}(\mathbf{X}, 0) \right) - \mathbb{E}_{n,0} \left( u \circ \widehat{f}_{\mathbf{T}_0}(\mathbf{X}, 0) \right) - \mathbb{E}_0 \left( u \circ \widehat{f}_{\mathbf{T}_0}(\mathbf{T}_0(\mathbf{X}), 0) \right) + \mathbb{E}_{n,0} \left( u \circ \widehat{f}_{\mathbf{T}_0}(\mathbf{T}_0(\mathbf{X}), 0) \right) \right|$$

$$\le \sup_{u \in \mathcal{L}_1} \left| \mathbb{E}_0 \left( u \circ \widehat{f}_{\mathbf{T}_0}(\mathbf{X}, 0) \right) - \mathbb{E}_{n,0} \left( u \circ \widehat{f}_{\mathbf{T}_0}(\mathbf{X}, 0) \right) \right|$$

$$+ \sup_{u \in \mathcal{L}_1} \left| \mathbb{E}_0 \left( u \circ \widehat{f}_{\mathbf{T}_0}(\mathbf{T}_0(\mathbf{X}), 0) \right) - \mathbb{E}_{n,0} \left( u \circ \widehat{f}_{\mathbf{T}_0}(\mathbf{T}_0(\mathbf{X}), 0) \right) \right|$$

$$\le C \left( \mathcal{R}_n(\mathcal{L}_1 \circ \mathcal{F}_0) + \sqrt{\frac{\log n}{n}} \right)$$

$$\overset{\mathcal{R}_n(\mathcal{L}_1 \circ \mathcal{F}_0) \le 1 \cdot \mathcal{R}_n(\mathcal{F}_0)}{\le} C \left( \mathcal{R}_n(\mathcal{F}_0) + \sqrt{\frac{\log n}{n}} \right)$$

$$\tag{20}$$

with probability $1 - 1/n$ at least.

By definition of $\widehat{f}_{\mathbf{T}_0}$, we have that $\mathbb{E}_{n,0} \left( |\widehat{f}_{\mathbf{T}_0}(\mathbf{X}, 0) - \widehat{f}_{\mathbf{T}_0}(\mathbf{T}_0(\mathbf{X}), 0)| \right) \le \delta$, which directly implies

$$\sup_{u \in \mathcal{L}_1} \left| \mathbb{E}_{n,0} \left( u \circ \widehat{f}_{\mathbf{T}_0}(\mathbf{X}, 0) \right) - \mathbb{E}_{n,0} \left( u \circ \widehat{f}_{\mathbf{T}_0}(\mathbf{T}_0(\mathbf{X}), 0) \right) \right|$$

$$\le \sup_{u \in \mathcal{L}_1} \mathbb{E}_{n,0} \left| u \circ \widehat{f}_{\mathbf{T}_0}(\mathbf{X}, 0) - u \circ \widehat{f}_{\mathbf{T}_0}(\mathbf{T}_0(\mathbf{X}), 0) \right| \tag{21}$$

$$\overset{u \in \mathcal{L}_1}{\le} \mathbb{E}_{n,0} \left| \widehat{f}_{\mathbf{T}_0}(\mathbf{X}, 0) - \widehat{f}_{\mathbf{T}_0}(\mathbf{T}_0(\mathbf{X}), 0) \right| \le \delta.$$

Combining the above two inequalities (20) and (21) concludes that with probability $1 - 1/n$,

$$\sup_{u \in \mathcal{L}_1} \left| \mathbb{E}_0 \left( u \circ \widehat{f}_{\mathbf{T}_0}(\mathbf{X}, 0) \right) - \mathbb{E}_0 \left( u \circ \widehat{f}_{\mathbf{T}_0}(\mathbf{T}_0(\mathbf{X}), 0) \right) \right|$$

$$\le \sup_{u \in \mathcal{L}_1} \left| \mathbb{E}_{n,0} \left( u \circ \widehat{f}_{\mathbf{T}_0}(\mathbf{X}, 0) \right) - \mathbb{E}_{n,0} \left( u \circ \widehat{f}_{\mathbf{T}_0}(\mathbf{T}_0(\mathbf{X}), 0) \right) \right| + C \left( \mathcal{R}_n(\mathcal{F}_0) + \sqrt{\frac{\log n}{n}} \right) \tag{22}$$

$$\le \delta + C \left( \mathcal{R}_n(\mathcal{F}_0) + \sqrt{\frac{\log n}{n}} \right).$$

The second term of the equation (19) is bounded as follows.

$$\sup_{u \in \mathcal{L}_1} \left| \mathbb{E}_0 \left( u \circ \widehat{f}_{\mathbf{T}_0}(\mathbf{T}_0(\mathbf{X}), 0) \right) - \mathbb{E}_1 \left( u \circ \widehat{f}_{\mathbf{T}_0}(\mathbf{X}, 1) \right) \right|$$

$$\overset{(C4)}{\le} C \sup_{h \in \mathcal{H}_\gamma} \left| \int h \circ \mathbf{T}_0(\mathbf{X}) d\mathcal{P}_0(\mathbf{X}) - \int h(\mathbf{X}) d\mathcal{P}_1(\mathbf{X}) \right| \tag{23}$$

$$= C d_{\mathcal{H}_\gamma}(\mathbf{T}_{0\#}\mathcal{P}_0, \mathcal{P}_1)$$

$$\overset{\text{by Lemma} B.8}{\le} \mathcal{O}(n^{-\frac{\gamma}{2\gamma + d}}).$$

with probability at least $1 - 1/n$.

Applying the upper bounds of (22) and (23) to (19), we conclude $\Delta \text{WDP}(\widehat{f}_{\mathbf{T}_0}) \le \delta + \mathcal{O}(\mathcal{R}_n(\mathcal{F}_0) + n^{-\frac{\gamma}{2\gamma + d}})$ up to a logarithm factor with probability at least $1 - 1/n$.

By using exactly the same arguments, we can derive the bound $\Delta\text{WDP}(\widehat{f}_{\mathbf{T}_1}) \leq \delta + \mathcal{O}(\mathcal{R}_n(\mathcal{F}_1) + n^{-\frac{\gamma}{2\gamma+d}})$ up to a logarithm factor with probability at least $1 - 1/n$.

Thus, we have the desired result that $\Delta\text{WDP}(\widehat{f}_{\mathbf{T}_s}) \leq \delta + \mathcal{O}(\mathcal{R}_n(\mathcal{F}_s) + n^{-\frac{\gamma}{2\gamma+d}})$ up to a logarithm factor with probability at least $1 - 1/n$.

Note that if we choose $\epsilon_n = \mathcal{O}(n^{-\frac{\gamma}{2\gamma+d}})$ up to a logarithmic factor, then $\widehat{\mathbf{T}}_s$ satisfies $d_{\mathcal{H}_n^{\text{DNN}}}(\widehat{\mathbf{T}}_{s\#}\mathcal{P}_{n,s}, \mathcal{P}_{n,s'}) \leq \mathcal{O}(n^{-\frac{\gamma}{2\gamma+d}})$ up to a logarithmic factor by the definition of $\widehat{\mathbf{T}}_s$. Thus, we conclude that

$$\Delta\text{WDP}(\widehat{f}_s^{\text{FTM}}) \leq \delta + \mathcal{O}(\mathcal{R}_n(\mathcal{F}_s) + n^{-\frac{\gamma}{2\gamma+d}})$$

up to a logarithmic factor with probability at least $1 - 1/n$. Finally, Lemmas B.6 and B.7 ensure the existence of $\widehat{\mathbf{T}}_s$ satisfying $d_{\mathcal{H}_n^{\text{DNN}}}(\widehat{\mathbf{T}}_{s\#}\mathcal{P}_{n,s}, \mathcal{P}_{n,s'}) \leq \mathcal{O}(n^{-\frac{\gamma}{2\gamma+d}})$ up to a logarithmic factor, which completes the proof.

**($\Delta\overline{\text{DP}}$)**

Without loss of generality, consider $\widehat{f}_{\mathbf{T}_0}$ only, because it is proved clearly with the same arguments for the case of $\widehat{f}_{\mathbf{T}_1}$. We decompose the level of fairness $\Delta\overline{\text{DP}}(\widehat{f}_{\mathbf{T}_0})$ as:

$$\begin{aligned}
\Delta\overline{\text{DP}}(\widehat{f}_{\mathbf{T}_0}) &= \left| \mathbb{E}_0(\widehat{f}_{\mathbf{T}_0}(\mathbf{X}, 0)) - \mathbb{E}_1(\widehat{f}_{\mathbf{T}_0}(\mathbf{X}, 1)) \right| \\
&\leq \left| \mathbb{E}_0(\widehat{f}_{\mathbf{T}_0}(\mathbf{X}, 0)) - \mathbb{E}_0(\widehat{f}_{\mathbf{T}_0}(\mathbf{T}_0(\mathbf{X}), 0) \right| + \left| \mathbb{E}_0(\widehat{f}_{\mathbf{T}_0}(\mathbf{T}_0(\mathbf{X}), 0) - \mathbb{E}_1(\widehat{f}_{\mathbf{T}_0}(\mathbf{X}, 1)) \right|.
\end{aligned}$$
(24)

The first term of the equation (24) is bounded as follows.

By Lemma A.6 and $|a - b| - |c - d| \leq |a - b - c + d| \leq |a - c| + |b - d|$, we have the generalization bound as

$$\begin{aligned}
&\left| \mathbb{E}_0\left(\widehat{f}_{\mathbf{T}_0}(\mathbf{X}, 0)\right) - \mathbb{E}_0\left(\widehat{f}_{\mathbf{T}_0}(\mathbf{T}_0(\mathbf{X}), 0)\right) \right| - \left| \mathbb{E}_{n,0}\left(\widehat{f}_{\mathbf{T}_0}(\mathbf{X}, 0)\right) - \mathbb{E}_{n,0}\left(\widehat{f}_{\mathbf{T}_0}(\mathbf{T}_0(\mathbf{X}), 0)\right) \right| \\
&\leq \left| \mathbb{E}_0\left(\widehat{f}_{\mathbf{T}_0}(\mathbf{X}, 0)\right) - \mathbb{E}_{n,0}\left(\widehat{f}_{\mathbf{T}_0}(\mathbf{X}, 0)\right) - \mathbb{E}_0\left(\widehat{f}_{\mathbf{T}_0}(\mathbf{T}_0(\mathbf{X}), 0)\right) + \mathbb{E}_{n,0}\left(\widehat{f}_{\mathbf{T}_0}(\mathbf{T}_0(\mathbf{X}), 0)\right) \right| \\
&\leq \left| \mathbb{E}_0\left(\widehat{f}_{\mathbf{T}_0}(\mathbf{X}, 0)\right) - \mathbb{E}_{n,0}\left(\widehat{f}_{\mathbf{T}_0}(\mathbf{X}, 0)\right) \right| + \left| \mathbb{E}_0\left(\widehat{f}_{\mathbf{T}_0}(\mathbf{T}_0(\mathbf{X}), 0)\right) - \mathbb{E}_{n,0}\left(\widehat{f}_{\mathbf{T}_0}(\mathbf{T}_0(\mathbf{X}), 0)\right) \right| \\
&\leq C\left( \mathcal{R}_n(\mathcal{F}_0) + \sqrt{\frac{\log n}{n}} \right)
\end{aligned}$$
(25)

with probability $1 - 1/n$ at least.

By definition of $\widehat{f}_{\mathbf{T}_0}$, we have that $\mathbb{E}_{n,0}\left(|\widehat{f}_{\mathbf{T}_0}(\mathbf{X}, 0) - \widehat{f}_{\mathbf{T}_0}(\mathbf{T}_0(\mathbf{X}), 0)|\right) \leq \delta$, which directly implies

$$\left| \mathbb{E}_{n,0}\left(\widehat{f}_{\mathbf{T}_0}(\mathbf{X}, 0)\right) - \mathbb{E}_{n,0}\left(\widehat{f}_{\mathbf{T}_0}(\mathbf{T}_0(\mathbf{X}), 0)\right) \right| \leq \mathbb{E}_{n,0}\left| \widehat{f}_{\mathbf{T}_0}(\mathbf{X}, 0) - \widehat{f}_{\mathbf{T}_0}(\mathbf{T}_0(\mathbf{X}), 0) \right| \leq \delta. \quad (26)$$

Combining the above two inequalities (25) and (26) concludes that with probability $1 - 1/n$,

$$\begin{aligned}
&\left| \mathbb{E}_0\left(\widehat{f}_{\mathbf{T}_0}(\mathbf{X}, 0)\right) - \mathbb{E}_0\left(\widehat{f}_{\mathbf{T}_0}(\mathbf{T}_0(\mathbf{X}), 0)\right) \right| \\
&\leq \left| \mathbb{E}_{n,0}\left(\widehat{f}_{\mathbf{T}_0}(\mathbf{X}, 0)\right) - \mathbb{E}_{n,0}\left(\widehat{f}_{\mathbf{T}_0}(\mathbf{T}_0(\mathbf{X}), 0)\right) \right| + C\left( \mathcal{R}_n(\mathcal{F}_0) + \sqrt{\frac{\log n}{n}} \right) \\
&\leq \delta + C\left( \mathcal{R}_n(\mathcal{F}_0) + \sqrt{\frac{\log n}{n}} \right).
\end{aligned}$$
(27)

The second term of the equation (24) is bounded as follows.

$$
\left| \mathbb{E}_0\left( \widehat{f}_{\mathbf{T}_0}(\mathbf{T}_0(\mathbf{X}),0) \right) - \mathbb{E}_1\left( \widehat{f}_{\mathbf{T}_0}(\mathbf{X},1) \right) \right| \overset{\text{(C4)}}{\leq} C \sup_{h \in \mathcal{H}_\gamma} \left| \int h \circ \mathbf{T}_0(\mathbf{X}) d\mathcal{P}_0(\mathbf{X}) - \int h(\mathbf{X}) d\mathcal{P}_1(\mathbf{X}) \right|
$$

$$
= C d_{\mathcal{H}_\gamma}(\mathbf{T}_{0\#}\mathcal{P}_0, \mathcal{P}_1)
$$

$$
\overset{\text{by Lemma} B.8}{\leq} \mathcal{O}(n^{-\frac{\gamma}{2\gamma+d}}).
$$

$$(28)$$

with probability at least $1 - 1/n$.

Applying the upper bounds of (27) and (28) to (24), we conclude $\Delta\overline{\mathrm{DP}}(\widehat{f}_{\mathbf{T}_0}) \leq \delta + \mathcal{O}(\mathcal{R}_n(\mathcal{F}_0) + n^{-\frac{\gamma}{2\gamma+d}})$ up to a logarithm factor with probability at least $1 - 1/n$.

By using exactly the same arguments, we can derive the bound $\Delta\overline{\mathrm{DP}}(\widehat{f}_{\mathbf{T}_1}) \leq \delta + \mathcal{O}(\mathcal{R}_n(\mathcal{F}_1) + n^{-\frac{\gamma}{2\gamma+d}})$ up to a logarithm factor with probability at least $1 - 1/n$.

Thus, we have the desired result that $\Delta\overline{\mathrm{DP}}(\widehat{f}_{\mathbf{T}_s}) \leq \delta + \mathcal{O}(\mathcal{R}_n(\mathcal{F}_s) + n^{-\frac{\gamma}{2\gamma+d}})$ up to a logarithm factor with probability at least $1 - 1/n$.

Note that if we choose $\epsilon_n = \mathcal{O}(n^{-\frac{\gamma}{2\gamma+d}})$ up to a logarithmic factor, then $\widehat{\mathbf{T}}_s$ satisfies $d_{\mathcal{H}_n^{\mathrm{DNN}}}(\widehat{\mathbf{T}}_{s\#}\mathcal{P}_{n,s}, \mathcal{P}_{n,s'}) \leq \mathcal{O}(n^{-\frac{\gamma}{2\gamma+d}})$ up to a logarithmic factor by the definition of $\widehat{\mathbf{T}}_s$. Thus, we conclude that

$$
\Delta\overline{\mathrm{DP}}(\widehat{f}_s^{\mathrm{FTM}}) \leq \delta + \mathcal{O}(\mathcal{R}_n(\mathcal{F}_s) + n^{-\frac{\gamma}{2\gamma+d}})
$$

up to a logarithmic factor with probability at least $1 - 1/n$. Finally, Lemmas B.6 and B.7 ensure the existence of $\widehat{\mathbf{T}}_s$ satisfying $d_{\mathcal{H}_n^{\mathrm{DNN}}}(\widehat{\mathbf{T}}_{s\#}\mathcal{P}_{n,s}, \mathcal{P}_{n,s'}) \leq \mathcal{O}(n^{-\frac{\gamma}{2\gamma+d}})$ up to a logarithmic factor, which completes the proof. $\square$

## C  FURTHER DISCUSSIONS

### C.1  PRELIMINARIES ON THE OPTIMAL TRANSPORT THEORY

The Optimal Transport (OT) theory provides an approach for geometric comparison between two probability measures. The OT map is the *optimal* choice among all transport maps from a source distribution $\mathcal{P}$ to a target distribution $\mathcal{Q}$. In this context, *optimal* refers to minimizing a given *transport cost*, such as $L_p$ distance in Euclidean space.

Initially formulated by Monge (1781), the OT problem is expressed as

$$\min_{\mathbf{T}:\mathbf{T}_{\#}\mathcal{P}=\mathcal{Q}} \mathbb{E}\left(c(\mathbf{T}(\mathbf{X}),\mathbf{X})\right)$$

for some given cost function $c : \mathcal{X} \times \mathcal{X} \to \mathbb{R}$. The semi-dual problem of Monge problem is to find the *potential function* $\phi$, which minimizes $\mathcal{S}(\phi) := \int \phi(\mathbf{X})d\mathcal{P}(\mathbf{X}) + \int \tilde{\phi}(\mathbf{Y})d\mathcal{Q}(\mathbf{Y})$ where $\tilde{\phi}(\cdot) := \sup_{\mathbf{x}\in\mathbb{R}^d}\langle\mathbf{x},\cdot\rangle - \phi(\mathbf{x})$ is the conjugate of $\phi$. The solution $\phi^{\star} = \arg\min_{\phi\in L^1(P)}\mathcal{S}(\phi)$, is the anti-derivative of the OT map.

When the given source distribution $\mathcal{P}$ is absolutely continuous, a unique solution exists (Brenier's theorem (McCann, 1995; Villani & Society, 2003)). Moreover, if the densities of $\mathcal{P}$ and $\mathcal{Q}$ are smooth and satisfy the strong density assumption (i.e., with strict lower and upper bounds), then the unique OT map is smooth (Caffarelli's theorem (Caffarelli, 1996b)).

Unfortunately, the existence or uniqueness of the solution to the Monge problem, which is hard to solve due to its non-linearity, is not guaranteed when the source and target distributions are discrete and have different numbers of support (Villani, 2008). Kantorovich relaxed the Monge problem by seeking the optimal coupling between two distributions rather than the map with functional form (Kantorovich, 2006). The optimization objective is $\inf_{\pi\in\Pi(\mathcal{P},\mathcal{Q})}\mathbb{E}C(\mathbf{X},\mathbf{Y})$ where $\Pi(\mathcal{P},\mathcal{Q})$ is the set of all joint measures of $\mathcal{P}$ and $\mathcal{Q}$. This approach enables finding the OT map between two discrete measures.

Beyond theoretical minimax estimation of the OT map (Deb et al., 2021; Hütter & Rigollet, 2021; Seguy et al., 2018; Yang & Uhler, 2019), various computationally feasible estimators have been developed (Cuturi, 2013; Genevay et al., 2016), and applied to various tasks such as domain adaptation (Damodaran et al., 2018; Forrow et al., 2019), computer vision (Su et al., 2015; Li et al., 2015; Salimans et al., 2018), and economics (Galichon, 2016; Chiappori et al., 2010), to name a few.

Regularized OT maps (Cuturi, 2013; Genevay et al., 2016), which are similar to the relaxed OT map, have been proposed to calculate the OT map efficiently. The advantages of the relaxed OT map over the (regularized) OT map are: (1) computationally simpler since standard gradient descent algorithms can be used to learn a good relaxed OT map, (2) provides a functional form of the transport map, ensuring that exactly the same individuals in a sensitive group are always matched with the same individual in the opposite sensitive group, (3) theoretical studies for group fairness can be done relatively easily, and (4) has empirical evidence supporting that the relaxed OT map is more effective than the (regularized) OT map as the matching function in FTM algorithm (see Section 4.4).

### C.2 SUBSET FAIRNESS

**Definition** The mathematical definition of the subset fairness would be as follows: Let $\mathcal{C}$ be a given collection of subsets of $\mathcal{X}$. Then, the measure of the subset unfairness with respect to $\mathcal{C}$ is defined as

$$\Delta\mathrm{DP}_{\mathcal{C}}(f) = \sup_{A \in \mathcal{C}} |\mathbb{E}(\phi \circ f(\mathbf{X}, 0)|S = 0, \mathbf{X} \in A) - \mathbb{E}(\phi \circ f(\mathbf{X}, 1)|S = 1, \mathbf{X} \in A)|$$

for a given function $\phi : \mathbb{R} \to \mathbb{R}$. When $\mathcal{C} = \{\mathcal{X}\}$ and $\phi(z) = \mathbb{I}(z > 0.5)$, it becomes the standard $\Delta\mathrm{DP}$, and when $\mathcal{C} = \{\mathcal{X}\}$ and $\phi(z) = z$, then it becomes $\Delta\overline{\mathrm{DP}}$.

Note that $\Delta\mathrm{DP}_{\mathcal{C}}(f)$ measures how much $f$ is unfair on all subsets in $\mathcal{C}$. If $\mathcal{C}$ is known a priori, we can try to find an accurate model $f$ that minimizes $\Delta\mathrm{DP}_{\mathcal{C}}(f)$. This is similar to the problem of the subgroup fairness of Kearns et al. (2018), which considers groups defined by multiple sensitive jointly.

The problem we consider is different in the sense that we do not know $\mathcal{C}$ in advance. The goal of our problem is to improve the subset fairness among group-fair models. Note that we cannot find a model that minimizes $\Delta\mathrm{DP}_{\mathcal{C}}(f)$ because we do not observe $\mathcal{C}$ in the training phase. However, we can try to select one among various fair learning algorithms that are better with respect to the subset fairness, and FTM is such an algorithm.

**FTM improves subset fairness** In Section 3, we explained that FTM can improve the subset fairness. The following Theorem 3.1 and its proof provide rigorous mathematical evidence. Let $\phi(z) = z$ and consider $\Delta\overline{\mathrm{DP}}_{\mathcal{C}}(f) = \sup_{A \in \mathcal{C}} |\mathbb{E}(f(\mathbf{X}, 0)|S = 0, \mathbf{X} \in A) - \mathbb{E}(f(\mathbf{X}, 1)|S = 1, \mathbf{X} \in A)|$.

**Theorem 3.1** Suppose $\mathcal{F}$ is the collection of $L$-Lipschitz functions. Then, for all $f \in \mathcal{F}^{\Delta\mathrm{MDP}}(\delta)$, we have $\Delta\overline{\mathrm{DP}}_{\mathcal{C}}(f) \leq L\left(\min_s \left(\mathbb{E}_s\|\mathbf{X} - \mathbf{T}_s^f(\mathbf{X})\|^2\right)^{1/2} + U_1\right) + U_2\delta$, where $U_1, U_2 > 0$ are constants only depending on $\mathcal{C}$ and $\mathcal{P}_s, s = 0, 1$.

*Proof.* We write $\mathbf{T}_s = \mathbf{T}_s^f$ for notational simplicity. For any $A \subset \mathcal{C}$,

$$\begin{aligned}
&|\mathbb{E}(f(\mathbf{X}, 0)|S = 0, \mathbf{X} \in A) - \mathbb{E}(f(\mathbf{X}, 1)|S = 1, \mathbf{X} \in A)| \\
&\leq |\mathbb{E}(f(\mathbf{X}, 0)|S = 1, \mathbf{X} \in A) - \mathbb{E}(f(\mathbf{T}_1(\mathbf{X}), 0)|S = 1, \mathbf{X} \in A)| \\
&+ |\mathbb{E}(f(\mathbf{T}_1(\mathbf{X}), 0)|S = 1, \mathbf{X} \in A) - \mathbb{E}(f(\mathbf{X}, 1)|S = 1, \mathbf{X} \in A)| \\
&+ |\mathbb{E}(f(\mathbf{X}, 0)|S = 0, \mathbf{X} \in A) - \mathbb{E}(f(\mathbf{X}, 0)|S = 1, \mathbf{X} \in A)|.
\end{aligned} \tag{29}$$

By (C1), the first term is bounded by $L\mathbb{E}_1\|\mathbf{X} - \mathbf{T}_1(\mathbf{X})\|$, which is also bounded by $L\left(\mathbb{E}_1\|\mathbf{X} - \mathbf{T}_1(\mathbf{X})\|^2\right)^{1/2}$.

The second term is bounded by $\delta$ up to a constant due to the fairness constraint in the FTM algorithm. To be more specific, let $\mathcal{P}_s$ denote the distribution of $\mathbf{X}|S = s$. Then, we have

$$\begin{aligned}
&|\mathbb{E}(f(\mathbf{T}_1(\mathbf{X}), 0)|S = 1, \mathbf{X} \in A) - \mathbb{E}(f(\mathbf{X}, 1)|S = 1, \mathbf{X} \in A)| \\
&= \left| \frac{\int f(\mathbf{T}_1(\mathbf{X}), 0)\mathbb{I}(\mathbf{X} \in A)d\mathcal{P}_1(\mathbf{X})}{\int \mathbb{I}(\mathbf{X} \in A)d\mathcal{P}_1(\mathbf{X})} - \frac{\int f(\mathbf{X}, 1)\mathbb{I}(\mathbf{X} \in A)d\mathcal{P}_1(\mathbf{X})}{\int \mathbb{I}(\mathbf{X} \in A)d\mathcal{P}_1(\mathbf{X})} \right| \\
&\leq \frac{1}{\int \mathbb{I}(\mathbf{X} \in A)d\mathcal{P}_1(\mathbf{X})} \int_{\mathbf{X} \in A} |f(\mathbf{T}_1(\mathbf{X}), 0) - f(\mathbf{X}, 1)|d\mathcal{P}_1(\mathbf{X}) \\
&\leq \frac{1}{\int \mathbb{I}(\mathbf{X} \in A)d\mathcal{P}_1(\mathbf{X})} \int_{\mathbf{X} \in \mathcal{X}} |f(\mathbf{T}_1(\mathbf{X}), 0) - f(\mathbf{X}, 1)|d\mathcal{P}_1(\mathbf{X}) \\
&= U_2'(A, \mathcal{P}_1) \times \mathbb{E}_1|f(\mathbf{T}_1(\mathbf{X}), 0) - f(\mathbf{X}, 1)| \\
&\leq U_2'(A, \mathcal{P}_1) \times \delta
\end{aligned} \tag{30}$$

where $U_2'(A, \mathcal{P}_1) = 1/\int \mathbb{I}(\mathbf{X} \in A)d\mathcal{P}_1(\mathbf{X}) = 1/\mathcal{P}(\mathbf{X} \in A|S = 1)$ is a constant only depending on $\mathcal{P}_1$ and $A$.

The third term $|\mathbb{E}(f(\mathbf{X}, 0)|S = 0, \mathbf{X} \in A) - \mathbb{E}(f(\mathbf{X}, 0)|S = 1, \mathbf{X} \in A)|$ is not controllable by either the matching function or the fairness constraint but depends on the given distributions and $A$. Let

$\text{diam}(A) := \sup_{\mathbf{x}, \mathbf{y} \in A} \|\mathbf{x} - \mathbf{y}\|_2$ be the diameter of a given bounded set $A$. Then, the third term is bounded by $L\text{diam}(A)$ :

$$
\begin{aligned}
|\mathbb{E}(f(\mathbf{X}, 0)|S = 0, \mathbf{X} \in A) - \mathbb{E}(f(\mathbf{X}, 1)|S = 1, \mathbf{X} \in A)| &\leq \max_{\mathbf{x} \in A} f(\mathbf{x}, 0) - \min_{\mathbf{x} \in A} f(\mathbf{x}, 0) \\
&= f(\mathbf{x}_{\max}, 0) - f(\mathbf{x}_{\min}, 0) \\
&\leq L\|\mathbf{x}_{\max} - \mathbf{x}_{\min}\|_2 \\
&\leq L\text{diam}(A)
\end{aligned}
\tag{31}
$$

where $\mathbf{x}_{\max} = \arg\max_{\mathbf{x} \in A} f(\mathbf{x}, 0)$ and $\mathbf{x}_{\min} = \arg\min_{\mathbf{x} \in A} f(\mathbf{x}, 0)$.

For any $A$, we have

$$
\begin{aligned}
&|\mathbb{E}(f(\mathbf{X}, 0)|S = 0, \mathbf{X} \in A) - \mathbb{E}(f(\mathbf{X}, 1)|S = 1, \mathbf{X} \in A)| \\
&\leq L\left((\mathbb{E}_1\|\mathbf{X} - \mathbf{T}_1(\mathbf{X})\|^2)^{1/2} + \text{diam}(A)\right) + U_2'(A, 1)\delta.
\end{aligned}
\tag{32}
$$

By letting $U_1 := \sup_{A \in \mathcal{C}} \text{diam}(A)$ and $U_2(1) := \sup_{A \in \mathcal{C}} U_2'(A, \mathcal{P}_1)$, we conclude

$$
\begin{aligned}
&\sup_{A \in \mathcal{C}} |\mathbb{E}(f(\mathbf{X}, 0)|S = 0, \mathbf{X} \in A) - \mathbb{E}(f(\mathbf{X}, 1)|S = 1, \mathbf{X} \in A)| \\
&\leq L\left((\mathbb{E}_1\|\mathbf{X} - \mathbf{T}_1(\mathbf{X})\|^2)^{1/2} + U_1\right) + U_2(1)\delta.
\end{aligned}
\tag{33}
$$

We can similarly derive

$$
\begin{aligned}
&\sup_{A \in \mathcal{C}} |\mathbb{E}(f(\mathbf{X}, 0)|S = 0, \mathbf{X} \in A) - \mathbb{E}(f(\mathbf{X}, 1)|S = 1, \mathbf{X} \in A)| \\
&\leq L\left((\mathbb{E}_0\|\mathbf{X} - \mathbf{T}_0(\mathbf{X})\|^2)^{1/2} + U_1\right) + U_2(0)\delta.
\end{aligned}
\tag{34}
$$

Letting $U_2 := \max(U_2(0), U_2(1))$ completes the proof. $\square$

The first term of RHS, $L\mathbb{E}_s\|\mathbf{X} - \mathbf{T}_s(\mathbf{X})\|^2$, implies that using the (relaxed) OT map helps improve the subset fairness. The uncontrollable constant $U_1$, can be small for certain subsets. For example, for disjoint sets $C_1, \cdots, C_K$ of $\mathcal{C}$, suppose that $\mathcal{P}_s$ is a mixture of uniform distribution given as $\mathcal{P}_s(\cdot) = \sum_{k=1}^{K} p_{sk}\mathbb{I}(\cdot \in C_k)$ with $p_{sk} \geq 0$ and $\sum_{k=1}^{K} p_{sk} = 1$ (e.g., the histogram). Then, $U_1$ becomes zero for all $C$ in $\mathcal{C} = \{C_k, k = 1, \ldots, K\}$. The third term of RHS, $U_2\delta$ is small when $\delta$ is small.

## C.3 WITHIN-GROUP FAIRNESS

**FTM improves within-group fairness** In Section 3, we explained that FTM can improve the within-group fairness. The following Theorem 3.2 and its proof provide a theoretical justification of this assertion.

**Theorem 3.2** Let $\mathbf{x}^{(1)}$ and $\mathbf{x}^{(2)}$ be two individuals such that $(\mathbf{x}^{(1)}, \mathbf{x}^{(2)}) \in \mathcal{X}^2(f^\star)$. Suppose that $f^\star$ is $L$-Lipshitz. Then, for $s = 0, 1$, we have

$$f_{\mathbf{T}}(\mathbf{x}^{(1)}, s) - f_{\mathbf{T}}(\mathbf{x}^{(2)}, s) > \frac{M_s}{2} - \frac{L}{2} \max \left( \sum_{i=1,2} \|\mathbf{x}^{(i)} - \mathbf{T}_s(\mathbf{x}^{(i)})\|, \sum_{i=1,2} \|\mathbf{x}^{(i)} - \mathbf{T}_{s'}^{-1}(\mathbf{x}^{(i)})\| \right),$$

for $M_s = M_s(\mathbf{x}^{(1)}, \mathbf{x}^{(2)}) := f^\star(\mathbf{x}^{(1)}, s) - f^\star(\mathbf{x}^{(2)}, s) > 0$.

*Proof.* Assume $\min_s \mathbb{E}_s |f(\mathbf{X}, s) - f(\mathbf{T}_s(\mathbf{X}), s')| = \mathbb{E}_0 |f(\mathbf{X}, 0) - f(\mathbf{T}_0(\mathbf{X}), 1)|$.

We can similarly prove for the case when $\min_s \mathbb{E}_s |f(\mathbf{X}, s) - f(\mathbf{T}_s(\mathbf{X}), s')| = \mathbb{E}_1 |f(\mathbf{X}, 1) - f(\mathbf{T}_1(\mathbf{X}), 0)|$ to obtain the desired result.

**($s = 0$ case)**

Fix $\mathbf{x}^{(1)}, \mathbf{x}^{(2)} \in \mathcal{X}_0 = \mathcal{X}$ such that $(\mathbf{x}^{(1)}, \mathbf{x}^{(2)}) \in \mathcal{X}^2(f^\star)$. Since $f_{\mathbf{T}}$ is the minimizer of $\mathbb{E}(Y - f(\mathbf{X}, S))^2$ on $\{f \in \mathcal{F} : \min_s \mathbb{E}_s |f(\mathbf{X}, s) - f(\mathbf{T}_s(\mathbf{X}), s')| = 0\}$ and $\mathbb{E}_0 |f(\mathbf{X}, 0) - f(\mathbf{T}_0(\mathbf{X}), 1)| = 0$, we have

$$\left( f_{\mathbf{T}}(\mathbf{x}^{(1)}, 0), f_{\mathbf{T}}(\mathbf{T}_0(\mathbf{x}^{(1)}), 1) \right) = \underset{(z_1, z_2): z_1 = z_2}{\arg\min} \ (z_1 - f^\star(\mathbf{x}^{(1)}, 0))^2 + (z_2 - f^\star(\mathbf{T}_0(\mathbf{x}^{(1)}), 1))^2 \quad (35)$$

and

$$\left( f_{\mathbf{T}}(\mathbf{x}^{(2)}, 0), f_{\mathbf{T}}(\mathbf{T}_0(\mathbf{x}^{(2)}), 1) \right) = \underset{(z_1, z_2): z_1 = z_2}{\arg\min} \ (z_1 - f^\star(\mathbf{x}^{(2)}, 0))^2 + (z_2 - f^\star(\mathbf{T}_0(\mathbf{x}^{(2)}), 1))^2. \quad (36)$$

Then,

$$f_{\mathbf{T}}(\mathbf{x}^{(1)}, 0) = \frac{f^\star(\mathbf{x}^{(1)}, 0) + f^\star(\mathbf{T}_0(\mathbf{x}^{(1)}), 1)}{2}$$

and

$$f_{\mathbf{T}}(\mathbf{x}^{(2)}, 0) = \frac{f^\star(\mathbf{x}^{(2)}, 0) + f^\star(\mathbf{T}_0(\mathbf{x}^{(2)}), 1)}{2}.$$

Therefore, we have

$$
\begin{aligned}
f_{\mathbf{T}}(\mathbf{x}^{(1)}, 0) - f_{\mathbf{T}}(\mathbf{x}^{(2)}, 0) &= \frac{f^\star(\mathbf{x}^{(1)}, 0) + f^\star(\mathbf{T}_0(\mathbf{x}^{(1)}), 1)}{2} - \frac{f^\star(\mathbf{x}^{(2)}, 0) + f^\star(\mathbf{T}_0(\mathbf{x}^{(2)}), 1)}{2} \\
&\geq \frac{f^\star(\mathbf{x}^{(1)}, 0) + f^\star(\mathbf{x}^{(1)}, 1)}{2} - \frac{f^\star(\mathbf{x}^{(2)}, 0) + f^\star(\mathbf{x}^{(2)}, 1)}{2} \\
&\quad - \left| \frac{f^\star(\mathbf{T}_0(\mathbf{x}^{(1)}), 1) - f^\star(\mathbf{x}^{(1)}, 1)}{2} \right| - \left| \frac{f^\star(\mathbf{T}_0(\mathbf{x}^{(2)}), 1) - f^\star(\mathbf{x}^{(2)}, 1)}{2} \right| \\
&> \frac{M_0}{2} - \frac{L}{2} (\|\mathbf{x}^{(1)} - \mathbf{T}_0(\mathbf{x}^{(1)})\| + \|\mathbf{x}^{(2)} - \mathbf{T}_0(\mathbf{x}^{(2)})\|)
\end{aligned}
$$

where $M_0 = f^\star(\mathbf{x}^{(1)}, 0) - f^\star(\mathbf{x}^{(2)}, 0) > 0$. Hence, we have the assertion.

**($s = 1$ case)**

Fix $\mathbf{x}^{(1)}, \mathbf{x}^{(2)} \in \mathcal{X}_1 = \mathcal{X}$ such that $(\mathbf{x}^{(1)}, \mathbf{x}^{(2)}) \in \mathcal{X}^2(f^\star)$. Note that $\mathbf{T}_0^{-1}(\mathbf{x}^{(1)}), \mathbf{T}_0^{-1}(\mathbf{x}^{(2)}) \in \mathcal{X}_0 = \mathcal{X}$. Then, similar to the $s = 0$ case, since $f_{\mathbf{T}}$ is the minimizer of $\mathbb{E}(Y - f(\mathbf{X}, S))^2$ on $\{f \in \mathcal{F} : \min_s \mathbb{E}_s |f(\mathbf{X}, s) - f(\mathbf{T}_s(\mathbf{X}), s')| = 0\}$ and $\mathbb{E}_0 |f(\mathbf{X}, 0) - f(\mathbf{T}_0(\mathbf{X}), 1)| = 0$, we have

$$
\begin{aligned}
&\left( f_{\mathbf{T}}(\mathbf{T}_0^{-1}(\mathbf{x}^{(1)}), 0), f_{\mathbf{T}}(\mathbf{T}_0 \circ \mathbf{T}_0^{-1}(\mathbf{x}^{(1)}), 1) \right) \\
&= \underset{(z_1, z_2): z_1 = z_2}{\arg\min} \ (z_1 - f^\star(\mathbf{T}_0^{-1}(\mathbf{x}^{(1)}), 0))^2 + (z_2 - f^\star(\mathbf{T}_0 \circ \mathbf{T}_0^{-1}(\mathbf{x}^{(1)}), 1))^2
\end{aligned}
\quad (37)
$$

and

$$
\begin{aligned}
&\left(f_{\mathbf{T}}(\mathbf{T}_0^{-1}(\mathbf{x}^{(2)}), 0), f_{\mathbf{T}}(\mathbf{T}_0 \circ \mathbf{T}_0^{-1}(\mathbf{x}^{(2)}), 1)\right) \\
&= \underset{(z_1, z_2): z_1 = z_2}{\arg\min} \ (z_1 - f^{\star}(\mathbf{T}_0^{-1}(\mathbf{x}^{(2)}), 0))^2 + (z_2 - f^{\star}(\mathbf{T}_0 \circ \mathbf{T}_0^{-1}(\mathbf{x}^{(2)}), 1))^2.
\end{aligned}
\tag{38}
$$

Then,

$$
f_{\mathbf{T}}(\mathbf{T}_0 \circ \mathbf{T}_0^{-1}(\mathbf{x}^{(1)}), 1) = \frac{f^{\star}(\mathbf{T}_0^{-1}(\mathbf{x}^{(1)}), 0) + f^{\star}(\mathbf{T}_0 \circ \mathbf{T}_0^{-1}(\mathbf{x}^{(1)}), 1)}{2}
$$

and

$$
f_{\mathbf{T}}(\mathbf{T}_0 \circ \mathbf{T}_0^{-1}(\mathbf{x}^{(2)}), 1) = \frac{f^{\star}(\mathbf{T}_0^{-1}(\mathbf{x}^{(2)}), 0) + f^{\star}(\mathbf{T}_0 \circ \mathbf{T}_0^{-1}(\mathbf{x}^{(2)}), 1)}{2}.
$$

Since $\mathbf{T}_0 \circ \mathbf{T}_0^{-1}$ is the identity map, we rewrite the above equations as:

$$
f_{\mathbf{T}}(\mathbf{x}^{(1)}, 1) = \frac{f^{\star}(\mathbf{T}_0^{-1}(\mathbf{x}^{(1)}), 0) + f^{\star}(\mathbf{x}^{(1)}, 1)}{2}
$$

and

$$
f_{\mathbf{T}}(\mathbf{x}^{(2)}, 1) = \frac{f^{\star}(\mathbf{T}_0^{-1}(\mathbf{x}^{(2)}), 0) + f^{\star}(\mathbf{x}^{(2)}, 1)}{2}.
$$

Therefore, we have

$$
\begin{aligned}
f_{\mathbf{T}}(\mathbf{x}^{(1)}, 1) - f_{\mathbf{T}}(\mathbf{x}^{(2)}, 1) &= \frac{f^{\star}(\mathbf{x}^{(1)}, 1) + f^{\star}(\mathbf{T}_0^{-1}(\mathbf{x}^{(1)}), 0)}{2} - \frac{f^{\star}(\mathbf{x}^{(2)}, 1) + f^{\star}(\mathbf{T}_0^{-1}(\mathbf{x}^{(2)}), 0)}{2} \\
&\geq \frac{f^{\star}(\mathbf{x}^{(1)}, 1) + f^{\star}(\mathbf{x}^{(1)}, 0)}{2} - \frac{f^{\star}(\mathbf{x}^{(2)}, 1) + f^{\star}(\mathbf{x}^{(2)}, 0)}{2} \\
&\quad - \left| \frac{f^{\star}(\mathbf{T}_0^{-1}(\mathbf{x}^{(1)}), 0) - f^{\star}(\mathbf{x}^{(1)}, 0)}{2} \right| - \left| \frac{f^{\star}(\mathbf{T}_0^{-1}(\mathbf{x}^{(2)}), 0) - f^{\star}(\mathbf{x}^{(2)}, 0)}{2} \right| \\
&> \frac{M_1}{2} - \frac{L}{2}(\|\mathbf{x}^{(1)} - \mathbf{T}_0^{-1}(\mathbf{x}^{(1)})\| + \|\mathbf{x}^{(2)} - \mathbf{T}_0^{-1}(\mathbf{x}^{(2)})\|)
\end{aligned}
$$

where $M_1 = f^{\star}(\mathbf{x}^{(1)}, 1) - f^{\star}(\mathbf{x}^{(2)}, 1) > 0$. Hence, we have the assertion.

$\square$

## C.4 REMARK RELATED TO PROPOSITION 3.3

**How to use $\mathcal{F}^{\Delta\mathbf{MDP}}(\delta; \{\mathbf{T}^\star_{0,\epsilon}\}, \{\mathbf{T}^\star_{1,\epsilon}\})$ when $\gamma$ is large.** We can use $\mathcal{F}^{\Delta\mathbf{MDP}}(\delta; \{\mathbf{T}^\star_{0,\epsilon}\}, \{\mathbf{T}^\star_{1,\epsilon}\})$ to eliminate undesirable group-fair models. We first choose a target group fairness measure $\Delta$ (e.g., widely-used measures such as $\Delta\mathrm{DP}, \Delta\overline{\mathrm{DP}}$) and set the fairness level $\alpha \geq 0$. Then, we search for accurate models on $\mathcal{F}^\Delta(\alpha) \cap \mathcal{F}^{\Delta\mathbf{MDP}}(\delta; \{\mathbf{T}^\star_{0,\epsilon}\}, \{\mathbf{T}^\star_{1,\epsilon}\})$ for $\delta > \alpha$. Here, $\alpha$ controls the group fairness while $\delta$ controls other fairness such as subset/within-group fairness.

## C.5 COMPARISON WITH FRL

We here compare FTM and FRL with details as Section 3.4 discussed briefly.

Fair Representation Learning (FRL) algorithm aims at searching a fair representation space (Zemel et al., 2013) in the sense that the distributions of the encoded representation vector on each sensitive group are similar. After learning the fair representation, FRL constructs a fair model by applying a supervised learning algorithm to the fair representation space. Initiated by Edwards & Storkey (2016), various FRL algorithms have been developed (Madras et al., 2018; Zhang et al., 2018) motivated by the adversarial-learning technique used in GAN (Goodfellow et al., 2014).

The matching function $\mathbf{T}_s$ in FTM can be interpreted as a fair representation encoder, where the representation space is $\mathcal{X}_{s'}$. Conversely, we can construct a matching function from given fair representation encoders $\mathbf{E}_s, s \in \{0,1\}$ by letting $\mathbf{T}_s = \mathbf{E}_{s'}^{-1} \circ \mathbf{E}_s$, provided that $\mathbf{E}_{s'}^{-1}$ exists. Moreover, if $\mathbf{E}_s$ is a barycentric mapping, which lies on the path of the OT map (Villani, 2008; Santambrogio, 2015), then $\mathbf{T}_s = \mathbf{E}_{s'}^{-1} \circ \mathbf{E}_s$ becomes the OT map. Thus, FTM is a variant of FRL that uses barycentric mapping as the representation encoder. On the other hand, there is a difference in how they achieve a given level of fairness: FTM sets (or tunes) $\delta$ to control fairness under $\epsilon \approx 0$, whereas FRL sets $\epsilon$ to control fairness under $\delta = 0$. There is a clear advantage of FTM over FRL: while the prediction model of FRL is a map from the representation space to the output space, the prediction model of FTM is a map from the input space to the output space. Thus, FTM offers more flexibility in model selection. See Section 4.3 for more discussion.

In fact, Gordaliza et al. (2019) proposed to use the barycentric mapping explicitly for FRL. FTM has several advantages compared to the algorithm of Gordaliza et al. (2019): (1) It obtains the barycentric mapping in a coupling form using the training data only, requiring recalculation of the coupling during inference. In contrast, FTM obtains the matching function in a functional form, resulting in more direct and convenient inference. (2) FTM is empirically superior to Gordaliza et al. (2019). That is, we conduct a numerical comparison with 'FRL-bary' in the ablation study, which is a variant of Gordaliza et al. (2019), using the relaxed OT (barycentric) map. FRL-bary is alternatively considered because it cannot infer a single test datum. See Section 4.4 and Figure 8.

## C.6 EXTENSION OF FTM TO EQUAL OPPORTUNITY

MDP and FTM algorithm formulated in Section 3.3 is easily extended to equal opportunity by replacing the definition of groups. That is, once we obtain (relaxed) OT maps between $\mathcal{P}_{\mathbf{X}|S=0,Y=1}$ and $\mathcal{P}_{\mathbf{X}|S=1,Y=1}$ at the first step, then we can similarly train FTM classifier using the (relaxed) OT maps as matching functions in the second step. Furthermore, all the theoretical results shown in this paper would hold clearly, which guarantees that the trained classifier satisfies equal opportunity at a certain level. However, the label information $Y$ is required to train the matching function for equal opportunity while it is not for demographic parity.

## C.7 COMPARISON WITH THE INDIVIDUAL FAIRNESS

Since the main idea of the MDP and FTM is to treat two individuals from different sensitive groups similarly, it may seem that there is a connection with the concept of individual fairness. However, a clear distinction between the two is that FTM aims to treat two individuals similarly from different sensitive groups only in order to achieve group fairness, while the individual fairness requires to treat similar individuals similarly regardless of sensitive attribute (even it is unknown). That is, similar individuals in FTM can be dissimilar in view of individual fairness, especially when the two sensitive groups are significantly different.

Even though similar individuals in FTM is different from those in view of individual fairness, FTM empirically improves individual fairness compared to other strong group-fair models (e.g., FRL methods). The empirical results are reported in Table 3. For the measure of individual fairness, we use the consistency (Con) from Yurochkin et al. (2020); Yurochkin & Sun (2021), which is the ratio of consistently predicted labels when we only flip the sensitive variable among the input variables.

Table 3: **Individual fairness**: comparison of FTM and FRL algorithms with respect to individual fairness. The bold faces are the best consistency, and underlined ones are the second placers. Given a fixed $\Delta\overline{\mathrm{DP}}$, FTM achieves higher levels of the individual fairness than baselines in most cases.

| Dataset ($\Delta\overline{\mathrm{DP}}$) | Measure | Unfair | AE-MMD | FRL LAFTR | sIPM-LFR | FTM |
|---|---|---|---|---|---|---|
| *Adult* ($\approx 0.060$) | Acc | 0.845 | 0.787 | 0.797 | 0.801 | 0.820 |
| | Con | 0.882 | 0.921 | 0.936 | 0.918 | **0.965** |
| *German* ($\approx 0.050$) | Acc | 0.743 | 0.729 | 0.721 | 0.721 | 0.738 |
| | Con | 0.957 | 0.960 | 0.955 | 0.940 | **0.963** |
| *COMPAS* ($\approx 0.120$) | Acc | 0.677 | 0.576 | 0.629 | 0.630 | 0.661 |
| | Con | 0.899 | 0.913 | 0.891 | 0.902 | **0.918** |
| *Dutch* ($\approx 0.030$) | Acc | 0.824 | 0.767 | 0.755 | 0.758 | 0.784 |
| | Con | 0.844 | 0.963 | 0.957 | 0.959 | **0.973** |
| *Law* ($\approx 0.040$) | Acc | 0.898 | 0.885 | 0.886 | 0.888 | 0.888 |
| | Con | 0.955 | 0.992 | 0.996 | **0.997** | 0.996 |

## C.8 COMPARISON WITH FLIPTEST

In this section, we provide a concrete comparison between FTM and FlipTest (Black et al., 2020), as shortly discussed in Section 3.4.

For a given prediction model $f$ and $s \in \{0, 1\}$, FlipTest first finds two sets of individuals whose predictions are flipped, as defined by $F^+(f; s) := \{i : s_i = s, \mathbb{I}(f(\mathbf{x}_i, s) > 0) > \mathbb{I}(f(\widehat{\mathbf{T}}_s(\mathbf{x}_i), s') > 0)\}$ and $F^-(f; s) := \{i : s_i = s, \mathbb{I}(f(\mathbf{x}_i, s) > 0) < \mathbb{I}(f(\widehat{\mathbf{T}}_s(\mathbf{x}_i), s') > 0)\}$, where $\widehat{\mathbf{T}}_s, s \in \{0, 1\}$ are the optimal transport map. Using these sets, FlipTest can measure the unfairness of $f$ as $|F^+(f; s)| - |F^-(f; s)|$.

The regularization term induced by FlipTest would be formulated as:

$$\left| \mathbb{E}_{n,s}\left(\mathbb{I}(f(\mathbf{X}, s) > 0)\right) - \mathbb{E}_{n,s}\left(\mathbb{I}(f(\widehat{\mathbf{T}}_s(\mathbf{X}), s') > 0)\right) \right|$$

or

$$\left| \mathbb{E}_{n,s}\left(f(\mathbf{X}, s)\right) - \mathbb{E}_{n,s}\left(f(\widehat{\mathbf{T}}_s(\mathbf{X}), s')\right) \right|,$$

which is completely different from our regularization term:

$$\mathrm{REG}_s(f) = \mathbb{E}_{n,s}|f(\mathbf{X}, s) - f(\widehat{\mathbf{T}}_s(\mathbf{X}), s')|.$$

The former is the difference of the expectations, while the later is the expectation of the (absolute) differences. This seemingly tiny difference would make big differences in many ways. For example, the measure of FlipTest would not imply the strong group fairness, while our regularization term does (i.e., Proposition 2.1). Moreover, it would not be clear whether the measure of FlipTest improves better subset/within-group fairness.

# D IMPLEMENTATION DETAILS

Detailed descriptions for the implementation of the experiments are provided.

## D.1 DATASETS

Table 4: The description of each real dataset: *Adult*, *German*, *COMPAS*, *Dutch*, and *Law*. $\mathbf{X}$ is the input vector, $S$ is the sensitive attribute, $Y$ is the target label information, and $d$ is the dimension of $\mathbf{X}$. (Train/Test) data sizes are the number of samples.

| Dataset | Variable | Description | Dataset | Variable | Description |
|---------|----------|-------------|---------|----------|-------------|
| *Adult* | $\mathbf{X}$ | Personal attributes | *German* | $\mathbf{X}$ | Personal attributes |
| | $S$ | Gender | | $S$ | Gender |
| | $Y$ | Outcome over $\$50k$ | | $Y$ | Credit score is good |
| | $d$ | 101 | | $d$ | 60 |
| | Train data size | 30,136 | | Train data size | 800 |
| | Test data size | 15,086 | | Test data size | 200 |
| *COMPAS* | $\mathbf{X}$ | Personal attributes | *Dutch* | $\mathbf{X}$ | Personal attributes |
| | $S$ | Race | | $S$ | Gender |
| | $Y$ | Recidivism within 2 years | | $Y$ | Occupation is high-level |
| | $d$ | 399 | | $d$ | 58 |
| | Train data size | 4,933 | | Train data size | 48,336 |
| | Test data size | 1,234 | | Test data size | 12,084 |
| *Law* | $\mathbf{X}$ | Personal attributes | | | |
| | $S$ | Race | | | |
| | $Y$ | Passing exam on the 1st try | | | |
| | $d$ | 19 | | | |
| | Train data size | 16,638 | | | |
| | Test data size | 4,160 | | | |

Table 4 provides the summaries of the five datasets. For the references for downloading, we provide the URLs as follows.

1. *Adult*: the Adult income dataset (Dua & Graff, 2017) (abbr. *Adult*) can be downloaded from the UCI repository[4].

2. *German*: the German credit dataset (Dua & Graff, 2017) (abbr. *German*) can downloaded from the UCI repository[5].

3. *COMPAS*: the COMPAS dataset (abbr. *COMPAS*) can downloaded from the official GitHub of propublica[6].

4. *Dutch*: the Dutch census dataset (abbr. *Dutch*) can downloaded from the public Github of (Quy et al., 2022) [7].

5. *Law*: the Law school dataset (abbr. *Law*) can downloaded from the public Github of (Quy et al., 2022) [8].

For *Adult*, *German*, and *COMPAS*, we follow the pre-processing of AIF360 (Bellamy et al., 2018)'s implementation [9], For the remaining two datasets (i.e., *Dutch* and *Law*), we follow the pre-processing of (Quy et al., 2022)'s Github[10]. Except for *Adult* dataset, whose test dataset is already split, we randomly select $20\%$ of the whole data as the test data. For all datasets, we split each training data into 8:2 splits randomly and used the latter one for validation to choose the best iteration step during training.

---

[4]https://archive.ics.uci.edu/ml/datasets/adult

[5]https://archive.ics.uci.edu/ml/datasets/statlog+(german+credit+data)

[6]https://github.com/propublica/compas-analysis/

[7]https://github.com/tailequy/fairness_dataset/tree/main/experiments/data/dutch.csv

[8]https://github.com/tailequy/fairness_dataset/tree/main/experiments/data/law_school_clean.csv

[9]https://aif360.readthedocs.io/en/stable/

[10]https://github.com/tailequy/fairness_dataset/tree/main/experiments/data/

### D.2 BASELINE METHODS

**FRL**   Recall that the unsupervised FRL algorithm obtains a fair autoencoder $\mathbf{D}_s \circ \mathbf{E}_s$ for $s = 0, 1$ by learning fair representation space via eliminating sensitive information.

1. AE-MMD: This method learns autoencoders with MMD regularization on the representation space. It can be interpreted as a variant of two existing methods: FPCA (Lee et al., 2022), which learns linear fair representations by PCA with MMD regularization, and MMD-B-Fair (Deka & Sutherland, 2023), which learns non-linear fair representations with MMD regularization in a supervised manner, respectively. Note that VFAE (Louizos et al., 2015) also uses MMD regularization but with VAE (Kingma & Welling, 2013).

2. LAFTR (Madras et al., 2018): This algorithm is an advanced version of ALFR (Edwards & Storkey, 2016), which learns fair representation using adversarial learning, in that the cross-entropy loss for the adversarial network is replaced by smooth $L_1$ loss.

3. sIPM-LFR (Kim et al., 2022): Similar to LAFTR, the adversarial network is to separate representations by the sensitive variable, whose architecture is just a combination of sigmoid function and linear projection so that the training becomes more simple and stable. That is, it uses an IPM, not a divergence as LAFTR and ALFR has done.

**In-processing**

1. Reg: This method is a simple regularizing approach that minimizes cross-entropy + $\lambda \Delta \overline{\mathrm{DP}}$ for some $\lambda > 0$. In (Chuang & Mroueh, 2021), they call this algorithm GapReg. This is also similar to the approach of (Donini et al., 2018) in the sense that the model is learned with a constraint having a given level of $\Delta \overline{\mathrm{DP}}$.

2. Adv (Zhang et al., 2018): This algorithm is an in-processing method that regularizes the model outputs with an adversarial network so that the adversarial network prevents the model outputs from predicting sensitive variables.

3. Fair-Mixup (Chuang & Mroueh, 2021): This algorithm is an in-processing method that regularizes the model output's path to achieve low levels of $\Delta \overline{\mathrm{DP}}$.

4. Reduction (Agarwal et al., 2018): This algorithm is an in-processing method that learns a fair classifier with the lowest empirical fairness level $\Delta \mathrm{DP}$.

### D.3 TRAINING AND INFERENCE PROTOCOL

To control $\epsilon$ and $\delta$, we alter the constraints in STEP 1 and 2 by adding regularizations as follows.

**(STEP 1)**   At the first step, for each $s = 0, 1$, we train the matching network $\widehat{\mathbf{T}}_s = \widehat{\mathbf{T}}_{n,s}$ as

$$\widehat{\mathbf{T}}_s := \underset{\mathbf{T} \in \mathcal{T}_n^{\mathrm{DNN}}}{\arg \min} \mathbb{E}_{n,s} \left( \|\mathbf{X} - \mathbf{T}(\mathbf{X})\|^2 \right) + \lambda_T d_{\mathcal{H}_n^{\mathrm{DNN}}}(\mathbf{T}_\# \mathcal{P}_{n,s}, \mathcal{P}_{n,s'}) \tag{39}$$

for a transporting hyper-parameter $\lambda_T > 0$ (practically, we set $\lambda_T = 100.0$ in all experiments). Note that $\lambda_T$ takes the role of $\epsilon$. To select the best matching function network during training (STEP 1), we select the best epoch when the loss on the validation data is the lowest. Once we select the best matching network, we learn FTM classifier.

**(STEP 2)**   For the second step, we train the (empirical) FTM classifier $\widehat{f}_s^{\mathrm{FTM}}$ as

$$\widehat{f}_s^{\mathrm{FTM}} := \underset{f \in \mathcal{F}}{\arg \min} \, \mathbb{E}_n \left( l(Y, f(\mathbf{X}, S)) \right) + \lambda_F \mathrm{REG}_s(f) \tag{40}$$

with a fair hyper-parameter $\lambda_F > 0$.

Note that $\lambda_F$ take the role $\delta$. To select the best classifier during training (STEP 2), we select the best iteration step when the Acc$-\Delta \overline{\mathrm{DP}}$ is the highest on the validation data following (Edwards & Storkey, 2016; Madras et al., 2018; Kim et al., 2022). After that, we choose the final classifier from $\{\widehat{f}_0^{\mathrm{FTM}}, \widehat{f}_1^{\mathrm{FTM}}\}$ which has better Acc$-\Delta \overline{\mathrm{DP}}$ than the other one on validation data.

After selecting the final classifier, we infer the model performance on test data. We repeat this procedure with five random model initial parameters five times, then report the mean result and standard errors over the five test results. We repeat STEP 2 by varying $\lambda_F$s in an appropriate range to draw Pareto-front lines.

## D.4 Hyperparameter selection and model architectures

**FTM**  The selected hyperparameters for FTM of each dataset are described in Table 5 in below.

Table 5: Training hyperparameters on each dataset.

| Dataset | | Adult | German | COMPAS | Dutch | Law |
|---|---|---|---|---|---|---|
| Batch size | | 1024 | 64 | 64 | 1024 | 1024 |
| Learning rates of $(\mathbf{T}_s, f)$ | | (1e-3, 1e-4) | (5e-3, 1e-3) | (5e-3, 1e-3) | (5e-3, 1e-3) | (5e-3, 1e-3) |
| Epochs of learning $\mathbf{T}_s$ | | 200 | 500 | 500 | 500 | 500 |
| Iterations of learning $f$ | | | | 500 | | |
| Evaluation frequency | | | | 50 | | |
| Optimizer | | | | Adam (Kingma & Ba, 2014) | | |

We set $\mathcal{F}, \mathcal{T}_s$ and $\mathcal{H}$ from {1-layer MLP, 2-layer MLP, 3-layer MLP}. Then, we found the best combination of the three architectures as described below. For the OT map function class $\mathcal{T}_s$, we consider two-layer MLP with LeakyReLU activation (slope = 0.2) and a Dropout layer after the first layer. All numbers of nodes at each layer are equal to the input dimension. For MMD, we compute the average of 5 MMDs induced by Gaussian kernel $k(\mathbf{x}, \mathbf{y}) := e^{-\|\mathbf{x}-\mathbf{y}\|^2/2\sigma^2}$ with $\sigma = 0.01, 0.1, 1.0, 10.0, 100.0$. For the classifier in Figure 4, we consider a single-layer MLP with ReLU activation. All numbers of nodes of each layer are equal to the input dimension. For the classifier in Figure 7, we consider the linear classifier.

When computing the transport cost, for each categorical input variable, we encode it to the one-hot vector divided by the number of the categories to make the influence of categorical input variables on the transport cost of the (relaxed) OT map as equal as possible.

**FRL**  For FRL, we basically follow the training protocol and hyperparameter selection as done in Madras et al. (2018); Kim et al. (2022). At this time only, let $\mathcal{T}_s$ be the encoder network class. For the FRL, we set $\mathcal{F}, \mathcal{T}_s$ and $\mathcal{H}$ from {1-layer MLP, 2-layer MLP, 3-layer MLP}. Then, we select the best combination of the three architectures: 1-layer MLP with ReLU activation for $\mathcal{F}$, 2-layer MLP with LeakyReLU activation (slope = 0.2) for $\mathcal{T}_s$, and 2-layer MLP with ReLU activation for $\mathcal{H}$ (for LAFTR). For AE-MMD, for which we do not use the discriminator network $\mathcal{H}$, we compute the average of 5 MMDs as done in FTM. Adam optimizer is used with learning rate from {0.0001, 0.001, 0.01}. To select the best classifier during training, we select the best iteration step when the Acc$-\Delta\overline{\mathrm{DP}}$ is the highest on the validation data following Edwards & Storkey (2016); Madras et al. (2018); Kim et al. (2022) as done for FTM.

**In-processing**  For in-processing methods, we use the same architecture as FTM and FRL methods used. That is, the prediction model network is chosen as a single-layer MLP with ReLU activation. Adam optimizer is used with learning rate from {0.0001, 0.001, 0.01}. To select the best classifier during training, we select the best iteration step when the Acc$-\Delta\overline{\mathrm{DP}}$ is the highest on the validation data following Edwards & Storkey (2016); Madras et al. (2018); Kim et al. (2022) as done for FTM.

# E   ADDITIONAL EXPERIMENTAL RESULTS

## E.1   COMPARISON WITH GROUP-FAIR ALGORITHMS (SECTION 4.2)

**Subset fairness**   We provide Figure 5 for the remaining four datasets with the subset fairness measure: subset $\Delta\overline{\mathrm{DP}}$. We found that FTM has a relatively smaller number of outliers than other baseline algorithms which implies that FTM attains higher subset fairness.

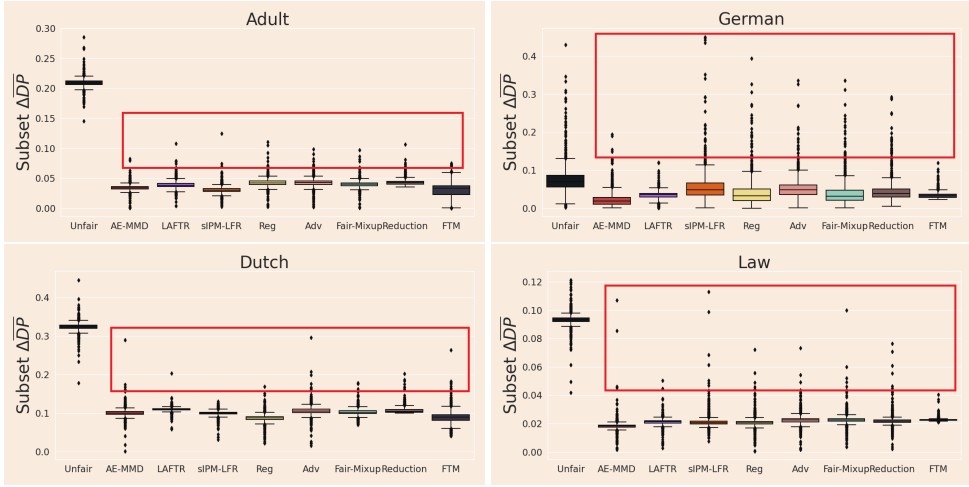

Figure 5: Boxplots of the levels of $\Delta\overline{\mathrm{DP}}$ on randomly generated subsets for remaining four datasets (*Adult*, *German*, *Dutch*, *Law*).

**Within-group fairness**   We provide Table 6, which is a copy of Table 1 with standard errors.

Table 6: A copy of Table 1 with standard errors.

| Dataset | | Adult | | German | | COMPAS | | Dutch | | Law | |
|---|---|---|---|---|---|---|---|---|---|---|---|
| $\Delta\overline{\mathrm{DP}}$ : Unfair → Fair | | $0.19 \to 0.10$ | | $0.09 \to 0.04$ | | $0.19 \to 0.10$ | | $0.34 \to 0.14$ | | $0.17 \to 0.07$ | |
| Sensitive attribute $S$ | | 0 | 1 | 0 | 1 | 0 | 1 | 0 | 1 | 0 | 1 |
| | AE-MMD | 0.771 | 0.872 | 0.651 | 0.779 | 0.436 | 0.463 | 0.825 | 0.929 | 0.790 | 0.585 |
| | s.e. | 0.038 | 0.026 | 0.052 | 0.018 | 0.022 | 0.023 | 0.002 | 0.003 | 0.037 | 0.019 |
| FRL | LAFTR | 0.710 | 0.876 | 0.677 | 0.772 | 0.457 | 0.468 | 0.835 | 0.912 | 0.820 | 0.703 |
| | s.e. | 0.030 | 0.015 | 0.052 | 0.018 | 0.018 | 0.023 | 0.001 | 0.011 | 0.041 | 0.007 |
| | sIPM-LFR | 0.745 | 0.880 | 0.698 | 0.809 | 0.402 | 0.587 | 0.794 | 0.920 | 0.674 | 0.710 |
| | s.e. | 0.010 | 0.025 | 0.005 | 0.021 | 0.003 | 0.006 | 0.012 | 0.024 | 0.015 | 0.019 |
| | Reg | 0.907 | 0.885 | **0.852** | 0.863 | 0.852 | 0.792 | 0.950 | 0.916 | 0.867 | 0.553 |
| | s.e. | 0.006 | 0.017 | 0.011 | 0.006 | 0.031 | 0.015 | 0.008 | 0.004 | 0.003 | 0.002 |
| | Adv | 0.885 | 0.845 | 0.840 | 0.804 | 0.795 | 0.742 | 0.944 | 0.927 | 0.867 | 0.599 |
| In-processing | s.e. | 0.008 | 0.011 | 0.010 | 0.004 | 0.003 | 0.007 | 0.004 | 0.012 | 0.026 | 0.051 |
| | Fair-Mixup | 0.894 | 0.905 | 0.829 | 0.758 | 0.904 | 0.812 | **0.953** | 0.907 | 0.781 | 0.653 |
| | s.e. | 0.015 | 0.002 | 0.001 | 0.023 | 0.031 | 0.000 | 0.002 | 0.003 | 0.012 | 0.006 |
| | Reduction | 0.905 | 0.890 | 0.840 | 0.851 | 0.848 | 0.800 | 0.950 | 0.916 | 0.867 | 0.583 |
| | s.e. | 0.008 | 0.002 | 0.011 | 0.024 | 0.014 | 0.052 | 0.019 | 0.023 | 0.019 | 0.015 |
| FTM | | **0.921** | **0.945** | 0.836 | **0.906** | **0.907** | **0.864** | 0.931 | **0.975** | **0.915** | **0.738** |
| s.e. | | 0.002 | 0.003 | 0.038 | 0.010 | 0.004 | 0.006 | 0.001 | 0.001 | 0.012 | 0.019 |

In Table 7, we further provide the $2 \times 2$ tables comparing the prediction results of the unfair model and the seven group-fair models for the five datasets. Individuals whose $\widehat{Y} = 1$ for the unfair model but $\widehat{Y} = 0$ for the fair model are thought to be treated unfairly in the context of within-group fairness. It is clear that the number of unfairly treated individuals for FTM is much less than those of baseline methods in six cases and competitive in four cases.

Table 7: $2 \times 2$ tables comparing the prediction results of the unfair model and the seven group-fair models for the five datasets. Individuals whose $\widehat{Y} = 1$ for the unfair model but $\widehat{Y} = 0$ for the fair model are those treated unfairly in view of within-group fairness. Bold faces are the best ones (the smallest number of within-group unfairness), and underlined ones are the second placers.

| Dataset | | Adult | | German | | COMPAS | | Dutch | | Law | |
|---|---|---|---|---|---|---|---|---|---|---|---|
| | | | | | | Unfair | | | | | |
| $S=0$ | | $\widehat{Y}=0$ | $\widehat{Y}=1$ | $\widehat{Y}=0$ | $\widehat{Y}=1$ | $\widehat{Y}=0$ | $\widehat{Y}=1$ | $\widehat{Y}=0$ | $\widehat{Y}=1$ | $\widehat{Y}=0$ | $\widehat{Y}=1$ |
| AE-MMD | $\widehat{Y}=0$ | 4483 | 56 | 7 | 5 | 328 | 95 | 4079 | 179 | 11 | 107 |
| | $\widehat{Y}=1$ | 125 | 238 | 9 | 48 | 180 | 197 | 35 | 1726 | 2 | 538 |
| LAFTR | $\widehat{Y}=0$ | 4525 | 15 | 7 | 5 | 358 | 65 | 4117 | **141** | 19 | 117 |
| | $\widehat{Y}=1$ | 120 | 248 | 8 | 49 | 203 | 173 | 30 | 1731 | 2 | 540 |
| sIPM-LFR | $\widehat{Y}=0$ | 4520 | 20 | 7 | 5 | 370 | 53 | 4001 | 257 | 11 | 107 |
| | $\widehat{Y}=1$ | 119 | 247 | 10 | 47 | 170 | 207 | 26 | 1735 | 1 | 541 |
| Reg | $\widehat{Y}=0$ | 4400 | 139 | 8 | 4 | 402 | 23 | 4055 | 201 | 15 | 103 |
| | $\widehat{Y}=1$ | 120 | 243 | 5 | 52 | 177 | 200 | 19 | 1742 | 1 | 539 |
| Adv | $\widehat{Y}=0$ | 4501 | 38 | 10 | **2** | 399 | 26 | 4005 | 251 | 20 | 98 |
| | $\widehat{Y}=1$ | 128 | 235 | 11 | 46 | 162 | 215 | 13 | 1748 | 4 | 536 |
| Fair-Mixup | $\widehat{Y}=0$ | 4489 | 50 | 10 | **2** | 389 | 36 | 4015 | 241 | 21 | **97** |
| | $\widehat{Y}=1$ | 130 | 233 | 6 | 51 | 180 | 197 | 15 | 1746 | 2 | 538 |
| Reduction | $\widehat{Y}=0$ | 4527 | 12 | 6 | 6 | 401 | 22 | 4005 | 251 | 11 | 105 |
| | $\widehat{Y}=1$ | 122 | 241 | 10 | 47 | 160 | 216 | 19 | 1742 | 1 | 541 |
| FTM | $\widehat{Y}=0$ | 4532 | **7** | 8 | 4 | 411 | **12** | 3993 | 265 | 10 | 106 |
| | $\widehat{Y}=1$ | 118 | 245 | 1 | 56 | 161 | 215 | 9 | 1752 | 1 | 541 |

| Dataset | | Adult | | German | | COMPAS | | Dutch | | Law | |
|---|---|---|---|---|---|---|---|---|---|---|---|
| | | | | | | Unfair | | | | | |
| $S=1$ | | $\widehat{Y}=0$ | $\widehat{Y}=1$ | $\widehat{Y}=0$ | $\widehat{Y}=1$ | $\widehat{Y}=0$ | $\widehat{Y}=1$ | $\widehat{Y}=0$ | $\widehat{Y}=1$ | $\widehat{Y}=0$ | $\widehat{Y}=1$ |
| AE-MMD | $\widehat{Y}=0$ | 7423 | 131 | 11 | 7 | 267 | 70 | 2218 | 12 | 1167 | 1168 |
| | $\widehat{Y}=1$ | 820 | 1810 | 15 | 98 | 65 | 35 | 1048 | 2787 | 1171 | 3498 |
| LAFTR | $\widehat{Y}=0$ | 7366 | 187 | 10 | 8 | 288 | 47 | 2212 | 18 | 1167 | 1177 |
| | $\widehat{Y}=1$ | 691 | 1939 | 13 | 100 | 73 | 27 | 1069 | 2766 | 1179 | 3470 |
| sIPM-LFR | $\widehat{Y}=0$ | 7440 | 113 | 11 | 7 | 301 | 34 | 2220 | 11 | 1167 | 1177 |
| | $\widehat{Y}=1$ | 901 | 1729 | 10 | 103 | 50 | 51 | 923 | 2912 | 1201 | 3458 |
| Reg | $\widehat{Y}=0$ | 7519 | 24 | 9 | 9 | 312 | 23 | 2222 | 11 | 1168 | 1176 |
| | $\widehat{Y}=1$ | 1072 | 1559 | 18 | 95 | 59 | 42 | 915 | 2920 | 1205 | 3454 |
| Adv | $\widehat{Y}=0$ | 7376 | 177 | 8 | 10 | 317 | 18 | 2228 | 3 | 1167 | **1167** |
| | $\widehat{Y}=1$ | 1080 | 1575 | 6 | 107 | 55 | 46 | 902 | 2933 | 1173 | 3497 |
| Fair-Mixup | $\widehat{Y}=0$ | 7559 | 32 | 12 | **6** | 324 | 11 | 2220 | 11 | 1167 | **1167** |
| | $\widehat{Y}=1$ | 983 | 1672 | 5 | 108 | 52 | 49 | 947 | 2888 | 1201 | 3458 |
| Reduction | $\widehat{Y}=0$ | 7535 | 18 | 11 | 7 | 300 | 35 | 2220 | 11 | 1167 | **1167** |
| | $\widehat{Y}=1$ | 1060 | 1571 | 8 | 105 | 48 | 53 | 890 | 2945 | 1158 | 3512 |
| FTM | $\widehat{Y}=0$ | 7540 | **13** | 11 | 7 | 327 | **8** | 2229 | **2** | 1167 | **1167** |
| | $\widehat{Y}=1$ | 1063 | 1568 | 3 | 110 | 47 | 54 | 895 | 2940 | 1170 | 3500 |

**Accuracy**   We provide the comparison of Acc under fixed level of $\Delta\overline{\text{DP}}$ in the top panel of Figure 8. The bottom panel of Figure 8 provides the relative change of FTM's Acc compared to averaged Acc of all baselines.

Table 8: Comparison of Accs under given levels of fairness.

| Dataset ($\Delta\overline{\text{DP}}$) | Unfair | FRL | | | In-processing | | | | FTM |
| --- | --- | --- | --- | --- | --- | --- | --- | --- | --- |
| | | AE-MMD | LAFTR | sIPM-LFR | Fair-Reg | Fair-Adv | Fair-Mixup | Reduction | |
| *Adult* ($\approx 0.060$) | 0.845 | 0.787 | 0.797 | 0.801 | 0.832 | 0.824 | 0.834 | 0.833 | 0.820 |
| *German* ($\approx 0.050$) | 0.743 | 0.729 | 0.721 | 0.721 | 0.740 | 0.748 | 0.734 | 0.735 | 0.738 |
| *COMPAS* ($\approx 0.120$) | 0.677 | 0.576 | 0.629 | 0.630 | 0.666 | 0.677 | 0.660 | 0.647 | 0.661 |
| *Dutch* ($\approx 0.030$) | 0.824 | 0.767 | 0.755 | 0.758 | 0.799 | 0.791 | 0.789 | 0.690 | 0.784 |
| *Law* ($\approx 0.040$) | 0.898 | 0.885 | 0.886 | 0.888 | 0.887 | 0.886 | 0.886 | 0.886 | 0.888 |

| Dataset | *Adult* | *German* | *COMPAS* | *Dutch* | *Law* | Average |
| --- | --- | --- | --- | --- | --- | --- |
| Relative change of accuracy | -1.3% | -0.4% | -1.1% | -1.2% | +0.2% | -0.8% |

## E.2 COMPARISON WITH FRL (SECTION 4.3)

We provide Figure 6, which is a copy of Figure 4 of larger size with standard error bands.

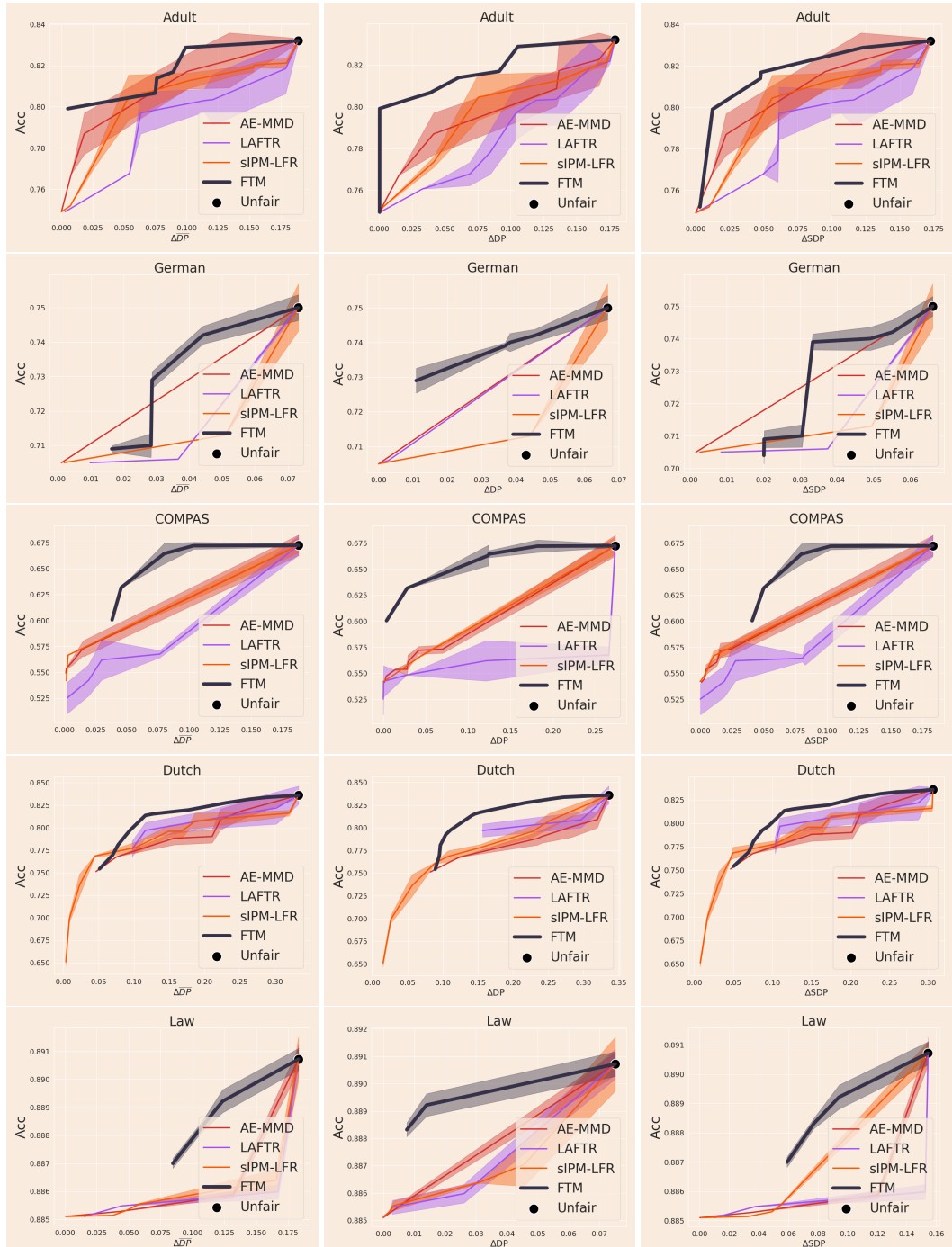

Figure 6: Larger plots of Figure 4 with standard error bands. The grid of plots becomes $3 \times 5 \to 5 \times 3$.
—: FTM, —: AE-MMD, —: LAFTR, —: sIPM-LFR.

For the results of fairness-accuracy trade-offs, see Figure 7, which presents Pareto-front lines.

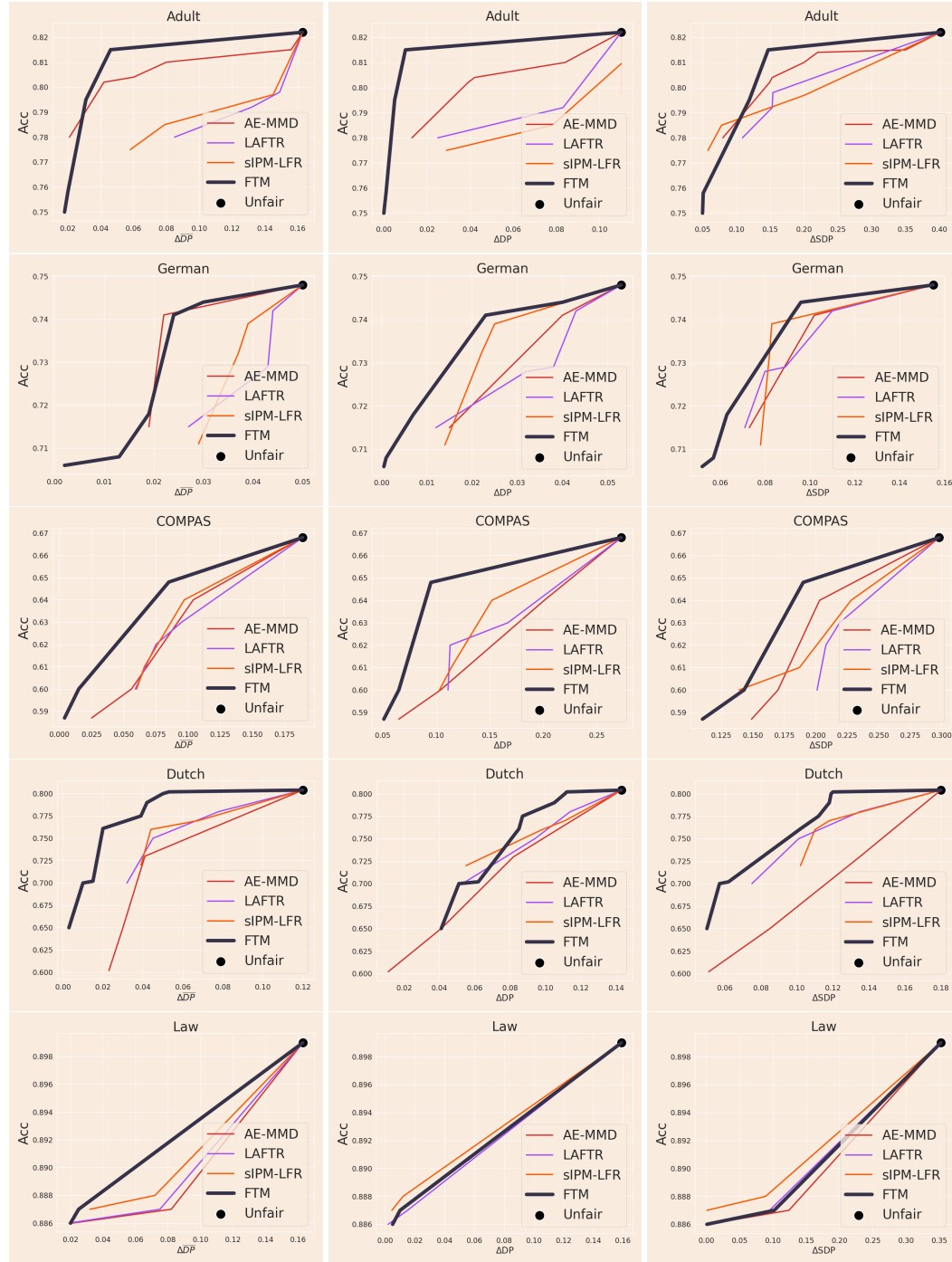

Figure 7: Linear classifier: fairness-accuracy trade-offs (1st column: Fairness = $\Delta\overline{\text{DP}}$, 2nd column: Fairness = $\Delta\text{DP}$, 3rd column: Fairness = $\Delta\text{SDP}$) represented by Pareto-front lines on (1st row) *Adult*, (2nd row) *German*, (3rd row) *COMPAS*, (4th row) *Dutch*, and (5th row) *Law* dataset. —: FTM, —: AE-MMD, —: LAFTR, —: sIPM-LFR.

### E.3 ABLATION STUDY (SECTION 4.4)

**Comparison methods**

1. **FTM-coupling**

   One can alternatively use the optimal coupling instead of the OT map for the matching function. We call this method FTM-coupling. However, since the computational complexity of finding the optimal coupling strongly depends on the sample size and dimension, it is not preferable in practice.

2. **FRL-bary**

   The FRL-bary is an FRL with the smallest transport cost. That is, we train the fair encoder by minimizing the transport cost instead of the reconstruction error. Since it is equivalent to searching the barycenter of two conditional distributions, we name it FRL-bary. We control the IPM between two representation distributions to obtain the fairness-accuracy trade-off.

3. **FRL-match**

   The FRL-match is a variant of FRL in that we use the learned fair autoencoder for matching. That is, once we obtain a fair encoder via running an FRL algorithm, we feed the encoded representation to the decoder of the opposite group. The output is now considered as the matched input of the representation's original input. Then, we train a classifier by the matching algorithm, considering the reversely decoded inputs as matched inputs. That is, for all $\mathbf{x}^{(0)}$ having $s = 0$, we map it to $\mathbf{x}^{(1,\mathrm{FRL})} := \mathbf{D}_1 \circ \mathbf{E}_0(\mathbf{x}^{(0)})$ with pre-trained autoencoders $\mathbf{D}_s \circ \mathbf{E}_s, s = 0, 1$ by the unsupervised FRL algorithm. Do the same matching for all $\mathbf{x}^{(1)}$ having $s = 1$. Then, we replace $\widehat{\mathbf{T}}_s(\mathbf{x})$ in the regularizer $\mathrm{REG}(f)$ in STEP 2 by each $\mathbf{x}^{(s,\mathrm{FRL})}$.

**Results** The results of ablation experiments for the five datasets are in Figure 8.

For FTM-coupling, we observe similar performances on *Adult* and *Dutch* datasets, but FTM beats FTM-coupling with large margins on *COMPAS* and *German* datasets. We guess the sample size is insufficient to obtain the optimal coupling close to the true one.

For FRL-bary, we expected the performance of FTM and FRL-bary to be not much different. However, we observe that the practical performances of FRL-bary are not better than that of FTM.

For FRL-match, we found that the autoencoder networks used in FRL are not properly utilized as a matching function because certain lower levels of fairness are not achieved on *Adult* and *Dutch* datasets. Furthermore, the performances are worse or never better than that of FTM.

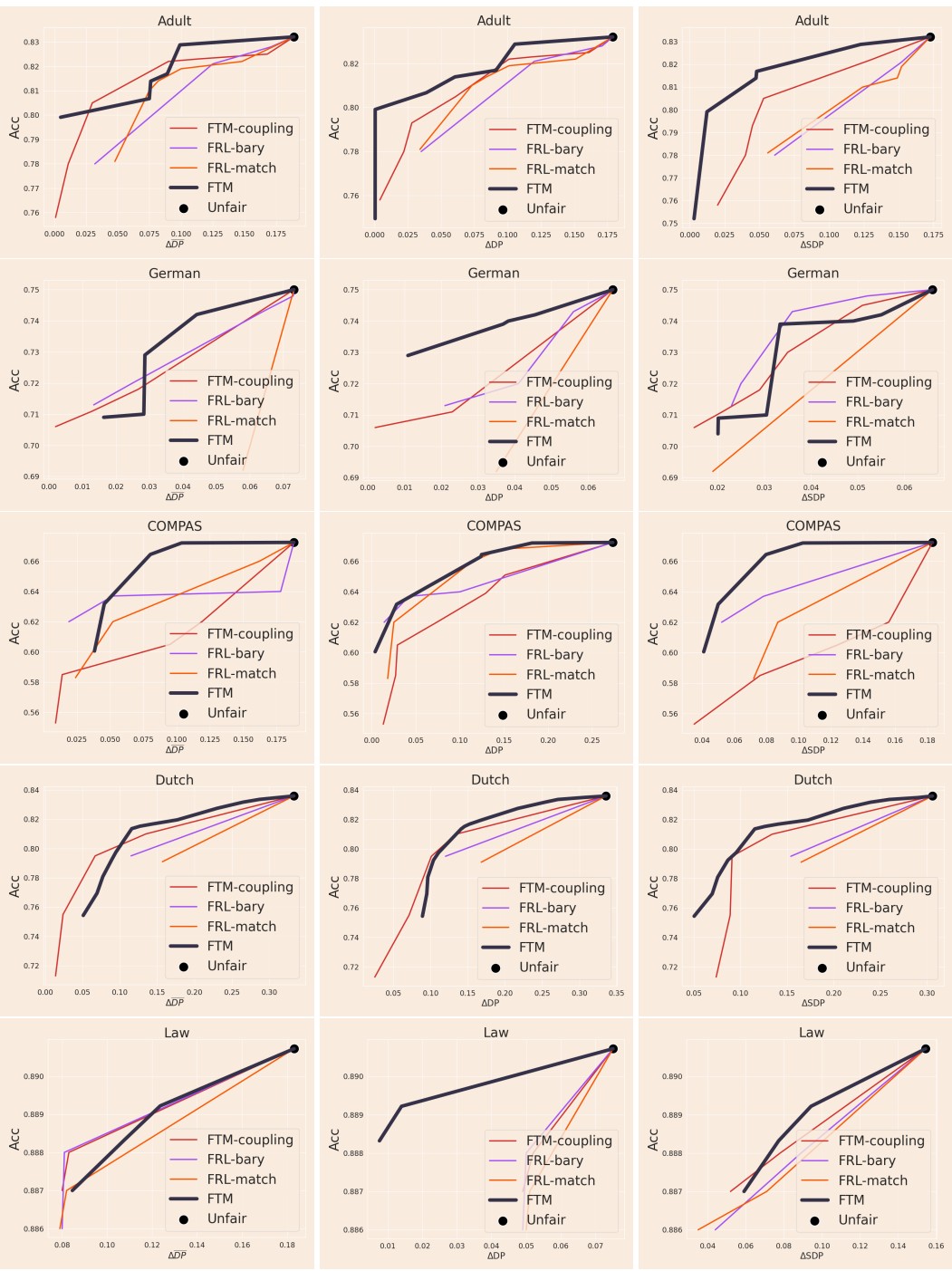

Figure 8: Comparison of accuracy-fairness trade-offs of FTM vs. FTM-coupling, FRL-bary, and FRL-match on *Adult* (1st row), *German* (2nd row), *COMPAS* (3rd row), *Dutch* (4th row), and *Law* (5th row) datasets. —: FTM, —: FTM-coupling, —: FRL-bary, —: FRL-match.

