# OpenReview forum: "Fairness Through Matching for better group fairness"
_ICLR.cc/2024/Conference — Submitted to ICLR 2024_

### Official Review · Reviewer_9FFn · 2023-10-30

**Soundness:** 4 excellent
**Presentation:** 3 good
**Contribution:** 3 good
**Rating:** 6
**Confidence:** 3

**Summary:**

The authors propose a refined group fairness regularization through matchings. Specifically, they introduce matched demographic parity, which treats individuals in the same demographic group more fairly.

**Strengths:**

The authors present a novel method, which is interesting and performs well. In addition, they propose a sound and detailed theoretical analysis of the methods.

**Weaknesses:**

Two main limitations of the proposed work and presentation stand out:
- Regarding the motivation. Subgroup discrimination is indeed a problem of Group-Fairness approaches. However, since you do not require any specific structure of the matching, it seems that you also enforce non-discrimination against features for which we want to discriminate. Take for example $X=[gender, race, skill]$ is a job application. If I apply matched group fairness on gender, then I agree that this should not lead to discrimination against e.g. african american woman. However, I am very happy with discriminating agains unskilled workers. Could you please explain how your approach would work in this case?
- While motivated from the side of group fairness, your approach has many relations to individual fairness. Specifically, (Step 1) identifies "similar" individuals while (Step 2) requires the "similar individuals" to be treated similarly by the classifier. I see that the "similar individuals" in step 2 are synthetic, but I still believe that the relation to individual fairness ought to be discussed.

**Questions:**

Some questions and comments in the order they appear in the paper:
- In the first paragraph of section 3.2, you use $\|\cdot\|^2$ to find the OT map. Which distance do you choose, and how is the performance influenced by (a) the distance and (b) the preprocessing? (no need to run experiments, I would just like to understand it better)
- Figure 4 is quite small. If you find some space increasing the size would be nice

---

> ### Author Response · Authors · 2023-11-20
> **Point-by-point reply**
>
> > **W1**:
> Regarding the motivation. Subgroup discrimination is indeed a problem of Group-Fairness approaches. However, since you do not require any specific structure of the matching, it seems that you also enforce non-discrimination against features for which we want to discriminate. Take for example $X=[gender, race, skill]$ is a job application. If I apply matched group fairness on gender, then I agree that this should not lead to discrimination against e.g. african american woman. However, I am very happy with discriminating agains unskilled workers. Could you please explain how your approach would work in this case?
>
> **Response:**
> In the above example, when the model trained by FTM with $S=gender$ similarly treats a man and a corresponding matched woman, whose skill levels are similar (also having similar race attributes), it is because the matching function searches for such pairs of a man and a woman with similar features.
> That is, the matching function has **a structure of finding the most similar pairs with respect to given features between two sensitive groups.**
>
> When $gender$ and $skill$ are independent, a group-fair learned by FTM can **freely discriminate individuals based on $skill$** since it does not affect the fairness of FTM at all.
> However, whey they are dependent, we should be careful because discrimination based on $skill$ may lead to discrimination based on $gender$.
> FTM is designed to minimize the effect of discrimination based on $skill$ to discrimination based on $gender$.
> The concept of the OT map is used for this purpose.
>
> > **W2**:
> While motivated from the side of group fairness, your approach has many relations to individual fairness. Specifically, (Step 1) identifies "similar" individuals while (Step 2) requires the "similar individuals" to be treated similarly by the classifier. I see that the "similar individuals" in step 2 are synthetic, but I still believe that the relation to individual fairness ought to be discussed.
>
> **Response:**
> A clear difference between our proposed approach and the individual fairness is that
> - our proposed approach aims to treat two individuals similarly **from different sensitive groups only** in order to achieve group fairness,
> - while the individual fairness requires to treat similar individuals similarly **regardless of sensitive attribute** (even when it is unknown).
>
> That is, similar individuals in FTM can be dissimilar in view of individual fairness, especially when the two sensitive groups are significantly different.
>
> Even though similar individuals in FTM can be dissimilar from those in view of individual fairness, we observed that FTM improves individual fairness compared to other strong group-fair models (e.g., FRL methods).
> We included this additional discussion and related empirical results in the revision (Section C.6).
>
> > **Q1**:
> In the first paragraph of section 3.2, you use $\vert \cdot \vert^{2}$ to find the OT map. Which distance do you choose, and how is the performance influenced by (a) the distance and (b) the preprocessing? (no need to run experiments, I would just like to understand it better)
>
> **Response:**
>
> - (a) Distance: We use the Euclidean distance ($L_{2}$) as the cost function when finding the (relaxed) OT map,
> as many previous works have used it with some theoretical guarantees (Seguy et al., 2018; Yang & Uhler, 2019; Hutter & Rigollet, 2021).
> In the paper, we denoted it as $\Vert \cdot \Vert^{2}.$
> Furthermore, in practice, we observed that learning the OT map with the $L_{2}$ distance tends to yield more stable results compared to the $L_{1}$ distance.
>
> - (b) Preprocessing:
> Preprocessing is necessary to make the scales of all features similar.
> This is indispensable since the scales have a significant impact on the transport cost.
> Except for the scaling, we observed that FTM is not significantly sensitive to other data preprocessing techniques (e.g., handling outliers).
> The reported performances in this paper were computed only using the standardization.
>
> > **Q2**:
> Figure 4 is quite small. If you find some space increasing the size would be nice
>
> **Response:**
> Thank you for the suggestion.
> For better readability, we selected one dataset (Adult) and increased its size in the revision.
> You can still find the remaining results in Figure 6 in Section E.2 of the Appendix.

---

> > ### Comment · Reviewer_9FFn · 2023-11-20
> > **Thank you for your clarifications**
> >
> > I thank the authors for their clarifications. I would like to retain my score.

---

### Official Review · Reviewer_diD4 · 2023-11-01

**Soundness:** 3 good
**Presentation:** 2 fair
**Contribution:** 2 fair
**Rating:** 5
**Confidence:** 4

**Summary:**

The authors introduce an algorithm to find models which satisfy both group fairness and within-group fairness called FTM or fairness through matching. This algorithm uses a new group fairness measure called MDP or matched demographic parity. They provide theoretical justification as well as some empirical results.

**Strengths:**

The authors provide good theoretical justification for their algorithm.
The authors provide some empirical results that show better performance to other similar methods - with fewer outliers when looking at subset fairness (or group fairness).

**Weaknesses:**

The language in the paper is hard to follow. Both grammatically as well as inconsistencies in terms used throughout the paper. The authors should be sure to update grammar throughout the paper (for example "a group fair model that less discriminates subsets or individuals in the same sensitive group" -> a group fairness model that discriminates less between subsets or individuals in the same sensitive group), as well as making sure their terminology throughout the paper is consistent (example: group fairness, subset fairness).

Unless I missed it in the proofs of the appendix, it is not made clear why MDP is necessary, and why total variation, strong demographic parity, or 1-Wasserstein distances should not be used. The authors provide the similarity between the measures but do not clearly state why MDP is important.

The authors make the claim that one of their contributions is the new group fairness measure MDP, but state in section 3.4 that Black et al. (2020) employs the MDP constraint. Could the authors please clarify if and how their MDP definition is different from the earlier paper.

The plots in the paper are not at all readable with very small text.

This paper seems incremental in nature, being very close to FRL, Gorsaliza et al, and pulls together techniques from other areas.

Minor nits:
It would be good to include the accuracy table in the main paper.

**Questions:**

Please see the questions associated with "Weaknesses" above.

---

> ### Author Response · Authors · 2023-11-20
> **Point-by-point reply (1)**
>
> > **W1**:
> The language in the paper is hard to follow. Both grammatically as well as inconsistencies in terms used throughout the paper. The authors should be sure to update grammar throughout the paper (for example "a group fair model that less discriminates subsets or individuals in the same sensitive group" -> a group fairness model that discriminates less between subsets or individuals in the same sensitive group), as well as making sure their terminology throughout the paper is consistent (example: group fairness, subset fairness).
>
> **Response:**
> We apologize for the poor English.
> We tried our best to enhance the readability in the revision.
> In particular, a native English colleague proofread our manuscript.
> However, the contents of our manuscript are highly technical, and thus proofreading by a non-expert may be insufficient.
> **Your understanding would be greatly appreciated.**
>
> > **W2**:
> Unless I missed it in the proofs of the appendix, it is not made clear why MDP is necessary, and why total variation, strong demographic parity, or 1-Wasserstein distances should not be used. The authors provide the similarity between the measures but do not clearly state why MDP is important.
>
> **Response:**
> For a given $f \in \mathcal{F}^{\textup{MDP}} (\delta),$ we define $ \mathbf{T}\_{s}^{f} = argmin_{\mathbf{T}\_{s} \in \mathcal{T}\_{s, 0}} \mathbb{E}\_{s} \vert f( \mathbf{X}, s ) - f( \mathbf{T}\_{s}(\mathbf{X}), s' ) \vert $ as the corresponding matching function (Section 3).
>
> We would like to remind that the aim of this paper is to eliminate undesirable and problematic group-fair models  (e.g., subset unfairness, within-group unfairness).
> To achieve this objective, we discard $f \in \mathcal{F}^{\textup{MDP}} (\delta)$ those whose corresponding matching function ($\mathbf{T}_{s}^{f}$) has too large transport cost (Section 3.2).
>
> In contrast, **such restriction would not be possible for the total variation or 1-Wasserstein distance**, at least to the best of our knowledge.
> To us, it was surprising that MDP is similar to strong demographic parity with respect to the total variation or 1-Wasserstein distance (i.e., Proposition 2.1), because any distance between the distributions of two sensitive groups is not explicitly used in the definition of MDP.
>
> > **W3**:
> The authors make the claim that one of their contributions is the new group fairness measure MDP, but state in section 3.4 that Black et al. (2020) employs the MDP constraint. Could the authors please clarify if and how their MDP definition is different from the earlier paper.
>
> **Response:**
> We apologize for the rough and insufficient description of FlipTest (Black et al. (2020)).
> We did not pay much attention to FlipTest because it is not used for learning group-fair model.
> Indeed, the measure used in FlipTest is similar to the MDP constraint, but they are not exactly the same.
> Below, we explain how **the measures for FlipTest and the MDP constraint are fundamentally different.**
>
> For a given prediction model $f$ and $s \in \{0, 1\},$ FlipTest first finds two sets of individuals whose predictions are flipped, as defined by
> $ F\^{+}(f; s) := \\{ i : s_i = s, \mathrm{I} ( f(\mathbf{x}_i, s) > 0 ) > \mathrm{I}( f(\hat{\mathbf{T}}\_{s}(\mathbf{x}_i), s') > 0 ) \\} $
> and
> $ F\^{-}(f; s) := \\{ i : s_i = s, \mathrm{I} ( f(\mathbf{x}_i, s) > 0 ) < \mathrm{I} ( f(\hat{\mathbf{T}}\_{s}(\mathbf{x}_i), s') > 0 ) \\}, $
> where $\hat{\mathbf{T}}_s, s \in \\{ 0,1 \\}$ are the OT map.
> Using these sets, FlipTest measures the unfairness of $f$ as $|F\^{+}(f; s)| - |F\^{-}(f; s)|$.
>
> One may argue that our proposed constraint ($REG_{s}(f)$ in STEP 2) is similar to $|F\^{+}(f; s)| - |F\^{-}(f; s)|$ from FlipTest.
> However, it is not the case because the regularization term induced by FlipTest would be formulated as:
> $
> \left\vert \mathbb{E}\_{n, s} \left( \mathrm{I}( f(\mathbf{X}, s) > 0 )\right) - \mathbb{E}\_{n, s} \left( \mathrm{I} ( f(\hat{\mathbf{T}}\_{s}(\mathbf{X}), s') > 0 ) \right) \right\vert
> $
> or
> $
> \left\vert \mathbb{E}\_{n, s} \left( f(\mathbf{X}, s)\right) -  \mathbb{E}\_{n, s} \left(f(\hat{\mathbf{T}}_{s}(\mathbf{X}), s') \right) \right\vert,
> $
> which is completely different from our regularization term:
> $ REG\_{s}(f) = \mathbb{E}\_{n, s} \vert f(\mathbf{X}, s) - f(\hat{\mathbf{T}}\_{s}(\mathbf{X}), s') \vert. $
>
> **The former (FlipTest) is the difference of the expectations, while the latter (FTM) is the expectation of the (absolute) differences.**
> This seemingly tiny difference would make big differences in many ways.
> For example, the measure of **FlipTest would not imply the strong group fairness, while our regularization term does** (i.e., Proposition 2.1).
> Moreover, it would **not be clear whether the measure of FlipTest improves better subset/within-group fairness.**
>
> In conclusion, the only thing FlipTest and MDP share is the use of the OT map, but they are fundamentally different.
> We added these additional explanation in the revision.

---

> ### Author Response · Authors · 2023-11-20
> **Point-by-point reply (2)**
>
> > **W4**:
> The plots in the paper are not at all readable with very small text.
>
> **Response:**
> We have made modifications in the revision by increasing the font sizes to improve readability (Figures 3 and 4).
>
> > **W5**:
> This paper seems incremental in nature, being very close to FRL, Gorsaliza et al, and pulls together techniques from other areas.
>
> **Response:**
> The key contribution of this paper is to verify the theoretical properties of MDP, including its relationship with existing well-known group fairness measures, including the strong demographic parity with respect to the total variation and 1-Wasserstein distance (Proposition 2.1).
> In addition, we showed that **FTM can improve subset and within-group fairness (Sections 3.2 and 4.2).**
>
> Similarity between FTM and FRL is **a kind of coincidence**, since the motivations of these two approaches are quite different.
> FRL methods have been developed to obtain fair representations which can be used for downstream tasks.
> In contrast, FTM is developed to eliminate undesirable group-fair models, specifically subset/within-group unfair models.
>
> Our empirical results amply illustrate that **FTM does improve subset/within-group fairness compared to FRL methods**.
> In our numerical studies, we compared FTM with FRL not only because FTM can be interpreted as a FRL algorithm but also because the both methods can learn strongly group-fair models.
> Additionally, there are other advantages of FTM over FRL, which are highlighted in Section 4.3.
>
> > **W6**:
> Minor nits: It would be good to include the accuracy table in the main paper.
>
> **Response:**
> Thank you for the suggestion.
> Following your thoughtful suggestion, we added the table showing the relative changes of accuracy in the revision (Table 2).

---

> ### Author Response · Authors · 2023-11-28
> **Dear reviewer diD4**
>
> First of all, we appreciate your thoughtful comments on our work.
>
> We have tried to address your concerns appropriately, but unfortunately, we have not yet received a response to our rebuttal, even though the discussion deadline is approaching (extended for this paper until **December 1st**).
>
> Therefore, we would like to recap our key responses to your comments as follows:
>
> - MDP with the matching function offers advantages over the total variation and Wasserstein distance, particularly in terms of eliminating problematic group-fair models.
> - FlipTest and FTM use fundamentally different regularizations (or measures), with FTM providing the capability to effectively learn group-fair models with improved subset/within-group fairness.
> - FRL and FTM methods serve different motivations and purposes, and we have emphasized the advantages of FTM over FRL, as shown in the paper.
>
> We kindly request you to consider our rebuttal responses and the revised manuscript carefully.
>
> Thank you for your time and dedicated efforts in reviewing this study.

---

### Official Review · Reviewer_ZKn5 · 2023-11-29

**Soundness:** 3 good
**Presentation:** 2 fair
**Contribution:** 3 good
**Rating:** 6
**Confidence:** 2

**Summary:**

The authors raise a concern that existing group fairness notions do not protect against unwarranted within-group performance disparities. The authors propose a new Matched Demographic Parity (MDP) fairness measure and accompanying learning approach — Fairness Through Matching — which is designed to improve within group fairness. The authors justify their approach theoretically and via experiments on several benchmark datasets.

**Strengths:**

This work provides a novel Matched Demographic Parity fairness measure and establishes connections with existing measures such as strong demographic parity. Matching for improved group fairness is an interesting and under explored area, and the authors develop a technically sound framework that demonstrates promising empirical performance. I also appreciated the ablation of different matching approaches in the experiments section.

**Weaknesses:**

Weaknesses (ordered by importance):
- Given the focus of this work, I would expect a more granular comparison against existing multi-calibration fairness notions and leaning algorithms. The stated goal of this work — to “find group-fair model that discriminates less between subsets or individuals in the same sensitive group” — bears strong resemblance to muliccalibration, which provides a guarantee that holds across many overlapping subsets of a protected group. Indeed, the authors’ definition of “subset fairness” is very similar to the formal definition of multi-calibration (the specified definition seems to specify a maximum violation over subsets rather than specifying a constraint that holds over all intersectional subgroups). A direct comparison against multi-calibrated predictors (theoretically and in experiments) is needed. If this is not the case, I encourage the authors to clearly differentiate early on in the work, as other readers are likely to have similar questions.

- I also have concerns regarding the scalability and robustness of the proposed approach in real-world settings of interest. It seems that a matching style approach would be challenging when subgroups are highly imbalanced, and that performing matching across multiple intersectional subgroups would also be challenging.

- There is an opportunity to strengthen the motivation of the work. I appreciate the authors’ approach of providing a toy example to highlight issues with group-fair models. However, Fig 1, speaks to similar known issues with intersectionality in fairness. It would be helpful to illustrate the intuition as to why matching is a useful approach for addressing this problem.

- Benchmarking fairness approaches via the COMPAS dataset has several known limitations [1]. I don’t have an issue with using this dataset in this work given the technical focus of the paper, but do think that an explicit disclaimer acknowledging these issues is warranted in the experiments section.

**Questions:**

Please see points raised in the weaknesses section above.

---

> ### Author Response · Authors · 2023-11-29
> **Point-by-point reply (1)**
>
> ### We greatly appreciate your effort in reviewing this paper. We have tried to address your concerns and comments point by point, as follows. Additionally, you will find the revisions highlighted in blue-colored text in the updated PDF document. Your consideration would be greatly appreciated.
>
> > **W1**: Given the focus of this work, I would expect a more granular comparison against existing multi-calibration fairness notions and leaning algorithms. The stated goal of this work — to “find group-fair model that discriminates less between subsets or individuals in the same sensitive group” — bears strong resemblance to muliccalibration, which provides a guarantee that holds across many overlapping subsets of a protected group. Indeed, the authors’ definition of “subset fairness” is very similar to the formal definition of multi-calibration (the specified definition seems to specify a maximum violation over subsets rather than specifying a constraint that holds over all intersectional subgroups). A direct comparison against multi-calibrated predictors (theoretically and in experiments) is needed. If this is not the case, I encourage the authors to clearly differentiate early on in the work, as other readers are likely to have similar questions.
>
> **Response**:
> Thank you for discussing the concept of `multicalibration' as presented in (Hebert-Johnson et al., 2018), and others.
> However, the fairness notion we have focused on (e.g., demographic parity) and the multicalibration notion **fundamentally differ**.
> We would like to provide clarifications supporting the claim as follows:
> - First, **multicalibration aims to mitigate accuracy(calibration)-based disparities within subsets, whereas our focus is on reducing the conventional statistical disparity between two sensitive groups** (e.g., demographic parity - the gap between positive prediction ratios from different sensitive groups).
> In our paper, the terms *group-fair* and *group fairness* are related to the statistical disparity, not the gap between the predictions and the true probabilities as considered in multicalibration.
>
> - To add a note, it is important to recognize that there exist two main streams of *fairness*; (1) statistical parity-based and (2) accuracy-based, which are generally **incompatible** (Kleinberg et al., 2017, Pleiss et al., 2017).
> Briefly, this is because multicalibration focuses on accurate predictions (reducing the gap between predictions and true probabilities), whereas sacrificing accuracy is unavoidable when we aim to achieve statistical parity fairness notions (Fish et al., 2016, Berk et al., 2017).
>
> - By a similar reason, subset fairness and multicalibration are also distinct concepts.
> **Subset fairness is a statistical parity notion defined over subsets** (its definition is provided in Section 3.2 in a solid manner).
> To clarify, we would like to revise the statement as follows:
> `find a group-fair model that discriminates less between subsets or individuals in the same sensitive group`
> $\rightarrow$
> `find a group-fair model that discriminates less between sensitive groups within a subset or among individuals in the same sensitive group`.
> We have modified the statement in the revision.
>
> (Kleinberg et al., 2017) https://arxiv.org/abs/1609.05807
>
> (Pleiss et al., 2017) https://arxiv.org/abs/1709.02012
>
> (Fish et al., 2016) https://arxiv.org/abs/1601.05764
>
> (Berk et al., 2017) https://arxiv.org/abs/1706.02409
>
> > **W2**: I also have concerns regarding the scalability and robustness of the proposed approach in real-world settings of interest. It seems that a matching style approach would be challenging when subgroups are highly imbalanced, and that performing matching across multiple intersectional subgroups would also be challenging.
>
> **Response**:
> - Firstly, we would like to clearly describe the flow of our proposed framework to avoid any misunderstanding.
> FTM **does not match individuals from an intersectional subgroup or subset**; instead, it applies the matching framework to the entire sensitive groups.
> Hence, **the risk of encountering such issues** related to data imbalance between sensitive groups **is not specifically higher for FTM** compared to other fairness algorithms; it is comparable.
>
> - Secondly, while scalability and robustness are essential requirements for any ML algorithms, we believe that these properties are quite general in nature. Consequently, we also believe that such properties are not specifically required for our proposed approach but may also required for other existing fairness algorithms or other ML algorithms as well.
>
> - On the other hand, as a potential solution to address your concern, we anticipate that various techniques such as oversampling could help mitigate the data imbalance problem.

---

> ### Author Response · Authors · 2023-11-29
> **Point-by-point reply (2)**
>
> > **W3**: There is an opportunity to strengthen the motivation of the work. I appreciate the authors’ approach of providing a toy example to highlight issues with group-fair models. However, Fig 1, speaks to similar known issues with intersectionality in fairness. It would be helpful to illustrate the intuition as to why matching is a useful approach for addressing this problem.
>
> **Response**:
> We appreciate your valuable suggestion aimed at improving the quality of our work.
> To provide a more intuitive explanation, when we use a reasonable matching function as proposed (i.e., OT map), we can effectively mitigate discrimination against individuals or subsets with similar features.
> We have modified the corresponding illustration in Figure 1 in the revision.
>
>
> > **W4**: Benchmarking fairness approaches via the COMPAS dataset has several known limitations [1]. I don’t have an issue with using this dataset in this work given the technical focus of the paper, but do think that an explicit disclaimer acknowledging these issues is warranted in the experiments section.
>
> **Response**:
> Thank you for your comment.
> We have added the additional disclaimer in the revision.

---

> ### Comment · Reviewer_ZKn5 · 2023-12-02
>
> I would like to thank the authors for considering and responding to my points.
>
> I acknowledge the key difference between the demographic parity notions studied by the authors and the accuracy-based measures. My underlying concern is that prior work has developed alternative approaches for auditing and post-processing algorithms to satisfy fairness constraints over computationally large subgroups, and I am not convinced that the proposed approach is sufficiently differentiated against these.
> - The authors do draw connections to Kearns et al.(2018) in Section C.2 and suggest “The problem we consider is different in the sense that we do not know $C$ (a given collection of subsets) in advance”. However, it isn’t clear to me that this is a key differentiating factor against this work. While the authors include a theoretical argument in Thm 3.1., it would be helpful to include a direct comparison in the experiments.
>
> -Work has also shown that it is possible to leverage boosting-style approaches for multi-accuracy guarantees to satisfy demographic parity constraints (see Diana et al, 2022 Sec 1.1).
>
> Thank you for clarifying your approach regarding the imbalance issue I mentioned. This addresses this concern.
>
> In light of the authors’ response I have increased my score. However, I will reduce my confidence because I am not confident of my assessment of the comparison against Kearns et al. (2018). The meta-reviewer should weight my review accordingly.
>
>
> Diana et al.,(2022): https://dl.acm.org/doi/fullHtml/10.1145/3531146.3533180#BibPLXBIB0014
>
> Kearns et al.,(2018): https://arxiv.org/abs/1711.05144

---

### Author Response · Authors · 2023-11-20
**Common reply**

> ### **To all the reviewers**:

We sincerely appreciate the valuable and constructive feedback received from all the reviewers.
We have addressed each reviewer's comments in a detailed, **point-by-point manner.**
In the updated PDF document, you will find the revisions highlighted in **blue-colored text.**
We kindly ask you to consult the revised file for these updates.

---

### Meta-Review · Area_Chair_ddz2 · 2023-12-12

**Metareview:**

This paper looks at group unfairness in ML.  There was, and remains, considerable confusion amongst the reviewers around the paper's placement in the fairness literature -- e.g., group vs individual fairness (of which there are flavors of both), multi-calibration, the optimal transport literature vs the proposed MDP test, and so on.  Coupled with concerns around related work and comparison to other baselines, this paper's impact could be improved dramatically by placing it better in the literature, and describing what it offers relative to what exists already.

**Justification For Why Not Higher Score:**

Reviewers had substantial issues with the exposition of the paper, which may color their scores.  Still, reviewers were lukewarm even after the rebuttal.  While, to my read, no reviewers surfaced a fatal flaw in the paper, all reviewers complained about completeness -- benchmarking, reviewing, etc -- which leads me to recommend rejection.

**Justification For Why Not Lower Score:**

N/A

---

### Decision · Program_Chairs · 2024-01-16

Reject